# Earth-Agent: Unlocking the Full Landscape of Earth Observation with Agents

**Peilin Feng**[1][*][‡]   **Zhutao Lv**[2,1][*][‡]   **Junyan Ye**[2,1][‡]   **Xiaolei Wang**[2]   **Xinjie Huo**[2]
**Jinhua Yu**[2]   **Wanghan Xu**[1]   **Wenlong Zhang**[1]   **Lei Bai**[1]   **Conghui He**[1]   **Weijia Li**[3,2,1][†]
[1] Shanghai Artificial Intelligence Laboratory   [2] Sun Yat-sen University
[3] Tsinghua Shenzhen International Graduate School, Tsinghua University

 **Github:**       https://github.com/opendatalab/Earth-Agent
 **Dataset:**      https://huggingface.co/datasets/Sssunset/Earth-Bench
 **Project Page:** https://opendatalab.github.io/Earth-Agent/

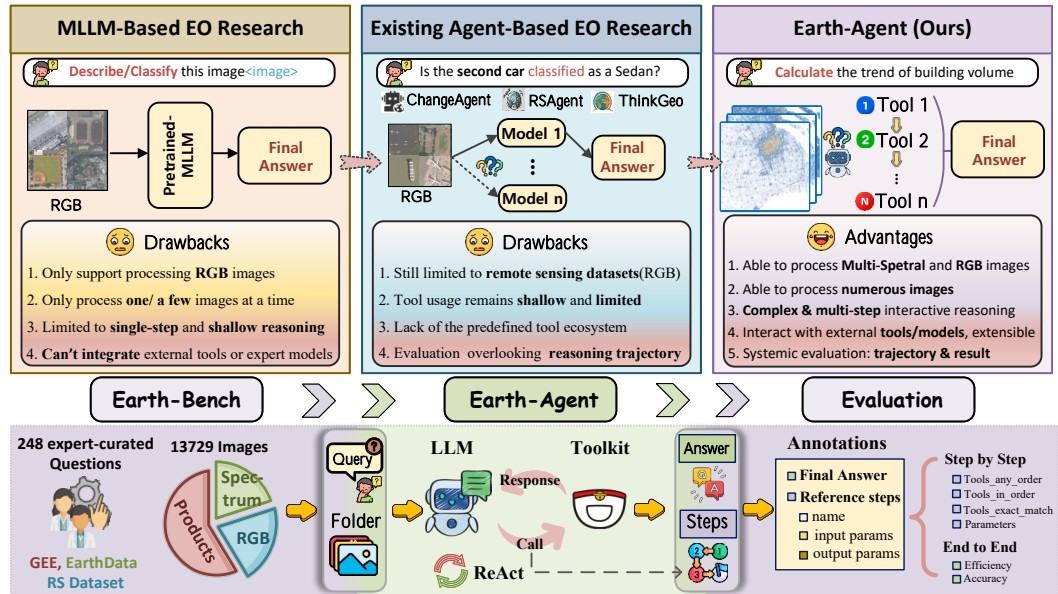

Figure 1: **Overview of our work:** The top panel contrasts prior paradigms: MLLM-based EO research (left), Existing agent-based EO research (middle), and our Earth-Agent (right). The bottom panel illustrates our contributions, including Earth-Bench construction, Earth-Agent ReAct with the predefined toolkit, and dual-level evaluation of both reasoning trajectories and final results.

## Abstract

Earth observation (EO) is essential for understanding the evolving states of the Earth system. Although recent MLLMs have advanced EO research, they still lack the capability to tackle complex tasks that require multi-step reasoning and the use of domain-specific tools. Agent-based methods offer a promising direction, but current attempts remain in their infancy, confined to RGB perception, shallow reasoning, and lacking systematic evaluation protocols. To overcome these limitations, we introduce Earth-Agent, the first agentic framework that unifies RGB and spectral EO data within an MCP-based tool ecosystem, enabling cross-modal, multi-step, and quantitative spatiotemporal reasoning beyond pretrained MLLMs. Earth-Agent supports complex scientific tasks such as geophysical parameter retrieval and quantitative spatiotemporal analysis by dynamically invoking expert

---

[*]Equal contribution.
[‡]This work was done during their internship at OpenDataLab of Shanghai AI Lab.
[†]Corresponding author. E-mail: liweijia@sz.tsinghua.edu.cn

tools and models across modalities. To support comprehensive evaluation, we further propose Earth-Bench, a benchmark of 248 expert-curated tasks with 13,729 images, spanning spectrum, products and RGB modalities, and equipped with a dual-level evaluation protocol that assesses both reasoning trajectories and final outcomes. We conduct comprehensive experiments varying different LLM backbones, comparisons with general agent frameworks, and comparisons with MLLMs on remote sensing benchmarks, demonstrating both the effectiveness and potential of Earth-Agent. Earth-Agent establishes a new paradigm for EO analysis, moving the field toward scientifically grounded, next-generation applications of LLMs in Earth observation. More information about Earth-Agent can be found at `https://github.com/opendatalab/Earth-Agent`

# 1 INTRODUCTION

Earth observation (EO) (Transon et al., 2018; Kokkoris et al., 2024; Li et al., 2023a) plays a critical role in understanding the evolving states of the Earth system in spatial and temporal dimensions (Anderson et al., 2017; Li et al., 2024a; Brown et al., 2025), and has been successfully applied to urban planning (Shaker et al., 2019), agriculture (Wójtowicz et al., 2016), resources management (Li et al., 2020; Wang et al., 2025a), building extraction (Li et al., 2023b; 2024c), disaster monitoring (Joyce et al., 2009; Van Westen, 2000), etc. Typically, EO data is categorized into two types (Samadzadegan et al., 2025): **Perceptual data**, such as *RGB Imagery (RGB)* aligned with human vision, and **Raw Observational Data**, including *Raw Spectral Data (Spectrum)* and *Processed Earth Products (Products)* stored in geodatabases such as Google Earth Engine (GEE)[*] and NASA Earthdata[†]. Both types of data are indispensable for EO research: perceptual data provides intuitive and human-interpretable insights, while raw observational data offers rich spectral and spatiotemporal information that enables quantitative analysis (Valipour et al., 2025; Xiong et al., 2022).

In recent years, multimodal large language models (MLLMs) have achieved excellent performance on classical **remote sensing perceptual tasks** such as VQA (Kuckreja et al., 2024; Muhtar et al., 2024), scene classification (Kuckreja et al., 2024; Muhtar et al., 2024; Liu et al., 2024c; Wang et al., 2024e; Hu et al., 2025b; Zhan et al., 2025), object detection (Zhang et al., 2024b), and semantic segmentation (Mall et al., 2023; Guo et al., 2024a). However, despite their promising results, existing MLLM-based EO research still faces several fundamental drawbacks: **(1)** they cannot process diverse EO modalities beyond RGB, such as thermal infrared (TIR), synthetic aperture radar (SAR), or hyperspectral imagery (Zhang et al., 2024b); **(2)** they typically operate on only one or a few images at a time (Li et al., 2024b), making it difficult to scale to large EO corpora; **(3)** they are limited to executing only single-step or shallow reasoning like VQA and classification, struggling with complex multi-hop analytical tasks; and **(4)** their reasoning is bounded by the static knowledge encoded in pretrained parameters, without the ability to integrate external scientific tools or expert models, making it difficult to extend beyond the generic capabilities of the foundation model; This naturally raises the question: *how can we move beyond basic RGB perception and single-step reasoning to design models that integrate diverse EO modalities and support complex multi-step scientific analysis?*

Tool-augmented LLM agents represent a promising trajectory beyond MLLMs (Xi et al., 2025; Sun et al., 2025; Si et al., 2024; Tian et al., 2024; Tang et al., 2025). Unlike MLLMs that are restricted to RGB inputs, simple reasoning, and limited image contexts, agents are not inherently constrained by input modality or data volume (Xie et al., 2024; Gao et al., 2024). By leveraging the reasoning capabilities of LLMs and dynamically interacting with external tools (Xu et al., 2025), they can flexibly process diverse EO modalities, perform multi-step analytical reasoning, and integrate domain-specific tools and expert models that go beyond the scope of the pretrained MLLM model (Ding et al., 2025; Wang et al., 2024c). This mechanism directly tackles the core weaknesses of MLLMs, extending beyond RGB to diverse modalities, scaling from single-image inputs to tasks involving hundreds of images, advancing from shallow perception to multi-step reasoning, and bridging LLMs with external scientific tools for domain-specific analysis.

However, existing agent-based research in Earth science is still at an early stage (Pantiukhin et al., 2025), with existing attempts largely confined to perceptual tasks such as change detection (Liu

---

[*]`https://earthengine.google.com`
[†]`https://search.earthdata.nasa.gov`

et al., 2024b; 2025) and classification (Xu et al., 2024a; Hu et al., 2025a), often emphasizing caption ability rather than scientific analysis. Efforts on Raw Observational Data are even more limited. Uni-vEarth (Kao et al., 2025) considers EO data from GEE but operates essentially as a code generation agent, without implementing genuine tool calling, making it difficult to handle complex and realistic geoscientific analysis tasks that require professional tool use. These efforts reveal several key limitations: **(1)** current EO agents support only limited data modalities, with most efforts still centered on conventional remote sensing datasets dominated by RGB imagery (Xu et al., 2024a); **(2)** their tool usage remains shallow, limited to a few expert models and reasoning steps, even some agents lack a predefined tool ecosystem, making them insufficient for complex analytical workflows (Shabbir et al., 2025); and **(3)** their evaluation remains unsystematic, with emphasis only on final answers while overlooking reasoning trajectory. This raises another question: *how can we design an EO agent with a structured tool ecosystem and systematic evaluation, capable of bridging perceptual and spectral data like Earth scientists?*

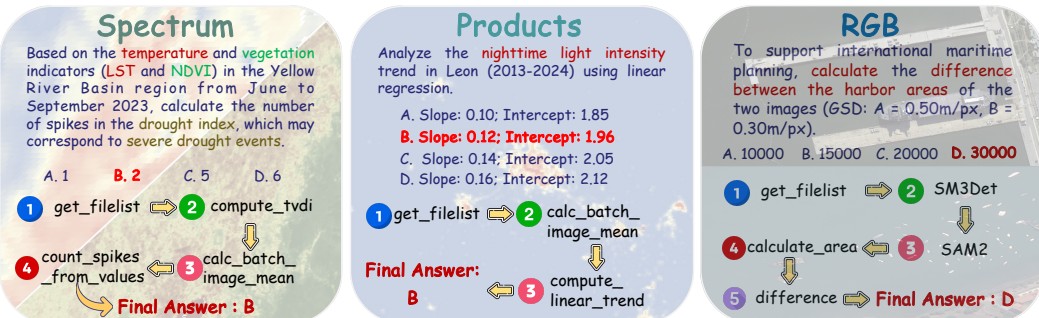

Figure 2: Earth-Agent solving tasks across Spectrum, Products, and RGB data through multi-step reasoning with expert tool calls.

To address these questions and unlock the full landscape of EO, we propose **Earth-Agent**, an agentic framework that unifies perceptual and spectral EO data within a single architecture in section 3. By coupling LLM reasoning with a structured toolkit in 3.2, Earth-Agent supports diverse modalities and complex multi-step analysis, enabling agents to tackle real-world geoscientific tasks beyond the limits of existing MLLMs and EO agents. Concretely, Earth-Agent integrates 104 specialized tools, built upon the **Model Context Protocol (MCP)** (Hou et al., 2025; Ray, 2025) for interoperability, and grouped into five domain-specific tool kits: **Index**, **Inversion**, **Perception**, **Analysis**, and **Statistics**. This structured design not only enables the agent to go beyond classical perceptual tasks toward quantitative analysis and spatiotemporal reasoning, but also makes the framework easily extensible with additional domain-specific tools. To systematically evaluate its effectiveness, we further introduce **Earth-Agent Benchmark (Earth-Bench)** in section 4, which reflects realistic EO workflows and supports both **Auto-Planning** and **Instruction-Following** query regimes, together with a **dual-level evaluation** protocol that measures reasoning trajectories as well as final outcomes. We comprehensively evaluate Earth-Agent by varying LLM backbones on Earth-Bench, comparing with general agents, and benchmarking against MLLMs on remote sensing datasets.

To sum up, our main contributions are summarized as follows:

- We propose Earth-Agent, the first agentic framework for EO, built upon the MCP and a ReAct (Yao et al., 2023) reasoning, integrating 104 specialized tools and expert models within predefined tool ecosystem, while remaining easily extensible with additional domain-specific tools and models.

- We construct Earth-Bench, a benchmark of 248 expert-curated questions with 13,729 images, spanning perceptual and spectral modalities beyond RGB. Each question requires multi-step reasoning with explicit tool use, and the benchmark supports a dual-level evaluation protocol that assesses both reasoning trajectories and final answers.

- Through comprehensive evaluation, we show that Earth-Agent substantially outperforms general agents such as Operator (OpenAI, 2025b) and Manus (Shen et al., 2025) on Earth-specific tasks in Earth-Bench, and also surpasses remote sensing MLLMs on remote sensing benchmarks, demonstrating both its effectiveness and potential for advancing EO research.

## 2 RELATED WORK

**MLLM-based Earth Observation Research**  The rise of multimodal large language models (MLLMs) has stimulated growing interest in their use for Earth observation (EO) (Aleissaee et al., 2023; Lu et al., 2025; Li et al., 2024b). Early studies mainly explored captioning (Hu et al., 2025b) and question answering (Kuckreja et al., 2024) for single remote sensing images (Shi & Zou, 2017; Wang et al., 2020), aiming to align visual features with natural language. With the availability of larger datasets (Xiong et al., 2022; Zhou et al., 2025) and stronger backbones (Team, 2024; Liu et al., 2024d), subsequent works extended this paradigm to broader perception tasks: for instance, GeoChat (Kuckreja et al., 2024) enabled interactive scene understanding, while RS-GPT (Hu et al., 2025b) combined captioning with visual question answering. More recently, simple temporal reasoning has been introduced, with ChangeCLIP (Dong et al., 2024) addressing bi-temporal change captioning and SkyEye-GPT (Zhan et al., 2025) extending to video-based analysis. However, the scope of MLLM-based EO research remains narrow: existing approaches are still centered on RGB imagery and struggle with complex multi-step reasoning without domain-specific tool integration.

**Agent-based Earth Observation Research.**  Tool-augmented agents have gained traction in general AI, achieving remarkable progress in domains such as code generation (Qian & Cong, 2023; Zhang et al., 2024a), web search (Xu et al., 2024b), and video understanding (Ren et al., 2025; Wang et al., 2024d), but their application to Earth observation (EO) is still at an early stage (Kao et al., 2025). Early systems such as Change-Agent (Liu et al., 2024b) focus on bi-temporal change detection, while RS-ChatGPT (Guo et al., 2024a) and RS-Agent (Xu et al., 2024a) combine LLMs with pretrained detectors or tool suites for scene classification, detection, and segmentation. More recently, ThinkGeo (Shabbir et al., 2025) introduces agentic workflows for simple geospatial calculations on perceptual data, and UnivEarth (Kao et al., 2025) requires LLMs to generate GEE code for spectral analysis, with high execution failure rates. Despite these advances, existing EO agents remain constrained: they operate mainly on RGB perception tasks, rely on remote sensing models for simple reasoning that does not extend to multi-step analysis, and lack a predefined tool ecosystem, making them insufficient for complex real-world geoscientific workflows. Moreover, current benchmarks cover limited task types and annotations, lacking systematic evaluation protocols that assess both the correctness of outcomes and the quality of reasoning trajectories. As a result, current frameworks remain limited in modality coverage, constrained to shallow reasoning with remote sensing models, and hindered by the absence of a predefined tool ecosystem, highlighting the necessity for EO agents and benchmarks that support diverse data, multi-step analytical workflows, and systematic evaluation.

## 3 EARTH-AGENT FRAMEWORK

In this section, we detail the operation mechanisms of Earth-Agent. We first formulate its operation pipeline as a ReAct-style (Yao et al., 2023) Partially Observable Markov Decision Processes (POMDP) formulation (Huang et al., 2024; Chala et al., 2025) in section 3.1 , including the observation process, policy reasoning and memory update, as shown in Figure 3. Then we introduce the functionality of the specialized tool kits that enable EO analysis across perceptual and spectral data in section 3.2. Finally, we define the dual-level evaluation protocol, which assesses EO agents in both end-to-end and step-by-step modes to evaluate not only final accuracy but also reasoning trajectories in section 3.3.

### 3.1 OPERATION MECHANISMS

Earth-Agent receives a task goal $g$, interprets user queries and intermediate tool outputs, and selects actions from a modular toolkit to progressively solve the task. This process is formulated as a POMDP, defined by the tuple $\langle g, \mathcal{S}, \mathcal{A}, \mathcal{O}, \mathcal{T} \rangle$, where $g$ is the task goal, $\mathcal{S}$ is the state space (unobservable environment status such as geospatial data files or raster values), $\mathcal{A}$ is the action space (tool calls in the kit), $\mathcal{O}$ is the observation space (outputs returned by tools, including text, numerical values, and images), and $\mathcal{T} : \mathcal{S} \times \mathcal{A} \rightarrow \mathcal{S}$ is the state transition function.

At each time step $t$, given a policy $\pi$, the agent selects an action conditioned on the goal $g$ and its interaction history, which records past observations and actions:

$$m_t = (o_0, a_0, o_1, a_1, \ldots, o_t),$$

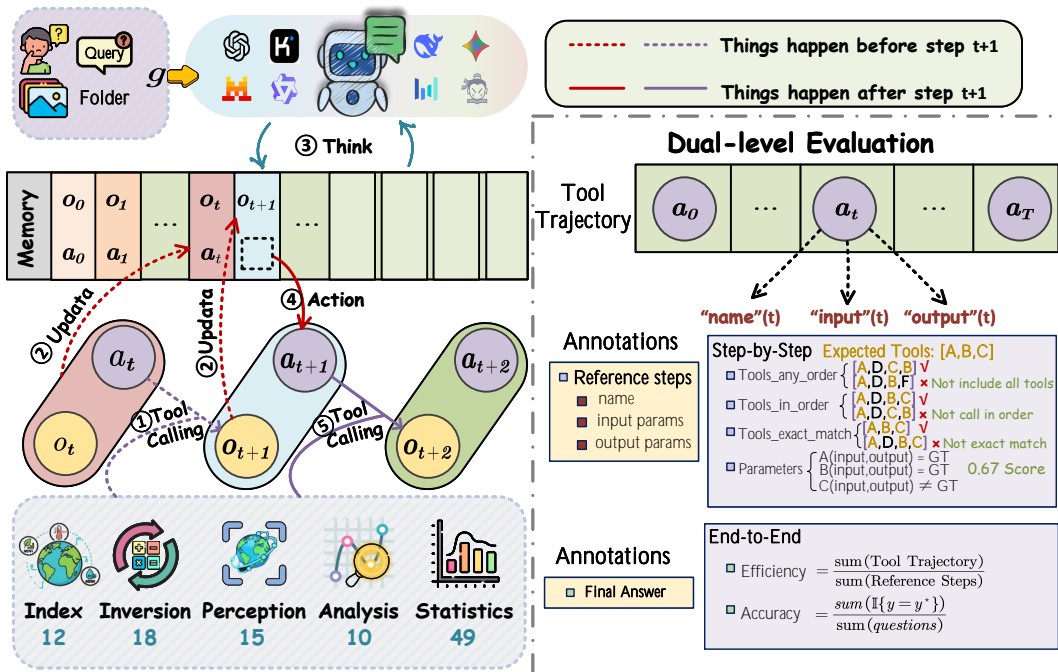

Figure 3: **Earth-Agent Framework:** The left part illustrates the ReAct-style workflow, where Earth-Agent iteratively performs tool calling, memory update, thinking, and action using domain-specific toolkits. The right panel presents the dual-level evaluation protocol, assessing both step-by-step reasoning trajectories and end-to-end outcomes.

The action distribution is modeled as:

$$a_t \sim \pi(a_t \mid g, m_t).$$

The full agent trajectory $\tau = [s_0, o_0, a_0, s_1, o_1, a_1, \ldots, s_T, o_T]$ is determined jointly by the policy $\pi$ and the environment dynamics:

$$p_\pi(\tau) = \underbrace{p(s_0)\, Z(o_0 \mid s_0)}_{\text{Initial state}} \prod_{t=0}^{T-1} \underbrace{\pi(a_t \mid g, m_t)}_{\text{③ Think \& ④ Action}} \overbrace{Z(o_{t+1} \mid s_{t+1})}^{\text{① Tool calling}} \mathcal{T}(s_{t+1} \mid s_t, a_t).$$

where $Z$ denotes the observation distribution induced by tool outputs.

In this formulation, the LLM controller functions as the policy $\pi$, reasoning over the history $m_t$ and task goal $g$ to decide the next tool calling, while the Toolkit provides executable atomic actions categorized into *Index, Inversion, Perception, Analysis,* and *Statistics*. Concretely, as illustrated in Figure 3, each loop proceeds as follows: ① **Tool calling** At step $t$, the agent invokes the most suitable tool conditioned on the current memory $m_t$ and task goal $g$, which yields the tool response for observation $o_{t+1}$. ② **Memory update** At step $t$, the agent appends the pair $(o_t, a_t)$ together with the resulting observation $o_{t+1}$ into the memory stack, ensuring that the complete interaction history is preserved for subsequent reasoning. ③ **Think** At step $t+1$, the LLM controller reasons over the updated memory $m_t$ together with the task goal $g$ to plan the next action, determining which tool to invoke and how to configure its inputs. ④ **Action** Selecting and executing the subsequent tool call $a_{t+1}$ that produces $o_{t+2}$. The ReAct loop continues until the stopping condition is satisfied, yielding both the final answer and a reproducible sequence of tool calling trajectory.

### 3.2  TOOL KIT

To enable comprehensive EO analysis, Earth-Agent integrates 104 specialized tools organized into five functional kits. The *Index kit* provides implementations of widely used EO indices (e.g., NDVI, NDWI, NBR) (Montero et al., 2023) for rapid environmental characterization. The *Inversion kit* focuses on geophysical parameter retrieval, including land surface temperature (LST) (Li et al., 2013),

precipitable water vapor (PWV) (He & Liu, 2020), vegetation water content (Ceccato et al., 2001), sea ice concentration (DiGirolamo et al., 2022), and others. The ***Perception kit*** supports vision-oriented tasks such as scene classification (Ma et al., 2025), object detection (Li et al., 2024e), and segmentation (Ravi et al., 2024). The ***Analysis kit*** targets spatiotemporal reasoning, offering trend detection, seasonality decomposition, change point analysis, and spatial autocorrelation. Finally, the ***Statistics kit*** provides large-scale data preprocessing and statistical computation (e.g., variance, skewness, batch operations, cloud masking). Together, these tool kits cover the diverse types of EO tasks from perceptual to spectral, and from descriptive to quantitative analysis. The detailed list and description of tools can be found in Appendix G.

### 3.3 EVALUATION PROTOCOL

Prior benchmarks have primarily emphasized final accuracy, overlooking the reasoning trajectory that leads to the final output (Mialon et al., 2023; Jimenez et al., 2024; Chen et al., 2025). To address this, we adopt a **dual-level evaluation protocol**: agents are assessed in a ***step-by-step*** mode to capture the quality of their reasoning trajectories, and in an ***end-to-end*** mode to measure final task performance. This dual perspective enables fine-grained diagnostics of both reasoning and outcomes. The detailed calculation formulas can be found in Appendix B.2.

**End-to-End** evaluation measures task-level performance, including *Accuracy* of the final answer and *Efficiency* of the trajectory relative to expert solutions.

**Step-by-Step** evaluation assesses the quality of intermediate reasoning. We consider four complementary aspects: ***Tool-Any-Order***, which checks whether all necessary tools are used in LLM planning; ***Tool-In-Order***, which evaluates whether tools are invoked in the correct sequence; ***Tool-Exact-Match***, which evaluates the exact prefix-level accuracy between the predicted and expert trajectories; and ***Parameter Accuracy***, which verifies whether both tool identifiers and their arguments are correctly matched.

## 4 EARTH-AGENT BENCHMARK

### 4.1 OVERVIEW OF EARTH-AGENT BENCHMARK

We introduce **Earth-Agent Benchmark (Earth-Bench)**, a dataset designed to evaluate tool-augmented EO agents in realistic Earth science analysis scenarios. The benchmark integrates three major types of Earth observation data: ***RGB Imagery (RGB)***, ***Raw Spectral Data (Spectrum)***, and ***Processed Earth Products (Products)***. It supports 14 representative tasks, including classification, detection, temperature monitoring, weather forecasting, etc., with a particular emphasis on scientific analysis that requires quantitative reasoning rather than qualitative description.

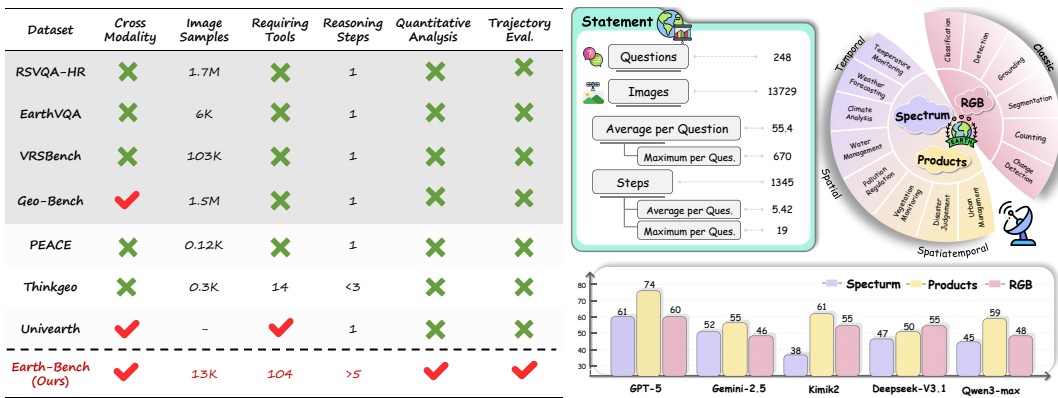

| Dataset | Cross Modality | Image Samples | Requiring Tools | Reasoning Steps | Quantitative Analysis | Trajectory Eval. |
|---|---|---|---|---|---|---|
| RSVQA-HR | ✗ | 1.7M | ✗ | 1 | ✗ | ✗ |
| EarthVQA | ✗ | 6K | ✗ | 1 | ✗ | ✗ |
| VRSBench | ✗ | 103K | ✗ | 1 | ✗ | ✗ |
| Geo-Bench | ✓ | 1.5M | ✗ | 1 | ✗ | ✗ |
| PEACE | ✗ | 0.12K | ✗ | 1 | ✗ | ✗ |
| Thinkgeo | ✗ | 0.3K | 14 | <3 | ✗ | ✗ |
| Univearth | ✓ | – | ✓ | 1 | ✗ | ✗ |
| Earth-Bench (Ours) | ✓ | 13K | 104 | >5 | ✓ | ✓ |

Figure 4: **Dataset Comparison and Overview:** The left panel compares Earth-Bench with prior MLLM and agentbased EO benchmarks. The right panel presents the statistics of Earth-Bench and its evaluation with SOTA LLMs using Earth-Agent, highlighting the difficulty of Earth-Bench.

As shown in Figure 4, **MLLM-based benchmarks** including RSVQA-HR (Lobry et al., 2020), EarthVQA (Wang et al., 2024b), VRSBench (Li et al., 2024d) and Geo-Bench (Lacoste et al., 2023) are mainly limited to single-step perceptual for RGB data using pretrained MLLMs (Liu

et al., 2024d; Team, 2024; OpenAI, 2024), without requiring external tool use for scientific quantitative analysis (e.g., spatiotemporal trend estimation), not to mention reasoning trajectory evaluation. On the other hand, **Earth-Bench** advances beyond prior **Agent-based EO benchmarks**, such as PEACE (Huang et al., 2025), Thinkgeo (Shabbir et al., 2025) and UnivEarth (Kao et al., 2025), by incorporating 13K+ images across spectrum, product and RGB modalities, while requiring interaction with 104 domain tools. It consists of 248 expert-curated questions, requiring an average of 5.4 reasoning steps of quantitative analysis. Even with the state-of-the-art (SOTA) LLM backbones, performance remains limited, which underscores not only the benchmark's difficulty and diagnostic value but also the necessity of reasoning trajectory evaluation. Therefore, we need to annotate on both trajectories and final answers in section 4.2.

## 4.2 DATA ANNOTATION PIPELINE

To construct Earth-Bench, we collected raw data from platforms such as Google Earth Engine, NASA EarthData, and public remote sensing datasets (Xia et al., 2017; Zhan et al., 2023; Xia et al., 2018; Su et al., 2019). From these data sources, a team of domain experts curated 248 problems that require multi-step quantitative reasoning. The annotation team was composed of **2** computer science experts, **7** remote sensing specialists, and **3** Earth science specialists. Each problem is accompanied by a step-by-step Python solution and is designed to reflect the complexity of real-world Earth science workflows, which demand the coordinated use of multiple tool kits.

In prior benchmarks, queries have been designed either as *step-implicit*, where no intermediate step guidance is provided (Mialon et al., 2023; Wang et al., 2024a), or as *step-explicit*, where the query itself contains step guidance (Guo et al., 2024b; Ma et al., 2024). Motivated by the complexity of EO workflows, which often require multi-step processing, Earth-Bench incorporates both regimes: **Auto-Planning** corresponds to the step-implicit setting and evaluates the agent's ability to autonomously plan its solution trajectory, while **Instruction-Following** corresponds to the step-explicit setting and evaluates the agent's ability to follow and translate human instructions into executable actions. Both regimes contain 248 complete questions for evaluation. Together, these two regimes provide a comprehensive assessment of both autonomous reasoning and guided execution in EO contexts. Details can refer to Appendix A.4.

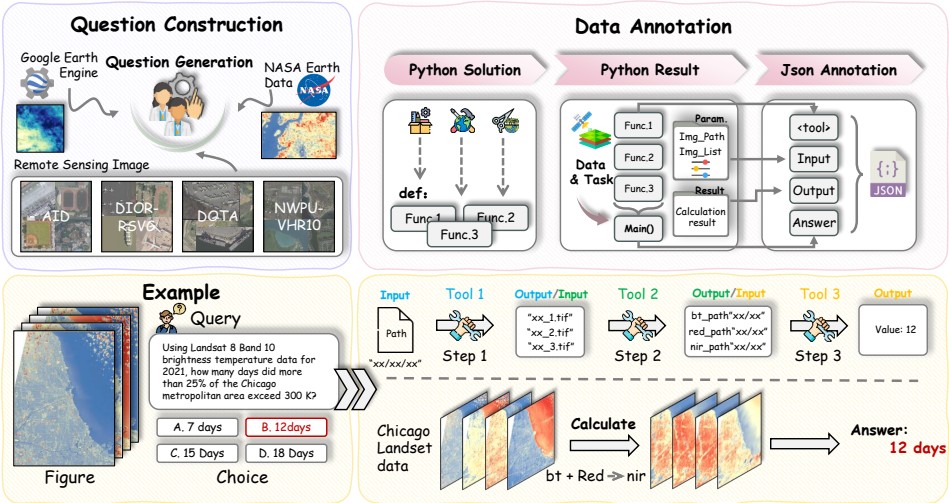

Figure 5: **Construction and Annotation of Earth-Bench.** The left shows question generation from EO data, the right illustrates the data annotation pipeline that simulates ReAct-style trajectories, and the bottom provides an example explaining the multi-step annotation process.

To enable dual-level evaluation, we explicitly annotated both the final answers and the full reasoning trajectories. As illustrated in Fig. 5, the annotation process was designed to mimic the ReAct loop of agents: **Python Solution.** Annotators first identify the domain tools such as compute_ndwi and split_window) required to solve a problem and then assemble them into a step-by-step main() program. Each tool is represented as a Python function, and the functions are planned in the correct order to form an executable workflow that mirrors the agent's reasoning trajectory. **Python Result.**

When executed, the program produces explicit input and output arguments for each function call, as well as the final output of the main() function. **JSON Annotation.** Each function call is then translated into a structured JSON record to align with the ReAct-style trajectory annotation. The function name corresponds to the action tool name, the function input arguments corresponds to the action passed by the agent, and the function output arguments corresponds to the tool responses returned to the agent. The final output of the main() function is recorded as the ground-truth answer for the problem. This provides a complete record of both the reasoning trajectory and the final answer.

## 5 EXPERIMENTS

### 5.1 EXPERIMENTAL SETUP

**Evaluated Models.** We evaluate 3 closed-source and 10 leading open-source LLMs. For *closed-source models*, we consider GPT-5 (OpenAI, 2025a), GPT-4o (OpenAI, 2024), and Gemini-2.5 (Comanici et al., 2025). For *open-source models*, including Deepseek-V3.1 (Liu et al., 2024a), Kimik2 (Team et al., 2025), Qwen3-max-Preview, Qwen3-32B (Yang et al., 2025), and InternVL3.5 (Wang et al., 2025b), which represent the smartest open LLMs available to date.

### 5.2 EARTH-AGENT WITH DIFFERENT LLM BACKBONES

As shown in Table 1, we evaluate the impact of different LLM backbones on Earth-Bench. Results are reported under both *step-by-step* and *end-to-end* evaluation protocols, allowing us to jointly assess the quality of reasoning trajectories and final outcomes. The following observations can be made:

Table 1: Performance of different LLM backbones on Earth-Bench under both *Auto-Planning (AP)* and *Instruction-Following (IF)* regimes. Results are reported with dual-level evaluation, covering Accuracy, Efficiency for final outcomes and Tool-Any-Order, Tool-In-Order, Tool-Exact-Match, Parameters for trajectory quality. We **bold** the best results and underline the runner-ups.

| Model | Accuracy | | Efficiency | | Tool-Any-Order | | Tool-In-Order | | Tool-Exact-Match | | Parameters | |
|---|---|---|---|---|---|---|---|---|---|---|---|---|
| | AP | IF | AP | IF | AP | IF | AP | IF | AP | IF | AP | IF |
| GPT-5 | **65.99** | **62.35**↓ | 2.3560 | 2.9093 | 68.74 | 71.41↑ | 57.71 | 61.06↑ | 44.97 | 46.01↑ | 26.11 | 25.91↓ |
| Gemini-2.5 | 52.23 | 53.04↓ | 2.9958 | 2.4595 | 58.04 | 61.63↑ | 45.31 | 50.72↑ | 31.32 | 41.04↑ | 17.26 | 23.43↑ |
| GPT-4o | 43.72 | 44.94↓ | 2.1211 | 2.6007 | 65.65 | 67.02↑ | 50.76 | 53.04↑ | 46.26 | 47.41↑ | 26.55 | 27.92↑ |
| Kimik2 | 50.61 | 56.68↓ | 1.8542 | 2.1793 | 71.03 | 78.86↑ | 57.57 | 68.83↑ | 42.11 | **51.15**↑ | 25.90 | 30.45↑ |
| DeepSeek-V3.1 | 51.42 | 52.23↑ | 2.6116 | 2.6303 | **78.31** | 78.66↓ | **62.73** | **64.50**↑ | **48.54** | 49.58↑ | **30.81** | **31.36**↑ |
| Qwen3-Max | 50.20 | 47.37↓ | 1.8810 | 1.9511↓ | 69.56 | 70.14↑ | 53.28 | 56.02↑ | 37.02 | 42.74↑ | 21.83 | 26.27↑ |
| Seed-1.6 | 52.48 | 59.51↓ | 1.3110 | 1.3408↓ | 55.43 | 59.44↑ | 40.67 | 46.79↑ | 28.39 | 35.47↑ | 18.32 | 23.13↑ |
| LLaMA-4 | 44.94 | 38.46↓ | 0.2886 | 0.3211↓ | 16.51 | 22.41↑ | 2.45 | 12.05↑ | 1.70 | 9.05↑ | 1.30 | 6.46↑ |
| Qwen-Plus | 42.51 | 38.46↓ | 1.5119 | 1.5854↓ | 52.04 | 55.96↑ | 30.75 | 39.77↑ | 11.69 | 25.51↑ | 9.12 | 16.95↑ |
| GLM-4.5v | 32.86 | 35.25↑ | 1.7123 | 2.0129↑ | 42.48 | 46.69↑ | 28.57 | 35.24↑ | 14.12 | 19.95↑ | 11.02 | 15.37↑ |
| Mistral | 29.96 | 22.67↓ | 0.9552 | 0.8802↑ | 27.73 | 29.64↑ | 11.78 | 20.90↑ | 9.13 | 18.13↑ | 7.24 | 11.66↑ |
| Qwen3-32B | 20.65 | 24.80↑ | 2.7274 | 1.9010↑ | 39.76 | 42.39↑ | 21.56 | 33.79↑ | 9.51 | 26.10↑ | 8.17 | 17.73↑ |
| InternVL-3.5 | 26.72 | 26.72 | 0.1206 | 0.1750↓ | 8.83 | 16.62↑ | 3.87 | 10.59↑ | 2.02 | 9.32↑ | 1.46 | 5.32↑ |

**Obs.1.** LLM models pretrained with tool calling, such as GPT-5, Gemini-2.5, DeepSeek-V3.1, Kimik2, and Qwen3, demonstrate **strong** performance across both *step-by-step* and *end-to-end* evaluations. Further, closed-source models like GPT-5 typically achieve **higher** *final accuracy*, while open-source models, particularly DeepSeek-V3.1 and Kimik2, **outperform** GPT-5 in *tool-use accuracy*, demonstrating superior performance in reasoning trajectory alignment. We have provided a detailed analysis of the LLMs' performance across the Spectrum, Products, and RGB modalities, which can be found in Appendix C.

**Obs.2.** Instruction-following regimes enhance tool calling by providing explicit step guidance in the query, leading to **improved** *tool calling accuracy* across nearly all models. Interestingly, despite the improved reasoning trajectories, this does **not necessarily** lead to *higher final accuracy*. In fact, for some advanced models, this added complexity may even result in a **decrease** in *final accuracy*. We have conducted a detailed *error analysis* of Earth-Agent's performance in the Earth-Bench benchmark, focusing on representative models such as GPT-5, DeepSeek-V3.1, Kimik2, and Qwen3-max. This can be found in Appendix D.

**Obs.3.** Across nearly all models, the ability to identify the correct set of tools, as reflected in *Tool-Any-Order* and *Tool-In-Order* metrics, remains **consistently strong**. However, models often introduce irrelevant steps during reasoning, which **undermines** their accuracy on *Tool-Exact-Match* and *parameter* execution. Crucially, these two fine-grained capabilities are indispensable in real EO analysis workflows. For example, if additional transformations are mistakenly applied to the EO data process, it becomes extremely difficult to obtain correct results in the subsequent steps. Their weakness therefore exposes a **key bottleneck** that prevents EO Agents from achieving higher final *accuracy* in EO data processing.

## 5.3 COMPARISON WITH GENERAL AGENTS

Since many Earth-Bench tasks involve processing hundreds of images, existing open-source agent frameworks cannot handle these questions due to input size constraints. To enable fair comparison, we construct **Earth-Bench-Lite**, a reduced yet representative subset that preserves modality diversity while remaining within the capacity of general-purpose agents. It consists of 60 questions evenly distributed across the three EO modalities: Spectrum, Products, and RGB. As shown in Table 2, we report results across three modalities: *Spectrum*, *Products*, and *RGB*.

Table 2: Comparison with general agents on Earth-Bench-Lite. Results are reported across Spectrum, Products, RGB modalities. We **bold** the best results and underline the runner-ups.

| Method | Spectrum | Products | RGB | Avg. | Latency |
|---|---|---|---|---|---|
| GPT-Agent | 45.00 | 31.60 | 45.26 | 40.42 | ≈ 300 min |
| MGX | 40.00 | 15.80 | 0.00 | 18.60 | **≈ 60 min** |
| Manus | 15.00 | 15.80 | 47.62 | 26.14 | ≈ 150 min |
| Coze | 35.00 | 10.50 | 0.00 | 15.30 | ≈ 120 min |
| Earth-Agent(GPT5) | **65.00** | **36.84** | **65.71** | **55.83** | 158 min |
| Earth-Agent(Deepseek-V3.1) | 50.00 | 42.11 | 51.43 | 47.84 | 79 min |
| Earth-Agent(Kimik2) | 36.84 | 50.00 | 50.00 | 45.95 | 131 min |

By comparison, general agents show limited modality coverage. They can handle relatively simple *Spectrum* tasks by writing ad-hoc code, but perform poorly on *Products* tasks due to the lack of domain-specific spatiotemporal analysis tools. For the *RGB* modality, MGX and Coze even fail to complete any tasks. In contrast, by interacting with 104 predefined geoscience tools, Earth-Agent consistently achieves superior performance across all three modalities, whether driven by the closed-source GPT-5 or the open-source DeepSeek-V3.1. We also compared the latency of our Earth-Agent framework with that of the baseline agents. The latency remains within a reasonable range when compared to existing general agents. The substantial performance improvements in task completion more than justify the additional computational cost. A detailed discussion can be found in the Appendix E.

## 5.4 COMPARISON WITH MLLM-BASED EO METHODS

We further compare Earth-Agent with remote sensing large models on classification, detection, and segmentation tasks. The results are summarized in Table 3.

Table 3: Comparison with MLLMs on Remote sense benchmarks. Results are reported on classification, detection, and segmentation tasks. We **bold** the best results and underline the runner-ups.

| Model | Classification | | Detection | | Grounding |
|---|---|---|---|---|---|
| | AID | WHU-RS19 | DOTA | HRSC2016 | DIOR-RSVG |
| MiniGPT-v2 (Chen et al., 2023) | 32.96 | 64.80 | 14.8 | 24.8 | 29.892 |
| LLaVA-1.5 (Liu et al., 2024d) | 51.00 | 74.52 | 17.5 | 22.1 | 12.085 |
| Sphinx (Lin et al., 2023) | 58.20 | - | 15.1 | 25.7 | 0.939 |
| Geochat (Kuckreja et al., 2024) | 72.03 | 86.47 | 16.5 | 24.0 | 10.024 |
| VHM (Pang et al., 2025) | 91.70 | 95.80 | - | - | - |
| LHRS-Bot (Muhtar et al., 2024) | 91.26 | 93.17 | 17.1 | 24.4 | 11.826 |
| Earth-Agent (ours) | **93.42** | **96.12** | **60.88** | **65.60** | **60.46** |

Earth-Agent demonstrates clear superiority over existing MLLMs across classification, detection, and segmentation benchmarks (Table 3). Prior MLLM-based approaches often lack generalization

across diverse EO tasks: for example, LHRS-Bot achieves strong results in classification but struggles on detection and grounding, while VHM attains high classification accuracy but cannot even handle detection or segmentation tasks. In contrast, Earth-Agent interacts with a predefined toolkit of 104 geoscience functions and expert models, allowing it to adaptively invoke specialized tools or models for each task type. This modular design enables Earth-Agent to maintain robust performance across modalities, overcoming the limited extensibility of previous MLLM-based EO systems.

## 6    CONCLUSION

Earth-Agent marks a significant advancement in applying (M)LLMs to EO analysis, extending RGB perception to Spectrum, Products and RGB modalities. By shifting from single-step inference with pretrained MLLMs to multi-step reasoning through external tool/model integration, it overcomes key limitations of prior MLLM-based approaches, such as handling numerous image inputs and quantitative spatiotemporal analysis. To support rigorous evaluation, we introduced Earth-Bench, which requires multi-step quantitative reasoning and dual-level evaluation, which evaluate both reasoning trajectories and final outcomes. Our experiments further reveal the current bottlenecks of Earth-Agent in EO applications and provide detailed diagnostics. Finally, by comparing with both general agents and domain MLLMs, we highlight the transformative potential of Earth-Agent as a foundation for the revolutio of LLM applications in Earth observation.

## ETHICS STATEMENT

This work adheres to the ICLR Code of Ethics. In this study, no human subjects or animal experimentation was involved. All datasets used, including GEE, NASA EarthData, AID, DOTA, HRSC2016, DIOR-RSVG, were sourced in compliance with relevant usage guidelines, ensuring no violation of privacy. We have taken care to avoid any biases or discriminatory outcomes in our research process. No personally identifiable information was used, and no experiments were conducted that could raise privacy or security concerns. We are committed to maintaining transparency and integrity throughout the research process.

## REPRODICIBILITY STATEMENT

We are committed to ensuring the full reproducibility of our work. The proposed Agent framework is described in Section 3, while the evaluated models and metrics are detailed in Appendix B. To further support transparency, we will release our codebase, which includes the Agent framework, evaluation scripts, and other relevant components, and we will also provide the evaluation logs from the agent experiments, enabling the community to replicate our results and build upon them.

## ACKNOWLEDGEMENTS

This project was funded by National Natural Science Foundation of China (Grant No. 62571560) and Shanghai AI Laboratory.

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

# A   Dataset Illustration

## A.1   Dataset Composition

The remote sensing imagery used in our dataset primarily comes from Landsat and MODIS, with additional high-resolution imagery sourced from open datasets. All the remote sensing data products are obtained through Google Earth Engine (GEE). The following Table 4 is a more detailed breakdown of the data sources, products, and their distribution:

Table 4: Sensor and Data Source Statistics

| Category | Sensor/Data Source | Dataset | Samples |
|---|---|---|---|
| **Spectrum** | Landsat | Landsat 8 / 9 | 1684 |
| | MODIS | - | 10273 |
| | ASTER | - | 110 |
| **Products** | GEE | LST | 183 |
| | | NDVI | 369 |
| | | GPM | 160 |
| | | VIIRS | 164 |
| | | GHSL | 68 |
| | | QA PIXEL | 212 |
| | | NDWI | 69 |
| | | fire MaxFRP | 194 |
| **RGB** | Public Datasets | AID | 169 |
| | | DIOR-RSVG | 7 |
| | | DOTA | 22 |
| | | NWPU-VHR-10 | 21 |
| | | xBD | 32 |
| **Total** | - | - | 13737 |

This table provides an overview of the data sources and the distribution of samples across three categories (Spectrum, Product, RGB). It includes a diverse set of remote sensing products, such as Landsat, MODIS, ASTER, and various other publicly available high-resolution datasets, ensuring comprehensive coverage for a wide range of Earth observation tasks.

## A.2   Benchmark Statistics

Table 5: Statistics of the benchmark dataset, including average verification code length, number of images, query length, reasoning steps, and question counts for different task types.

| Type | Avg. Code Lines | Avg. Images | Avg. Query Length AP | IF | Avg. Steps | Question Count |
|---|---|---|---|---|---|---|
| **Spectrum** | 361.19 | 96.82 | 331.54 | 497.94 | 4.38 | 100 |
| **Products** | 283.64 | 43.23 | 505.72 | 648.09 | 6.35 | 88 |
| **RGB** | 176.42 | 4.18 | 333.80 | 464.70 | 5.77 | 60 |
| **Avg.** | 288.97 | 55.39 | 393.89 | 543.18 | 5.42 | - |
| **Total** | 71664 | 13737 | 97685 | 134708 | 1343 | 248 |

As illustrated in Table 5, Earth-Bench consists of 248 questions associated with approximately 13.7K images. We recruited a team of domain experts to annotate these questions. The annotation

process was as follows: experts first designed solution steps based on their expertise, then selected appropriate tools from the Agent Toolkit, implemented the steps by writing Python code to invoke the selected tools for data processing, and finally generated the corresponding answers. In total, the expert team annotated 1345 solution steps and produced 71,664 lines of verification code for the 248 benchmark questions.

## A.3 QUESTION TYPES AND CATEGORY

As shown in Figure 5, Earth-Bench questions are categorized into three types based on their data sources: *Spectrum*, *Products*, and *RGB*. The Table 6 below summarizes the number and proportion of questions within each category:

Table 6: Distribution of Question Types and Their Proportions

| Question Types | Number | Proportion (%) |
|---|---|---|
| Temperature Monitoring | 44 | 17.74 |
| Weather Forecasting | 11 | 4.44 |
| Climate Analysis | 20 | 8.06 |
| Water Management | 22 | 8.87 |
| Pollution Regulation | 14 | 5.65 |
| Vegetation Monitoring | 28 | 11.29 |
| Disaster Judgement | 24 | 9.68 |
| Urban Management | 25 | 10.08 |
| Classification | 15 | 6.05 |
| Detection | 15 | 6.05 |
| Grounding | 7 | 2.82 |
| Segmentation | 3 | 1.21 |
| Counting | 7 | 2.82 |
| Change Detection | 13 | 5.24 |

This table provides a detailed distribution of question types within Earth-Bench, reflecting the variety of Earth observation tasks addressed by the dataset.

## A.4 QUERY REGIMES

Earth-Bench categorizes each task into two regimes: Auto-Planning and Instruction-Following. The key distinction is that in Instruction-Following, the query explicitly provides the Agent with a solution approach, while preserving the original intent of the task. As shown in Table 5, the statistics of query length highlight this difference: on average, queries in the Instruction-Following regime are about 150 characters longer than those in Auto-Planning, due to the inclusion of solution guidance. For illustration, Appendix H presents examples of the same task under both regimes, along with the performance of different LLMs. In summary, Instruction-Following emphasizes LLMs' instruction-following and tool-use capabilities, whereas Auto-Planning additionally evaluates their ability to decompose and plan Earth observation tasks.

## A.5 DATASET QUALITY CONTROL

Our annotation team was composed of **2** computer science experts, **7** remote sensing specialists (including one professor) and **3** Earth science specialists (including one professor).

Three core authors who served as senior reviewers led the pipeline of dataset construction. Each senior reviewer was responsible for guiding the development of the question sets and templates for

the ***Spectrum***, ***Products***, and ***RGB*** categories. They played a key role in providing strategic direction for the overall question pool.

The remaining 7 team members, consisting of graduate students and senior undergraduates, contributed in the following areas:

- Creation of initial questions (approximately 400 questions)
- Raw data collection
- Development of Python-based solution scripts

Once the questions were created, they were thoroughly reviewed by the 3 senior reviewers. The review process focused on two key criteria:

1. **Data Integrity**: Ensuring that the raw data involved (e.g., Landsat or MODIS) has complete band information within the task's time range and does not contain anomalies. Any questions and TIF files with missing or anomalous large values were discarded.

2. **Task Difficulty Assessment**: Senior reviewers assessed the difficulty of questions based on the number and complexity of the functions defined in the Python solutions. For simpler tasks (e.g., only calculating NDVI index to finish a task), these were removed to ensure an appropriate challenge across questions.

This collaborative structure ensured that the dataset was curated by a diverse team with expertise from the full landscape of Earth observation fields, enabling a well-rounded and comprehensive dataset for evaluation.

## A.6 BIAS ABLATION EXPERIMENT

To examine whether Earth-Bench exhibits bias toward specific models, i.e., whether certain models inherently find its tasks easier and thus risk skewing conclusions, we conducted an ablation study. Specifically, we removed all tools and allowed LLMs to directly answer questions given only the Query and Folder, then compared the results with those of tool-augmented Agents that had access to both Query and Data.

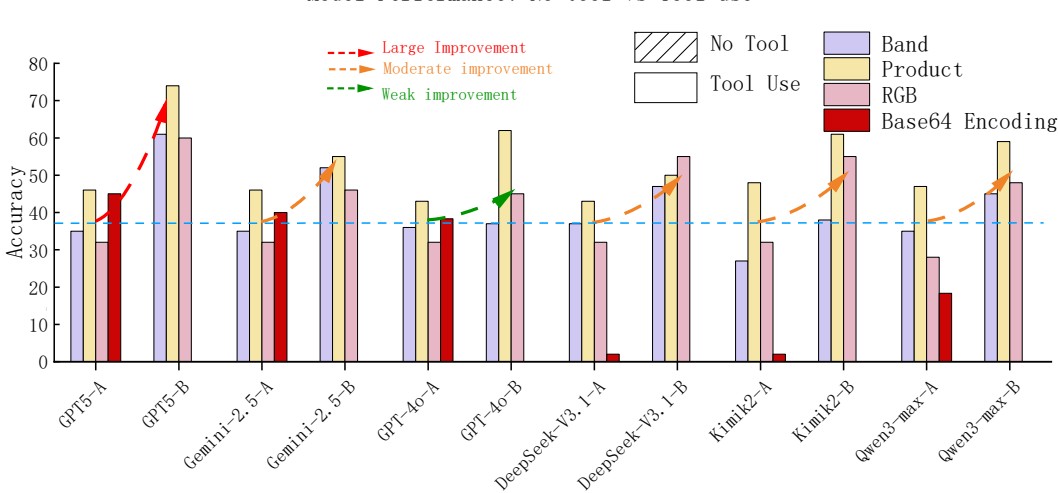

Figure 6: Dataset Ablation Experiment

As shown in Fig. 6, without tool access, mainstream LLMs achieved comparable performance across the three task types, with an overall Accuracy of about 37%. This indicates that the dataset is fair in its raw form and that models cannot rely solely on parametric knowledge to solve all benchmark questions. In contrast, with tool access, different models exhibited varying degrees of improvement:

GPT-5 achieved the largest gain, whereas GPT-4o showed a more modest increase. These results highlight differences in problem decomposition and tool-use capabilities among LLMs, and further corroborate the conclusions presented in the main experiments.

In the *No Tool* setting, LLMs/MLLMs were restricted from using any tools, and base64 encoding was not applied in RGB tasks specifically. To ensure integrity in the comparison, we added experiments with base64 encoding for RGB tasks. We observed that while models like GPT-5 showed some improvement in accuracy (compared with not applying base64 encoding), the performance remained at relatively low levels. DeepSeek and Kimik2 were unable to handle RGB tasks yet. The conclusions derived from the ablation study remain unchanged.

## B  EVALUATION

### B.1  EVALUATION MODEL

Our evaluation covers 13 recent LLMs, including both closed-source and open-source ones, to understand their capabilities across multiple Earth observation tasks. The baseline models are listed in Table 7.

Table 7: Models evaluated in our benchmark and their corresponding API references.

| Model | Model Version | API Links |
|---|---|---|
| GPT5 | GPT-5 | `https://platform.openai.com/docs/models/gpt-5` |
| GPT4 | GPT-4o | `https://platform.openai.com/docs/models/gpt-4o` |
| Gemini | Gemini-2.5-Flash | `https://ai.google.dev/gemini-api/docs/models#gemini-2.5-flash` |
| Mistral | Mistral-Large | `https://docs.mistral.ai/getting-started/models/` |
| Qwen | Qwen3-Max-Previous Qwen3-32B Qwen-Plus | `https://www.alibabacloud.com/help/en/model-studio/use-qwen-by-calling-api` |
| Kimi | Kimi-K2 | `https://platform.moonshot.ai/docs/guide/start-using-kimi-api` |
| Deepseek | Deepseek-V3.1 | `https://api-docs.deepseek.com` |
| Seed | Seed-1.6 | `https://www.volcengine.com/docs/82379/1099455` |
| LLaMA | LLaMA-4-Maverick | `https://www.llama.com/products/llama-api/` |
| GLM | GLM-4.5v | `https://docs.z.ai/guides/vlm/glm-4.5v` |
| InternVL | InternVL-3.5 | `https://internlm.intern-ai.org.cn/api/document` |

### B.2  EVALUATION METRIC

Formally, for each task goal $g$, our geoscience experts provide (i) a ground-truth final answer $y^\star$, and (ii) an expert-annotated reasoning trajectory.

$$\tau^\star = \big[(o_0^\star, a_0^\star),\, (o_1^\star, a_1^\star),\, \ldots,\, (o_m^\star, a_m^\star)\big],$$

where each action is defined as

$$a_k^\star = (t_k^\star,\, in_k^\star,\, out_k^\star),$$

with $t_k^\star \in \mathcal{V}$ denoting the tool identifier (from the tool vocabulary), $in_k^\star \in \mathcal{X}$ the input arguments, and $out_k^\star \in \mathcal{O}$ the corresponding output. In other words, each tool is characterized by its name in the vocabulary, its input arguments, and its output results.

Given a policy $\pi$, the agent generates an output trajectory

$$\tau = \big[(o_0, a_0),\, (o_1, a_1),\, \ldots,\, (o_n, a_n)\big],$$

together with a predicted final answer $y$.

To comprehensively evaluate the performance of the Agent on the Earth Benchmark, we assess its execution process from two perspectives: End-to-End and Step-by-Step. The corresponding evaluation metrics are defined as follows:

**End-to-End protocol.** End-to-end metrics evaluate the task-level performance of the agent, independent of its intermediate reasoning. We consider two complementary measures:

*(1) Accuracy.* The correctness of the final answer is computed as

$$\text{Acc} \;=\; \mathbb{E}_{g\sim\mathcal{G}}\big[\mathbb{I}\{y = y^\star\}\big], \tag{1}$$

where $\mathbb{I}\{\cdot\}$ is the indicator function and $\mathcal{G}$ is the distribution of benchmark tasks.

*(2) Efficiency.* To penalize unnecessarily long trajectories, we measure the relative optimality of tool usage:

$$\text{Eff}(\tau, \tau^\star) \;=\; \frac{|\tau|}{|\tau^\star|}, \tag{2}$$

where $|\tau|$ and $|\tau^\star|$ denote the number of tool calls in the predicted and ground-truth trajectories, respectively.

**Step-by-Step protocol.** In addition to outcome-based metrics, we also evaluate the fidelity of the reasoning trajectory relative to expert annotations. Let $\mathbf{t}^\star = (t_1^\star, \ldots, t_m^\star)$ and $\mathbf{t} = (t_1, \ldots, t_n)$ denote the tool sequences, and $\mathbf{in}^\star = (in_1^\star, \ldots, in_m^\star)$, $\mathbf{in} = (in_1, \ldots, in_n)$ the corresponding parameter sequences. We define four metrics:

*(1) Tools-Any-Order (TAO).* Coverage of required tools, ignoring order and duplicates:

$$TAO(\tau, \tau^\star) \;=\; \frac{\big|\, Set(\mathbf{t}^\star) \cap Set(\mathbf{t})\,\big|}{\big|\, Set(\mathbf{t}^\star)\,\big|}, \tag{3}$$

where $Set(\cdot)$ extracts the set of unique tools.

*(2) Tools-In-Order(TIO).* Fraction of ground-truth tools matched as a subsequence in the predicted sequence:

$$k^\star \;=\; \max\big\{\, k : \exists\, 1 \le j_1 < \cdots < j_k \le n,\; t_{j_i} = t_i^\star,\; \forall i \le k \big\}, \quad TIO(\tau, \tau^\star) \;=\; \frac{k^\star}{m}. \tag{4}$$

*(3) Tool-Exact-Match(TEM).* Length of the longest common prefix (LCP), normalized by the ground-truth length:

$$\ell_{\text{lcp}} \;=\; \max\big\{\, \ell \le \min(m, n) : t_i = t_i^\star,\; \forall i \le \ell \big\}, \quad TEM(\tau, \tau^\star) \;=\; \frac{\ell_{\text{lcp}}}{m}. \tag{5}$$

*(4) Parameter Accuracy.* Exact match of both tool identifiers and arguments under the prefix rule:

$$\ell_{\text{param}} \;=\; \max\big\{\, \ell \le \min(m, n) : t_i = t_i^\star \wedge in_i \equiv in_i^\star,\; \forall i \le \ell \big\}, \quad S_{\text{param}} \;=\; \frac{\ell_{\text{param}}}{m}. \tag{6}$$

Here $in_i \equiv in_i^\star$ denotes structural equality of arguments (e.g., dictionary match).

## C    Breakdown Results on Different Modalities

Table 8, Table 9, and Table 10 present the detailed evaluation results on different subsets of Earth-Bench. In the main analysis, we report only the overall performance across the entire benchmark to ensure clarity and comparability. Nevertheless, the breakdown results of individual subsets provide valuable insights into potential directions for improving LLMs in Earth Observation tasks.

Table 8: Performance of different LLM-based agents on the **Spectrum** subset of Earth-Bench. We **bold** the best results and underline the runner-ups.

| Model | Accuracy | | Efficiency | | Tool-Any-Order | | Tool-In-Order | | Tool-Exact-Match | | Parameters | |
|---|---|---|---|---|---|---|---|---|---|---|---|---|
| | AP | IF | AP | IF | AP | IF | AP | IF | AP | IF | AP | IF |
| GPT-5 | **61.00** | **64.00**↑ | 3.5510 | 4.6657↓ | 72.67 | 78.37↑ | 65.80 | 74.71↑ | 49.64 | 52.07↑ | 21.74 | 24.97↑ |
| Gemini-2.5 | 52.00 | 52.00 | 4.3584 | 4.5585↑ | 71.35 | 71.90↑ | 65.14 | 64.84↓ | 40.12 | 49.89↑ | 17.57 | 21.41↑ |
| GPT-4o | 37.00 | 42.00↑ | 3.7736 | 4.5726↓ | 69.55 | 72.83↑ | 56.12 | 62.07↑ | 48.27 | 51.02↑ | 21.62 | 24.08↑ |
| Kimik2 | 38.00 | 50.00↑ | 2.5758 | 3.1005↓ | 71.92 | **86.02**↑ | 62.27 | **78.91**↑ | 43.04 | **54.81**↑ | 20.73 | 25.67↑ |
| DeepSeek-V3.1 | 47.00 | 45.00↓ | 3.9014 | 4.0685↓ | 76.48 | 75.97↓ | **67.64** | 66.57↓ | **50.22** | 50.52↑ | **26.32** | **26.13**↓ |
| Qwen3-Max | 45.00 | 40.00↓ | 3.1981 | 3.2864↓ | **77.67** | 74.27↓ | 66.47 | 65.58↓ | 33.97 | 48.54↑ | 16.04 | 24.43↑ |
| Seed-1.6 | 40.00 | 57.00↑ | 1.7525 | 1.9186↓ | 55.07 | 63.92↑ | 42.00 | 54.94↑ | 24.53 | 34.37↑ | 12.56 | 16.83↑ |
| LLaMA-4 | 36.00 | 37.00↑ | 0.3648 | 0.4275↓ | 16.89 | 25.20↑ | 3.57 | 18.26↑ | 2.69 | 13.58↑ | 2.02 | 8.17↑ |
| Qwen-Plus | 33.00 | 36.00↑ | 2.2833 | 2.4157↓ | 55.95 | 57.38↑ | 36.27 | 48.89↑ | 5.67 | 35.14↑ | 2.87 | 17.47↑ |
| GLM-4.5v | 33.33 | 28.28↓ | 3.1121 | 3.0709↓ | 47.53 | 49.87↑ | 41.93 | 45.63↑ | 14.26 | 25.22↑ | 9.13 | 16.41↑ |
| Mistral | 24.00 | 18.00↓ | 1.3825 | 0.8316↓ | 23.73 | 19.58↓ | 4.58 | 16.13↑ | 1.87 | 13.37↑ | 1.33 | 6.15↑ |
| Qwen3-32B | 12.00 | 29.00↑ | 4.3328 | 3.4380↓ | 45.02 | 65.25↑ | 26.60 | 57.17↑ | 5.53 | 38.86↑ | 3.43 | 20.52↑ |
| InternVL-3.5 | 19.00 | 25.00↑ | **0.1127** | **0.2411**↓ | 7.50 | 18.77↑ | 3.58 | 16.09↑ | 0.58 | 13.02↑ | 0.33 | 5.65↑ |

On the Spectrum subset, the accuracy of the Agent's responses is generally below the average, and the overall efficiency is also lower than the benchmark mean. This indicates that the Agent encounters significant difficulties when addressing tasks in this subset. A likely reason is that the LLMs involved in the evaluation have limited familiarity with processing raw Earth Observation data. Furthermore, tasks in this subset often require analyzing a larger number of images, making them inherently more challenging compared to tasks in other subsets.

In contrast, on the Product subset, the Agent's responses are substantially above the average in terms of accuracy, and its efficiency is comparable to that of expert annotations. This suggests that LLMs are more adept at handling structured, product-level information, where tasks often align with general reasoning and statistical capabilities rather than requiring specialized domain expertise.

Table 9: Performance of different LLM-based agents on the **Products** subset of Earth-Bench. We **bold** the best results and underline the runner-ups.

| Model | Accuracy | | Efficiency | | Tool-Any-Order | | Tool-In-Order | | Tool-Exact-Match | | Parameters | |
|---|---|---|---|---|---|---|---|---|---|---|---|---|
| | AP | IF | AP | IF | AP | IF | AP | IF | AP | IF | AP | IF |
| GPT-5 | **75.00** | **71.59**↓ | 1.7154 | 1.6190↑ | 60.04 | 62.35↑ | 38.43 | 40.44↑ | 31.52 | 34.03↑ | 17.62 | 16.75↓ |
| Gemini-2.5 | 62.50 | 63.64↑ | 3.0055 | 1.1600↓ | 48.94 | 51.46↑ | 33.82 | 35.26↑ | 29.54 | 27.63↓ | 16.33 | 17.11↑ |
| GPT-4o | 54.55 | 54.55 | 1.1800 | 1.5270↓ | 57.27 | 60.11↑ | 33.89 | 37.80↑ | 31.31 | 35.02↑ | 19.13 | 18.71↓ |
| Kimik2 | 62.50 | 60.23↓ | 1.2489 | 1.6481↓ | 66.91 | 69.83↑ | 42.99 | **49.96**↑ | **34.75** | **38.84**↑ | 20.23 | **21.32**↑ |
| DeepSeek-V3.1 | 50.00 | 59.09↑ | 1.8111 | 1.6449↓ | **73.48** | **72.75**↑ | **43.35** | 46.16↑ | 32.18 | 33.32↑ | **20.50** | 19.92↓ |
| Qwen3-Max | 56.82 | 61.36↑ | 1.0688 | 1.0859↑ | 63.24 | 66.80↑ | 40.30 | 44.78↑ | 33.22 | 30.24↓ | 20.23 | 16.52↓ |
| Seed-1.6 | 65.06 | 67.05↑ | 0.8776 | 0.9359↓ | 54.02 | 55.57↑ | 36.75 | 37.20↑ | 28.84 | 31.27↑ | 16.79 | 19.82↑ |
| LLaMA-4 | 60.23 | 47.73↓ | 0.1641 | 0.1614↓ | 8.92 | 9.83↑ | 2.51 | 2.61↑ | 1.39 | 1.65↑ | 1.05 | 1.31↑ |
| Qwen-Plus | 53.41 | 40.91↓ | 0.9972 | 1.0016↑ | 47.71 | 51.57↑ | 25.33 | 27.37↑ | 10.16 | 4.14↓ | 6.52 | 2.97↓ |
| GLM-4.5v | 43.18 | 47.67↑ | 0.9342 | 1.1979↓ | 35.06 | 41.97↑ | 19.36 | 24.68↑ | 8.77 | 9.04↑ | 6.60 | 7.77↑ |
| Mistral | 36.36 | 22.73↓ | 0.6263 | 1.0206↓ | 26.61 | 30.36↑ | 12.80 | 13.43↑ | 8.83 | 10.40↑ | 5.42 | 4.40↓ |
| Qwen3-32B | 27.27 | 18.39↓ | 2.2108 | 0.6987↓ | 29.66 | 6.97↓ | 11.81 | 1.93↓ | 2.70 | 1.14↓ | 2.07 | 1.14↓ |
| InternVL-3.5 | 36.36 | 28.41↓ | **0.0495** | **0.0424**↑ | 4.91 | 5.27↑ | 1.94 | 2.32↑ | 0.52 | 2.26↑ | 0.52 | 0.55↑ |

For the RGB subset, the Agent demonstrates above-average performance in tool utilization and achieves efficiency close to that of expert annotations. However, its response accuracy remains substantially below the average. This limitation is closely tied to the capabilities of the tools within the Perception Toolkit. In certain cases, even when the Agent selects the same tools as those used in expert annotations, the outputs still diverge from the ground-truth answers due to the constraints of the underlying expert models. As the first attempt to develop an Agent framework for Earth Observation, our work highlights this challenge and encourages future EO Agent research to adopt more advanced expert models in order to overcome these limitations.

Table 10: Performance of different LLM-based agents on the **RGB** subset of Earth-Bench. We **bold** the best results and underline the runner-ups.

| Model | Accuracy | | Efficiency | | Tool-Any-Order | | Tool-In-Order | | Tool-Exact-Match | | Parameters | |
|---|---|---|---|---|---|---|---|---|---|---|---|---|
| | AP | IF | AP | IF | AP | IF | AP | IF | AP | IF | AP | IF |
| GPT-5 | **59.32** | 49.15↓ | 1.5312 | 1.5784↓ | 76.61 | 72.09↓ | 75.04 | 67.08↓ | 59.60 | 52.71↓ | 46.15 | 40.73↓ |
| Gemini-2.5 | 47.46 | 47.46 | 0.7926 | 0.8878↓ | 48.70 | 60.11↑ | 29.38 | 50.11↑ | 21.33 | 45.54↑ | 19.07 | 36.75↑ |
| GPT-4o | 45.76 | 35.59↓ | 0.8779 | 0.8939↓ | 71.86 | 66.89↓ | 66.60 | 61.13↓ | 64.76 | 60.00↓ | 46.65 | 47.54↑ |
| Kimik2 | 54.24 | **62.71**↑ | 1.5341 | 1.4104↑ | 75.65 | 80.62↑ | 71.37 | 79.90↑ | 51.48 | 63.32↑ | 43.12 | **52.19**↑ |
| DeepSeek-V3.1 | 55.93 | 54.24↓ | 1.6895 | 1.7966↓ | **88.98** | **89.21**↑ | **85.49** | **87.64**↑ | **71.54** | **74.05**↑ | **53.57** | 57.22↑ |
| Qwen3-Max | 49.15 | 38.98↓ | 0.8601 | 0.9785↓ | 65.25 | 68.14↑ | 50.28 | 56.58↑ | 47.88 | 51.55↑ | 34.04 | 43.93↑ |
| Seed-1.6 | 55.93 | 52.54↓ | 1.1948 | 0.9589↓ | 57.99 | 57.59↓ | 44.00 | 47.30↑ | 34.11 | 43.74↑ | 29.93 | 39.02↑ |
| LLaMA-4 | 37.29 | 27.12↓ | 0.3464 | 0.3790↑ | 27.20 | 36.52↑ | 0.47 | 15.61↑ | 0.47 | 12.43↑ | 0.47 | 11.25↑ |
| Qwen-Plus | 42.37 | 38.98↓ | 0.9721 | 1.0488↓ | 51.89 | 60.11↑ | 29.51 | 42.80↑ | 24.18 | 41.07↑ | 23.62 | 36.89↑ |
| GLM-4.5v | 16.95 | 28.81↑ | 1.3070 | 1.4535↓ | 47.91 | 48.42↑ | 27.36 | 33.54↑ | 21.93 | 27.38↑ | 19.73 | 24.98↑ |
| Mistral | 30.51 | 30.51 | 0.7215 | 0.7531↓ | 36.16 | 45.59↑ | 22.46 | 40.11↑ | 21.89 | 37.71↑ | 19.97 | 31.84↑ |
| Qwen3-32B | 25.42 | 27.12↑ | 0.7767 | 1.0891↓ | 45.93 | 56.47↑ | 27.54 | 41.69↑ | 26.41 | 41.69↑ | 25.28 | 37.74↑ |
| InternVL-3.5 | 25.42 | 27.12↑ | **0.2398** | **0.2606**↓ | 16.95 | 29.92↑ | 7.23 | 13.58↑ | 6.67 | 13.58↑ | 4.75 | 11.88↑ |

Overall, the comparative analysis across subsets highlights both the strengths and limitations of LLM-based Agents in Earth Observation. While LLMs achieve relatively strong performance on Product tasks, where success relies more on general reasoning and statistical skills, they remain less effective on tasks that demand specialized knowledge, such as those in the Spectrum subset, which involve interpreting raw spectral data. Moreover, Earth-Agent incorporates expert models within the Perception Toolkit for tasks such as segmentation and object detection, which significantly improves performance in the corresponding scenarios. However, the generalization ability of these expert models remains limited, as their outputs do not always align with ground-truth answers, even when the correct tools are selected. These findings suggest that future progress in EO Agents will depend not only on enhancing LLMs with domain-specific knowledge, but also on developing more robust and versatile expert models to ensure reliable performance across the diverse spectrum of Earth Observation tasks.

## D  ERROR ANALYSIS

To analyze the errors made by Agents with different LLM backbones on Earth-Bench tasks, we selected GPT-5 as a representative closed-source model, and Kimi-K2, Qwen3-Max, and Deepseek-V3.1 as representative open-source models. We counted the number of errors and categorized them into five types:

- Unaware of Termination Conditions: failure to recognize the task's termination condition, leading to repeated tool calls until reaching the maximum step limit;
- Tool Hallucination: attempts to invoke non-existent tools;
- File Hallucination: attempts to process non-existent files, i.e., providing invalid file or folder paths as tool inputs;
- Invalid Parameters: inputs that do not conform to the expected parameter format or are otherwise invalid;
- System Error: system-level failures caused by the runtime environment or external dependencies.

Figure 7 presents the frequency and distribution of these error types. Results show that GPT-5 produced the largest number of errors, while Kimi-K2 had the fewest. Except for GPT-5, the other models exhibited similar error counts across different regimes, and their error distributions did not vary significantly, suggesting that providing more detailed execution steps does not substantially improve tool-use proficiency.

In terms of error types, GPT-5 errors were dominated by Invalid Parameters, with occasional System Errors and File Hallucinations, but no Tool Hallucinations. In contrast, the three open-source

models demonstrated different error patterns: while Invalid Parameters remained a notable factor, it was not the primary source of errors. Instead, nearly 60% of their errors stemmed from Hallucinations and Unaware of Termination Conditions. We hypothesize that this difference is related to training strategies. Open-source models are often trained with reinforcement learning, which may encourage more exploratory outputs, thereby increasing the likelihood of hallucinations. Moreover, their reward functions are typically designed to shape behavioral style and output preferences rather than enhance factual knowledge, which could make models more prone to generating divergent or repetitive outputs and to overlooking termination conditions.

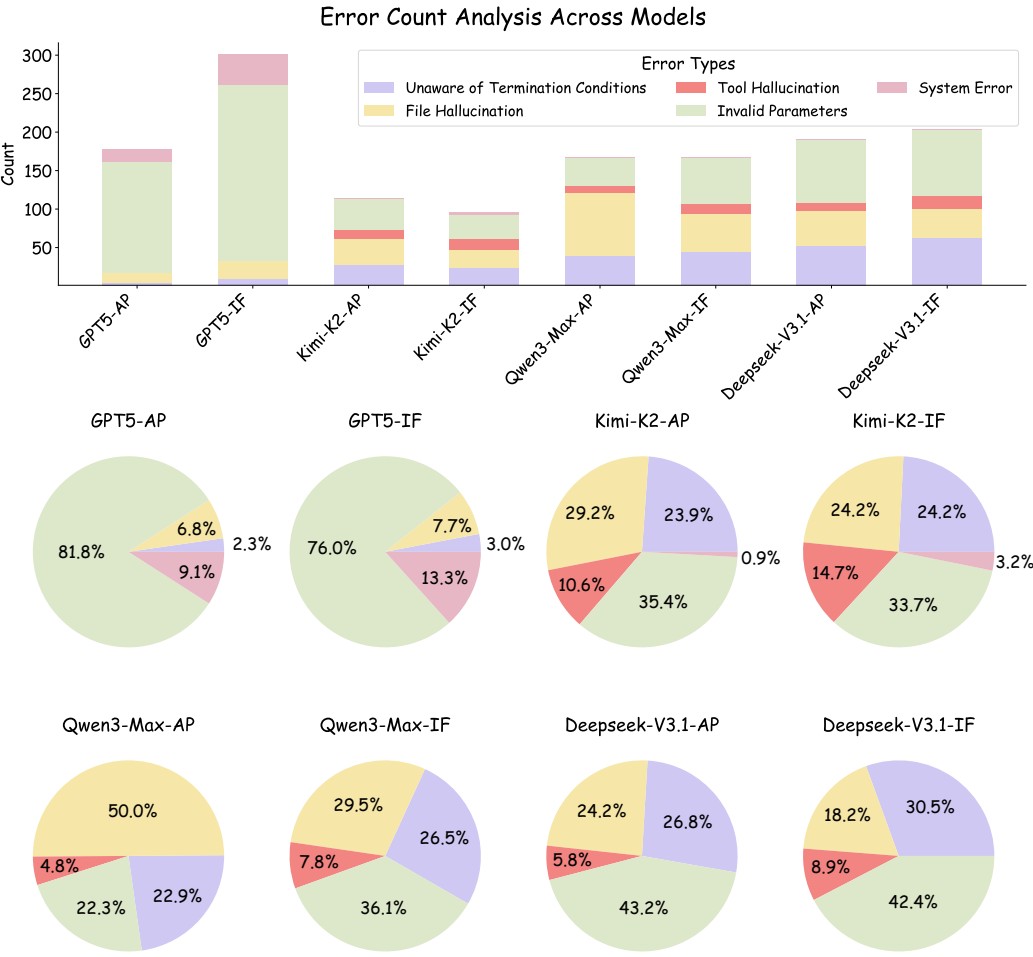

Figure 7: Error analysis.

# E LATENCY EXPERIMENT

## E.1 BREAKDOWN RESULTS ON LATENCY

We further break down the latency into **LLM Latency** and **Tool Latency**. It is evident that the majority of the latency is spent on model calls (LLM Latency), rather than on external tool calls (Tool Latency). This suggests that reducing the frequency of model calls could further improve latency. The results are shown in Table 11.

We observe that the majority of the latency is attributed to **LLM Latency**, while the impact of **Tool Latency** is relatively minimal. Therefore, reducing the frequency of model calls could significantly improve overall latency. This suggests that optimizing the model call frequency would further enhance system performance.

Table 11: Latency Breakdown for Different Models on Earth-Bench-Lite

| Model | Latency | Tool Call Number | LLM Latency | Tool Latency |
|---|---|---|---|---|
| Earth-Agent (GPT) | 9494.92s | 669 | 6969.26s | 2525.66s |
| Earth-Agent (DeepSeek-V3.1) | 4716.99s | 540 | 3066.76s | 1650.23s |
| Earth-Agent (Kimik2) | 7903.07s | 502 | 6306.59s | 1596.47s |

### E.2 OUR PROPOSED STRATEGY

We addressed this issue in the design of the Earth-Agent architecture. Specifically, we optimized the tool design to minimize unnecessary interactions with the model. For instance, by leveraging batch calculations for Earth indices, such as the NDVI calculation, we significantly reduce the frequency of model interactions, thereby lowering the overall latency. Below is an example of our approach:

**Batch Computing Strategy**

```python
def calculate_ndvi(input_nir_path, input_red_path, output_path):
    with rasterio.open(input_nir_path) as nir_src:
        nir_band = nir_src.read(1)  # Read the first band (assuming single-band rasters)
        nir_profile = nir_src.profile  # Get the metadata profile
    with rasterio.open(input_red_path) as red_src:
        red_band = red_src.read(1)  # Read the first band (assuming single-band rasters)
    nir_band = np.array(nir_band, dtype=np.float32)
    red_band = np.array(red_band, dtype=np.float32)
    nir_band = np.clip(nir_band, 0, 10000)
    red_band = np.clip(red_band, 0, 10000)
    valid_mask = (nir_band >= 0) & (nir_band <= 10000) & (red_band >= 0) & (red_band <= 10000)
    denominator = nir_band + red_band + 1e-6
    ndvi = (nir_band - red_band) / denominator
    # Set invalid pixels to -9999
    ndvi[~valid_mask] = -9999
    ndvi_profile = nir_profile.copy()
    ndvi_profile.update(
        dtype=rasterio.float32,  # NDVI values are floating-point numbers
        nodata=-9999,  # Set a NoData value
        compress='lzw'  # Optional: compress the output file
    )
    # Save the NDVI result to the specified output path
    os.makedirs((TEMP_DIR / output_path).parent, exist_ok=True)
    with rasterio.open(TEMP_DIR / output_path, 'w', **ndvi_profile) as dst:
        dst.write(ndvi.astype(rasterio.float32), 1)

    return f'Result save at {TEMP_DIR / output_path}'
@mcp.tool(description="""
Batch-calculate NDVI from multiple pairs of NIR/Red raster files and save results.

Parameters:
    input_nir_paths (list[str]): Paths to Near-Infrared (NIR) band raster files.
    input_red_paths (list[str]): Paths to Red band raster files.
    output_paths (list[str]): Relative output paths (e.g., "question17/ndvi_2022-01-16.
        tif") for each pair.

Returns:
    list[str]: A list of result messages (one per output), as returned by `
        calculate_ndvi`.
""")
def calculate_batch_ndvi(
    input_nir_paths: list[str],
    input_red_paths: list[str],
    output_paths: list[str]
) -> list[str]:
    return [
        calculate_ndvi(nir_path, red_path, out_path)
        for nir_path, red_path, out_path in zip(input_nir_paths, input_red_paths,
            output_paths)
    ]
```

## F  SCALABILITY DISCUSSION

Understanding the performance trends as the number of tool calls increases is crucial for evaluating the system's behavior, particularly in terms of latency and scalability under increasing task complexity.

### F.1  PERFORMANCE WITH RESPECT TO TOOL NUMBER

To address this, we conducted an ablation study examining the relationship between the **number of tools** used and **the system's performance** across three SOTA models: GPT5, DeepSeek-V3.1, and Kimik2. The following Table 12 presents the results, highlighting the high performance range for each model:

Table 12: Performance of Earth-Agent Models with respect to Tool Numbers. The **high performance range** is highlighted.

| Tool Numbers | GPT5 | | DeepSeek-V3.1 | | Kimik2 | |
|---|---|---|---|---|---|---|
| | Questions | Accuracy (%) | Questions | Accuracy (%) | Questions | Accuracy (%) |
| 0 | 25 | 8.00 | - | - | 25 | 4.00 |
| 1 | 3 | **66.67** | 2 | 0.00 | 1 | 0.00 |
| 2 | 8 | **75.00** | - | - | 5 | 40.00 |
| 3 | 42 | **80.95** | 23 | **73.91** | 17 | **76.47** |
| 4 | 29 | **89.66** | 23 | **86.96** | 21 | **66.67** |
| 5 | 30 | **60.00** | 19 | **68.42** | 21 | **71.43** |
| 6 | 7 | **85.71** | 15 | **60.00** | 27 | **70.37** |
| 7 | 10 | **90.00** | 15 | 40.00 | 16 | **56.25** |
| 8 | 10 | **80.00** | 8 | **87.50** | 20 | **65.00** |
| 9 | 4 | **75.00** | 10 | **60.00** | 4 | 25.00 |
| 10 | 10 | **70.00** | 11 | **54.55** | 5 | **40.00** |
| 11 | 4 | **50.00** | 8 | **75.00** | 11 | **54.55** |
| 12 | 3 | 33.33 | 8 | **75.00** | 9 | **66.67** |
| 13 | 6 | **100.00** | 17 | **70.59** | 20 | **80.00** |
| 14 | 4 | **75.00** | 4 | **50.00** | 3 | 33.33 |
| 15 | 2 | 0.00 | 5 | **100.00** | 4 | 50.00 |
| 16 | 2 | 100.00 | 3 | 66.67 | 1 | 0.00 |
| 17 | 4 | 50.00 | 4 | 25.00 | 3 | 66.67 |
| 18 | 4 | 50.00 | 4 | 25.00 | 2 | 50.00 |
| 19 | 3 | 0.00 | 3 | 0.00 | 1 | 0.00 |
| 20 | 3 | 33.33 | 3 | 33.33 | 1 | 100.00 |
| ... | ... | ... | ... | ... | ... | ... |
| 159 | 1 | 100.00 | - | - | - | - |

From the results, we observe distinct performance trends for each model:

- For Earth-Agent driven by GPT5, high accuracy is primarily concentrated within **the tool number range of 1 to 14**.

- For Earth-Agent driven by DeepSeek-V3.1, the high-performance range is within **the tool number range of 3 to 15**.

- For Earth-Agent driven by Kimik2, the high-performance range falls within **the tool number range of 3 to 13**.

These high performance ranges align with expectations and indicate that task complexity plays a key role in system performance. Using too few tools (with the extreme case being zero tools) results in low accuracy, as the agent is unable to solve the task effectively, often leading to early errors. Conversely, performance tends to degrade when too many tools are employed, suggesting that the current capabilities of the base LLMs may not be sufficient to handle long chains of reasoning efficiently.

## F.2 Performance with respect to Unique Tool Number

We also investigated the relationship between the **unique number of tools** used and **performance trends**. Below is a visual representation using a bubble chart, where the size of each bubble is proportional to the number of questions.

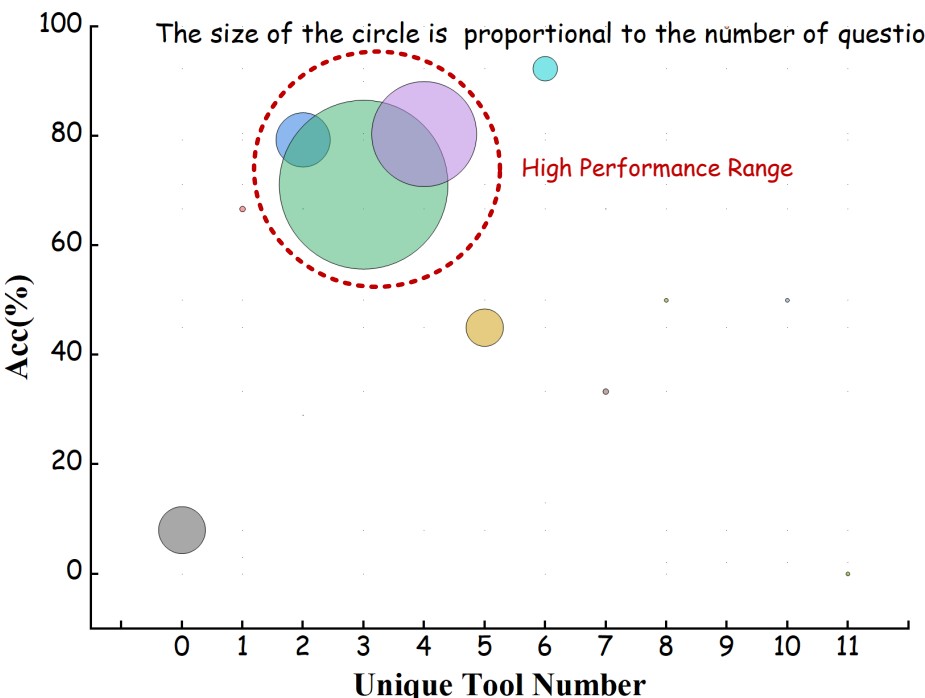

Figure 8: **GPT5: Performance with respect to Unique Tool Number**

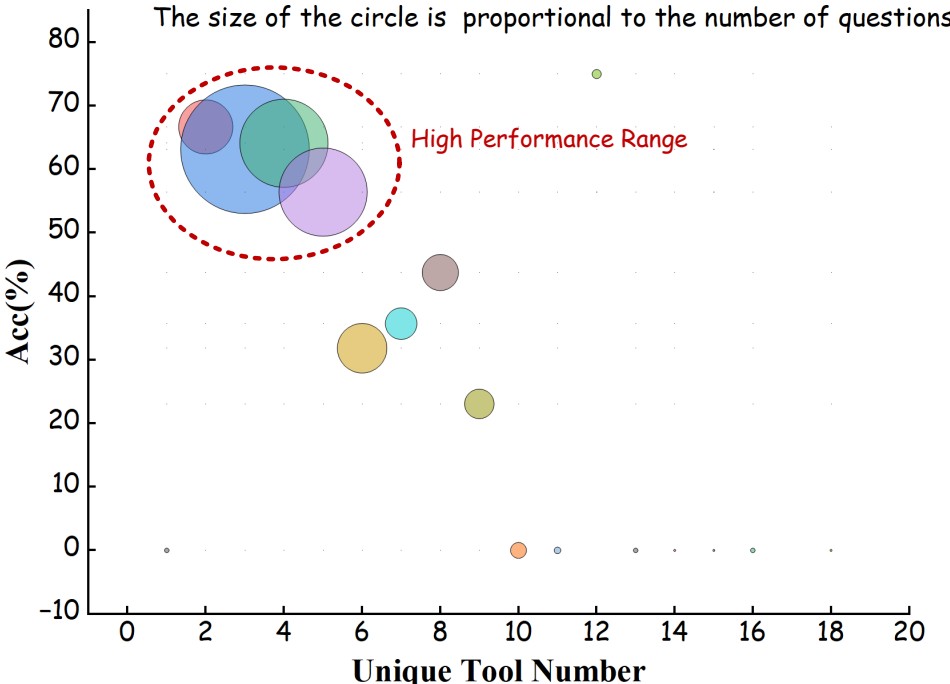

Figure 9: **DeepSeek-V3.1: Performance with respect to Unique Tool Number**

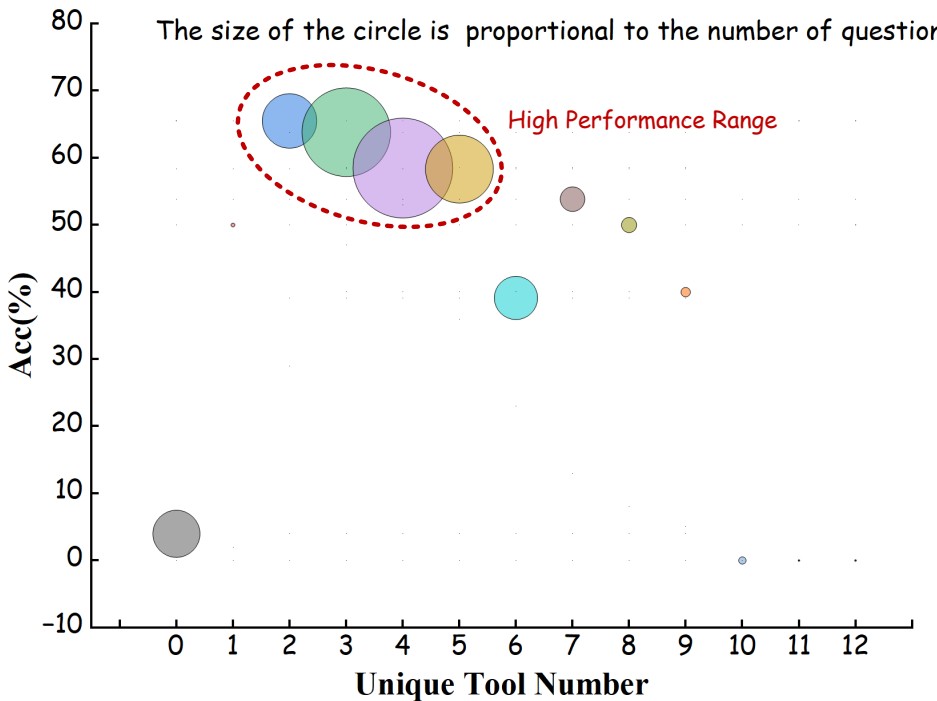

Figure 10: **Kimik2: Performance with respect to Unique Tool Number**

Based on Figures 8, 9, and 10, we observed the following trends in the Earth-Agent system: When driven by GPT5, high accuracy is primarily observed within the range of 2 to 5 unique tool calls. For DeepSeek, the optimal accuracy is concentrated in the range of 2 to 6 unique tool calls, while for Kimik2, high accuracy is predominantly found in the range of 1 to 6 unique tool calls. We also observed a performance decline when the number of tool calls becomes excessive. These findings align with our expectations, highlighting the current limitations of LLMs in handling tasks that involve long chains of reasoning or excessive tool interactions.

This analysis provides valuable insights into the scalability of the system as task complexity increases. It also offers important directions for future agent training, including (1) optimizing the agent's startup phase and (2) developing datasets for long-chain reasoning to enhance the agent's ability to handle multiple tool calls effectively.

## G  TOOL KIT LIST

The Index Toolkit offers a comprehensive suite of automated functions for computing a wide range of remote sensing indices directly from raster data. It supports efficient batch processing and covers commonly used indices related to vegetation, water, soil, snow, and burn severity, such as NDVI, NDWI, and NDBI. A detailed list of the implemented indices is provided in Table 13.

Table 13: List of detailed information of Index Toolkit.

| Tool Name | Category | Description Summary |
|---|---|---|
| calculate_batch_ndvi | Index | Batch-calculate NDVI from multiple pairs of NIR/Red raster files and save results. |
| calculate_batch_ndwi | Index | Batch-calculate NDWI from multiple pairs of NIR/SWIR raster files and save results. |
| calculate_batch_ndbi | Index | Batch-calculate NDBI from multiple pairs of SWIR/NIR raster files and save results. |

| | | |
|---|---|---|
| calculate_batch_evi | Index | Batch-calculate EVI from multiple sets of NIR/Red/Blue raster files and save results. |
| calculate_batch_nbr | Index | Batch-calculate NBR from multiple pairs of NIR/SWIR raster files and save results. |
| calculate_batch_fvc | Index | Batch-calculate FVC from multiple pairs of NIR/Red raster files and save results. |
| calculate_batch_wri | Index | Batch-calculate WRI from multiple sets of Green/Red/NIR/SWIR raster files and save results. |
| calculate_batch_ndti | Index | Batch-calculate NDTI from multiple pairs of Red/Green raster files and save results. |
| calculate_batch_frp | Index | Batch-calculate Fire Radiative Power (FRP) masks from multiple raster files and save results. |
| calculate_batch_ndsi | Index | Calculate NDSI for multiple pairs of Green and SWIR band images. |
| calc_extreme_snow_loss_percentage_from_binary_map | Index | Calculate the percentage of extreme snow and ice loss areas from a binary map. |
| compute_tvdi | Index | Compute TVDI (Temperature Vegetation Dryness Index) using NDVI and LST from local raster files. |

The Inversion Toolkit integrates a collection of algorithms for retrieving key geophysical and environmental parameters from optical, thermal infrared, and microwave remote sensing data. It supports multiple retrieval methods for parameters such as land surface temperature (LST), land surface emissivity, and precipitable water vapor (PWV), including single-channel, multi-channel, and split-window approaches. By enabling flexible, efficient, and reproducible parameter estimation across multi-source Earth Observation data, the toolkit provides a versatile foundation for quantitative remote sensing applications. A detailed list of the implemented algorithms is provided in Table 14.

Table 14: List of detailed information of Inversion Toolkit.

| Tool Name | Category | Description Summary |
|---|---|---|
| band_ratio | Inversion | Compute Precipitable Water Vapor (PWV) image from local MODIS surface reflectance band files using the band ratio method. |
| lst_single_channel | Inversion | Estimate Land Surface Temperature (LST) using the Single-Channel method, with NDVI-based emissivity estimation from RED and NIR bands. |
| lst_multi_channel | Inversion | Estimate Land Surface Temperature (LST) using the multi-channel algorithm. |
| split_window | Inversion | Estimate Land Surface Temperature (LST) or Precipitable Water Vapor (PWV) using the split-window algorithm. |
| temperature_emissivity_separation | Inversion | Estimate Land Surface Temperature (LST) using an enhanced Temperature Emissivity Separation (TES) algorithm with empirical emissivity estimation. |

| modis_day_night_lst | Inversion | Estimate land surface temperature (LST) from local MODIS Day and Night brightness temperatures using a single-channel correction method. |
|---|---|---|
| ttm_lst | Inversion | Estimate land surface temperature (LST) and emissivity using improved Three-Temperature Method (TTM) from three local thermal band GeoTIFF files. Uses all three bands to form a system of equations and solves per-pixel with physical constraints. |
| calculate_mean_lst_by_ndvi | Inversion | Calculate the average Land Surface Temperature (LST) across multiple images where NDVI is either above or below a given threshold. |
| calculate_max_lst_by_ndvi | Inversion | Calculate the maximum Land Surface Temperature (LST) in areas where NDVI is above or below a given threshold. |
| ATI | Inversion | Estimate Apparent Thermal Inertia (ATI) using the Thermal Inertia Method. This method calculates ATI as $(1 - albedo)/(day\_temp - night\_temp)$, which serves as a proxy for land surface temperature stability over diurnal cycles. |
| dual_polarization_differential | Inversion | Dual-Polarization Differential Method (DPDM) for microwave remote sensing parameter inversion. Supports soil moisture and vegetation index estimation with improved data handling and flexible parameters. |
| dual_frequency_diff | Inversion | Dual-frequency Differential Method (DDM) for parameter inversion using local raster data. Supports inversion of multiple parameters via empirical linear models: Soil Moisture (SM): param = alpha*(band1 - band2) + beta; Vegetation Index (VI): param = alpha*(band1 - band2) + beta; Leaf Area Index (LAI): param = alpha*(band1 - band2) + beta |
| multi_freq_bt | Inversion | Multi-frequency Brightness Temperature Method for parameter inversion using local raster data. |
| chang_single_param_inversion | Inversion | Chang algorithm for inversion of a single parameter using multi-frequency dual-polarized microwave brightness temperatures from local raster files. |
| nasa_team_sea_ice_concentration | Inversion | Estimate Sea Ice Concentration using NASA Team Algorithm from local passive microwave brightness temperature GeoTIFF files. |
| dual_polarization_ratio | Inversion | Estimate Vegetation Water Content (VWC) or Soil Moisture (SM) using Dual-Polarization Ratio Method (PRM) from local passive microwave brightness temperature GeoTIFF files. The polarization ratio is computed as: (V - H) / (V + H), where V and H are brightness temperatures of vertical and horizontal polarizations. |

| | | |
|---|---|---|
| calculate_water_turbidity_ntu | Inversion | Calculate water turbidity in NTU (Nephelometric Turbidity Units) from red band raster file and save the result to a specified output path. |

The Perception Toolkit provides a comprehensive set of remote sensing perception tools, covering a wide range of tasks such as scene classification, object detection, and change detection. In addition, it supports threshold-based segmentation and offers a series of post-processing utilities for bounding box and contour refinement. Overall, the toolkit enables diverse perception tasks on RGB remote sensing imagery, including scene recognition, semantic segmentation, and spatiotemporal change detection. A detailed list of the implemented tools is provided in Table 15.

Table 15: List of detailed information of Perception Toolkit.

| Tool Name | Category | Description Summary |
|---|---|---|
| MSCN | Perception | MSCN is a scene and land-use image classifier, effective for categories such as Airport, BareLand, BaseballField, Beach, Bridge, Center, Church, Commercial, DenseResidential, Desert, Farmland, Forest, Industrial, Meadow, MediumResidential, Mountain, Park, Parking, Playground, Pond, Port, RailwayStation, Resort, River, School, SparseResidential, Square, Stadium, StorageTanks, and Viaduct. |
| RemoteCLIP | Perception | RemoteCLIP is a scene and land-use image classifier, specialized for categories such as Airport, Beach, Bridge, Commercial, Desert, Farmland, FootballField, Forest, Industrial, Meadow, Mountain, Park, Parking, Pond, Port, RailwayStation, Residential, River, and Viaduct. |
| Strip_R_CNN | Perception | Strip_R_CNN is a remote sensing object detection model with a strong focus on maritime and ship-related targets. Compared to SM3Det, it is particularly specialized in detecting and localizing different types of ships and naval vessels. This model is highly effective at detecting the following categories: L3 ship, L3 warcraft, L3 merchant ship, L3 aircraft carrier, Arleigh Burke, Container, Ticonderoga, Perry, Tarawa, WhidbeyIsland, CommanderA, Austen, Nimitz, Sanantonio, Container, Car carrierB, Enterprise, Car carrierA, Medical |

| | | |
|---|---|---|
| SM3Det | Perception | SM3Det is a remote sensing object detection model. Given an input image and a natural language prompt specifying the target object (e.g., "plane", "ship", "storage tank"), it detects all instances of that object and returns their bounding boxes. This model is particularly strong at detecting and localizing the following categories:plane, ship, storage tank, baseball diamond, tennis court, basketball court, ground track field, harbor, bridge, large vehicle, small vehicle, helicopter, roundabout, soccer ball field, swimming pool. |
| RemoteSAM | Perception | RemoteSAM is a remote sensing visual grounding model. Given an input image and a text prompt describing a region of interest (e.g., "the football field located on the westernmost side"), it outputs the corresponding bounding box coordinates. |
| InstructSAM | Perception | InstructSAM is an instruction-guided counting model for remote sensing images. Given an input image and a natural language prompt specifying the target object (e.g., "storage tank", "football field"), it detects and counts the number of instances matching the description. |
| SAM2 | Perception | Use SAM2 to segment the input image and return the path of the segmented image. |
| ChangeOS | Perception | Use ChangeOS to detect the change between two images and return the change mask. Can also be used to segment building by providing same image path in pre_image_path and post_image_path. |
| threshold_segmentation | Perception | Perform threshold-based segmentation on a single-band raster image. The function reads a raster image from the specified path, converts it to a binary mask by applying a fixed threshold, and writes the resulting binary image to a new file. Pixel values greater than the threshold are set to 255 (white), and values less than or equal to the threshold are set to 0 (black). |
| bbox_expansion | Perception | Expands bounding boxes by a given radius and returns the expanded bounding boxes. |
| count_above_threshold | Perception | Count the number of pixels in an image whose values are greater than the specified threshold. |
| count_connected_components | Perception | Read a binary image and return the count of connected components. |
| bboxes2centroids | Perception | Convert bounding boxes from [x_min, y_min, x_max, y_max] format to centroid coordinates (x, y). |
| centroid_distance_extremes | Perception | Compute pairwise distances between centroids and return both the closest and farthest pairs with their indices and distances. |
| calculate_bbox_area | Perception | Calculate the total area of a list of bounding boxes in [x, y, w, h] format. |

The Analysis Toolkit provides a suite of statistical and spatiotemporal analysis methods tailored for remote sensing and geoscience data. Its functionalities include classical time-series trend detection and decomposition techniques such as linear regression, the Mann–Kendall test, Sen's slope estimation, and STL decomposition. It also supports change-point detection and seasonal analysis based on autocorrelation. In addition, the toolkit integrates spatial statistical approaches, including hotspot direction analysis, as well as methods for spike detection in numerical sequences. A detailed list of the implemented tools is provided in Table 16.

Table 16: List of detailed information of Analysis Toolkit.

| Tool Name | Category | Description Summary |
|---|---|---|
| compute_linear_trend | Analysis | Computes the linear trend (slope and intercept) of a time series by fitting a line of the form: $y = a \cdot x + b$ using the least squares method. |
| mann_kendall_test | Analysis | Perform the non-parametric Mann-Kendall trend test on a univariate time series. The test evaluates whether there is a monotonic upward or downward trend without requiring the data to conform to any particular distribution. |
| sens_slope | Analysis | Compute Sen's Slope estimator for a univariate time series. Sen's Slope is a robust non-parametric method for estimating the median rate of change over time, often used with the Mann-Kendall test to assess both trend and magnitude. |
| stl_decompose | Analysis | Apply Seasonal-Trend decomposition using LOESS (STL) to a univariate time series. Decomposes the series into trend, seasonal, and residual components. |
| detect_change_points | Analysis | Detect structural change points in a univariate time series using the ruptures library with the PELT algorithm. A change point marks a location where the statistical properties of the signal shift (e.g., mean or variance). |
| autocorrelation_function | Analysis | Compute the Autocorrelation Function (ACF) for a univariate time series. The ACF measures the correlation of the series with its own lags, which is useful for detecting seasonality, persistence, and lag dependence. |
| detect_seasonality_acf | Analysis | Detect the dominant seasonality (period) in a univariate time series using the Autocorrelation Function (ACF). The method searches for significant peaks in the ACF beyond lag=1 to identify repeating cycles. |
| getis_ord_gi_star | Analysis | Compute the Getis-Ord Gi* statistic for local spatial autocorrelation on a raster image. This method identifies statistically significant spatial clusters of high (hot spots) or low (cold spots) values using a user-specified spatial weight kernel. |

| analyze_hotspot_direction | Analysis | Analyze the main directional concentration of hotspots in a binary hotspot map. The function counts the number of hotspot pixels (value=1) in each cardinal direction relative to the map center, and returns the dominant direction. |
| count_spikes_from_values | Analysis | Count the number of upward spikes in a sequence of numerical values. A spike is defined as a positive difference between consecutive valid values greater than the given threshold. |

The Statistics Toolkit offers a comprehensive set of functions for descriptive statistics, image-based statistical analysis, and geospatial data processing. Its capabilities cover the calculation of classical statistical measures such as mean, variance, and skewness, as well as the extraction of statistical information from imagery and intersection-based threshold analysis. In addition, the toolkit provides fundamental arithmetic operations, temperature unit conversions, and image differencing functions. It also supports essential preprocessing tasks, including radiometric correction and cloud masking. Overall, the toolkit enables flexible and efficient extraction and analysis of statistical features from geoscience and remote sensing data. A detailed list of the implemented tools is provided in Table 17.

Table 17: List of detailed information of Statistics Toolkit.

| Tool Name | Category | Description Summary |
| --- | --- | --- |
| coefficient_of_variation | Statistics | Compute the Coefficient of Variation (CV) for a dataset. The CV is defined as the ratio of the standard deviation to the mean and is commonly used as a normalized measure of dispersion. |
| skewness | Statistics | Compute the skewness of a dataset, which measures the asymmetry of the probability distribution. |
| kurtosis | Statistics | Compute the kurtosis of a dataset, which measures the ẗailedness öf the distribution. |
| calc_batch_image_mean | Statistics | Compute mean value of an batch of images. |
| calc_batch_image_std | Statistics | Compute the standard deviation (spread of pixel values) for a batch of images. |
| calc_batch_image_median | Statistics | Compute the median pixel value for a batch of images. |
| calc_batch_image_min | Statistics | Compute the minimum pixel value for a batch of images. |
| calc_batch_image_max | Statistics | Compute the maximum pixel value for a batch of images. |
| calc_batch_image_skewness | Statistics | Compute the skewness of pixel value distributions for a batch of images. Skewness quantifies the asymmetry of the distribution:1. Positive skew → longer right tail; 2. Negative skew → longer left tail; 3. Zero skew → symmetric distribution. |
| calc_batch_image_kurtosis | Statistics | Compute the kurtosis of pixel value distributions for a batch of images. Kurtosis measures the ẗailedness öf the distribution relative to a normal distribution. |

| calc_batch_image_sum | Statistics | Compute the sum of pixel values for a batch of images. |
|---|---|---|
| calc_batch_image_hotspot _percentage | Statistics | Compute the hotspot percentage (fraction of pixels above a threshold) for a batch of images. |
| calc_batch_image_hotspot_tif | Statistics | Create binary hotspot maps for a batch of images, where pixels below a specified threshold are set to 1 (hotspot) and others set to 0. The output is saved as GeoTIFF files, preserving georeference metadata from the input images. |
| difference | Statistics | Compute the absolute difference between two numbers. |
| division | Statistics | Perform division between two numbers. |
| percentage_change | Statistics | Calculate the percentage change between two numbers, useful for comparing relative growth or decline. |
| kelvin_to_celsius | Statistics | Convert temperature from Kelvin to Celsius. |
| celsius_to_kelvin | Statistics | Convert temperature from Celsius to Kelvin. |
| max_value_and_index | Statistics | Find the maximum value in a list and return both the maximum value and its index. |
| min_value_and_index | Statistics | Find the minimum value in a list and return both the minimum value and its index. |
| ceil_number | Statistics | Return the ceiling (rounded up integer) of a given number. |
| multiply | Statistics | Multiply two numbers and return their product. |
| get_list_object_via_indexes | Statistics | Retrieve elements from a list using a list or tuple of indices. |
| mean | Statistics | Compute the arithmetic mean (average) of a dataset. |
| calculate_threshold_ratio | Statistics | Calculate the average percentage of pixels above a given threshold for one or more images and a specified band. |
| calc_batch_fire_pixels | Statistics | Compute the number of fire pixels (FRP ¿ threshold) for a batch of images. |
| create_fire_increase_map | Statistics | Create a binary map highlighting areas where fire increase exceeds a specified threshold. |
| identify_fire_prone_areas | Statistics | Identify fire-prone areas from a hotspot map based on a given percentile threshold. |
| get_percentile_value_from_image | Statistics | Calculate the N-th percentile value of pixel values in a raster image, and return it as a native Python type matching the image's data type. |
| image_division_mean | Statistics | Calculate the mean of pixel-wise division between two images or between two bands of the same image. |
| calculate_intersection_percentage | Statistics | Calculate the percentage of pixels that simultaneously satisfy threshold conditions in two raster images. |

| | | |
|---|---|---|
| calc_batch_image_mean_mean | Statistics | Compute the average of mean pixel values across a batch of images. |
| calc_batch_image_mean_max | Statistics | Compute the mean pixel values of a batch of images and return the maximum mean. |
| calc_batch_image_mean_max_min | Statistics | Compute the batch-wise statistics across multiple images, including: Mean of mean values, Maximum of maximum values, Minimum of minimum values. |
| calc_batch_image_mean_threshold | Statistics | Calculate the percentage or count of images whose mean pixel values (in a specified band) are above or below a given threshold. |
| calculate_multi_band_threshold _ratio | Statistics | Calculate the percentage of pixels that simultaneously satisfy multiple band threshold conditions. |
| count_pixels_satisfying_conditions | Statistics | Count the number of pixels that simultaneously satisfy multiple band threshold conditions. |
| count_images_exceeding_threshold _ratio | Statistics | Count how many images have a percentage of pixels above or below a threshold that exceeds a specified ratio. |
| average_ratio_exceeding_threshold | Statistics | Calculate the average percentage of pixels exceeding a value threshold, considering only images where the ratio is greater than a specified ratio threshold. |
| count_images_exceeding_mean _multiplier | Statistics | Count how many images have a mean pixel value above or below a multiple of the overall mean pixel value across all images. |
| calculate_band_mean_by_condition | Statistics | Calculate the mean value of a target band over pixels where a condition band satisfies a threshold. |
| calc_threshold_value_mean | Statistics | Calculate the mean value of corresponding raster pixels in path2 where the raster values in path1 exceed the given threshold. |
| calculate_tif_difference | Statistics | Calculate difference between two tif files (image_b - image_a) and save result. |
| subtract | Statistics | Subtract two images and save result. |
| calculate_area | Statistics | This function calculates the area of non-zero pixels in the input image and returns the result. |
| grayscale_to_colormap | Statistics | Apply a colormap to a grayscale image and save as a color image. |
| get_filelist | Statistics | Returns a list of files in the specified directory. |
| radiometric_correction_sr | Statistics | Apply Landsat 8 surface reflectance (SR_B*) radiometric correction. |
| apply_cloud_mask | Statistics | Apply cloud / shadow mask to a single Landsat 8 surface reflectance band using QA_PIXEL band. |

## G.1 Tool Prompt

To better illustrate the functionality of the toolkits, we provide a representative example. Specifically, we focus on the `lst_multi_channel` tool, which estimates LST using a multi-channel algorithm. This method leverages multiple thermal infrared bands from remote sensing imagery and applies empirical formulas to derive accurate LST values. The corresponding implementation is provided below:

**Tool Example**

```python
@mcp.tool(description='''
Estimate Land Surface Temperature (LST) using the multi-channel algorithm.
Requires local input files:
- Two thermal infrared bands (e.g., Band 31 and Band 32) as GeoTIFF files.

Parameters:
    band31_path (str): Path to local GeoTIFF file for thermal band 31 (~11 \mu m}).
    band32_path (str): Path to local GeoTIFF file for thermal band 32 (~12 \mu m).
    output_path (str): Relative path for the output raster file, e.g. "question17/
        lst_2022-01-16.tif"

Returns:
    str: Local file path of the exported LST image.
''')
def lst_multi_channel(band31_path: str, band32_path: str, output_path: str) -> str:
    """
    Description:
        Estimate Land Surface Temperature (LST) using the multi-channel algorithm.
        This method combines two thermal infrared bands to reduce atmospheric effects.

    Parameters:
        band31_path (str): Path to GeoTIFF file for thermal band 31 (~11 \mu m)
        band32_path (str): Path to GeoTIFF file for thermal band 32 (~12 \mu m)
        output_path (str): Relative path for the output LST GeoTIFF

    Returns:
        str: Full path to the saved LST GeoTIFF
    """
    import os, rasterio
    import numpy as np

    with rasterio.open(band31_path) as src31:
        band31 = src31.read(1).astype(np.float32)
        profile = src31.profile

    with rasterio.open(band32_path) as src32:
        band32 = src32.read(1).astype(np.float32)

    a = 1.022
    b = 0.47
    c = 0.43

    lst = a * band31 + b * (band31 - band32) + c

    profile.update(dtype=rasterio.float32, count=1, compress='lzw')

    os.makedirs((TEMP_DIR / output_path).parent, exist_ok=True)

    with rasterio.open(TEMP_DIR / output_path, 'w', **profile) as dst:
        dst.write(lst.astype(np.float32), 1)

    return f'Result saved at {TEMP_DIR / output_path}'
```

## H  EARTH-AGENT WITH DIFFERENT LLM BACKBONES

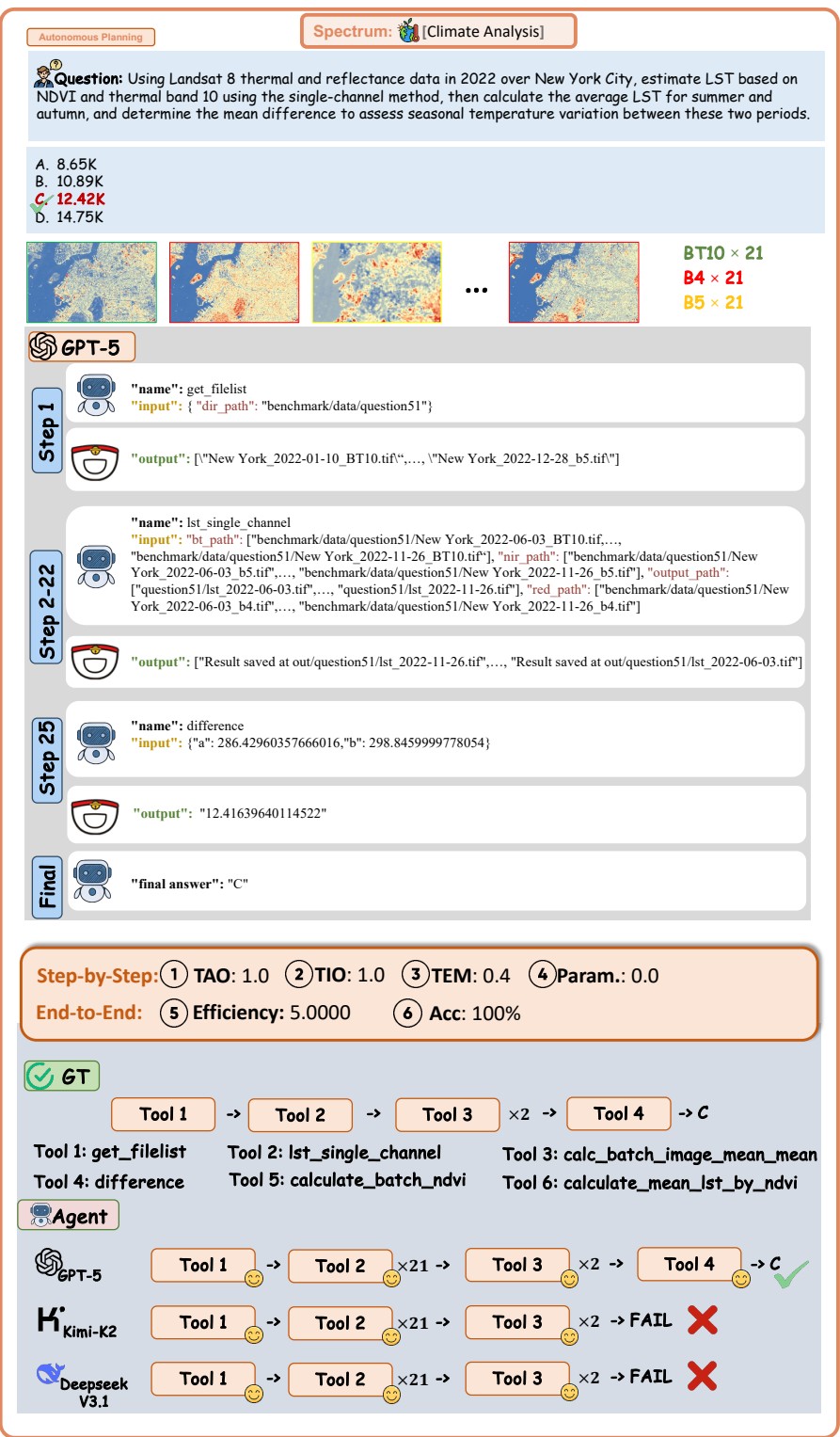

Figure 11: **Example of Climate Analysis with Spectrum Data under the Auto-Planning Regime.**

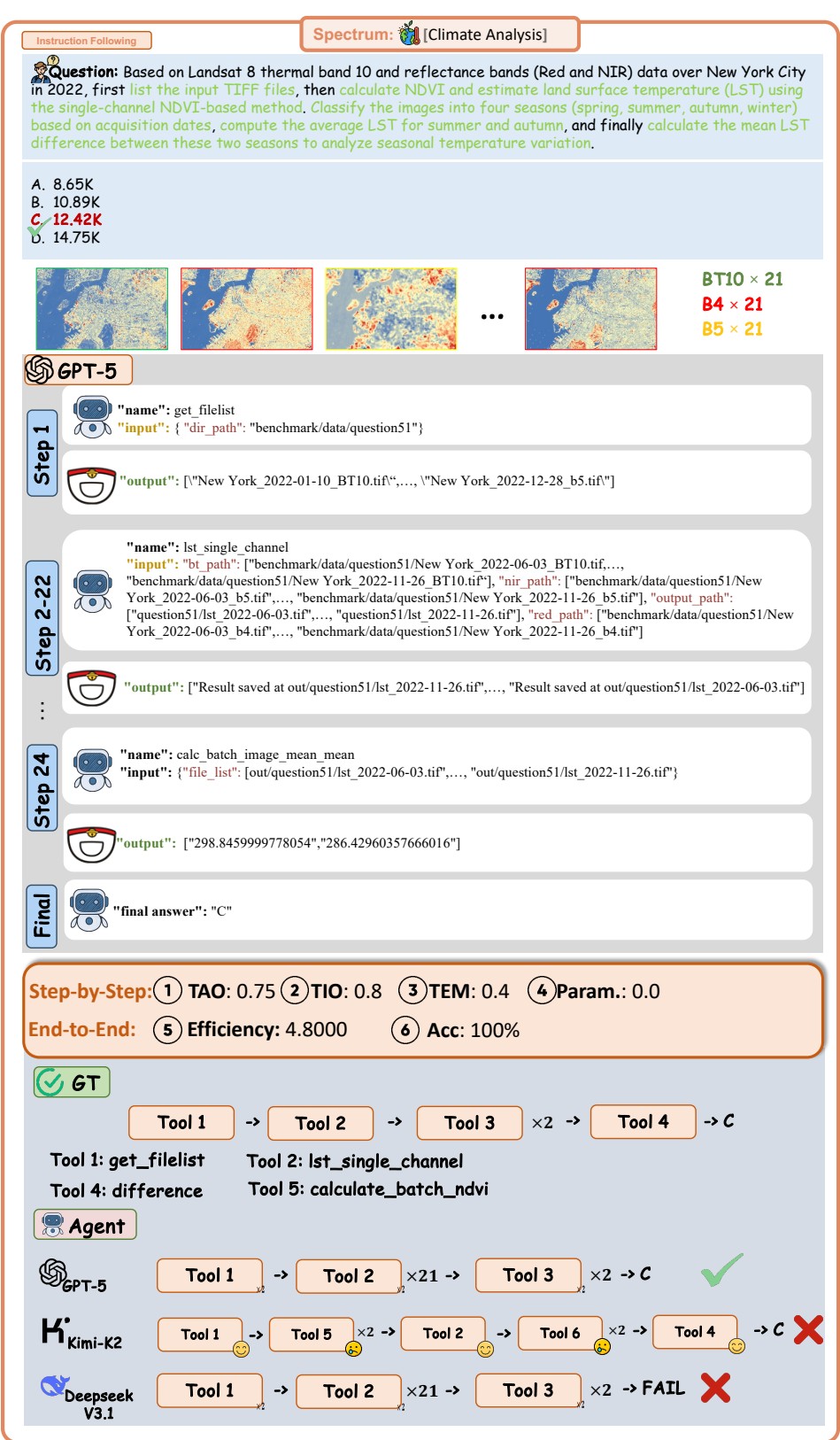

Figure 12: **Example of Climate Analysis with Spectrum Data under the Instruction-Following Regime.**

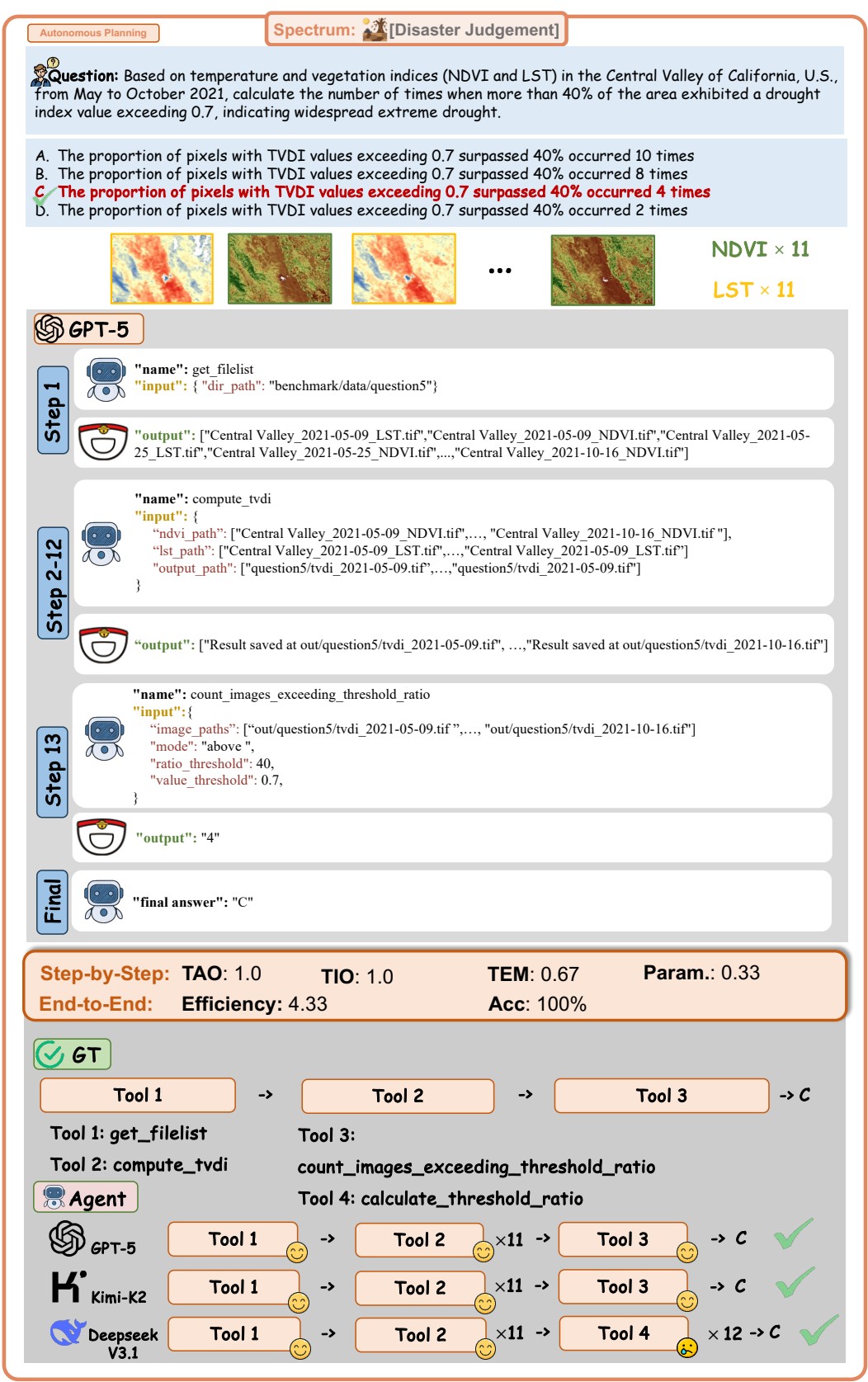

Figure 13: **Example of Disaster Judgement with Spectrum Data under the Auto-Planning Regime.**

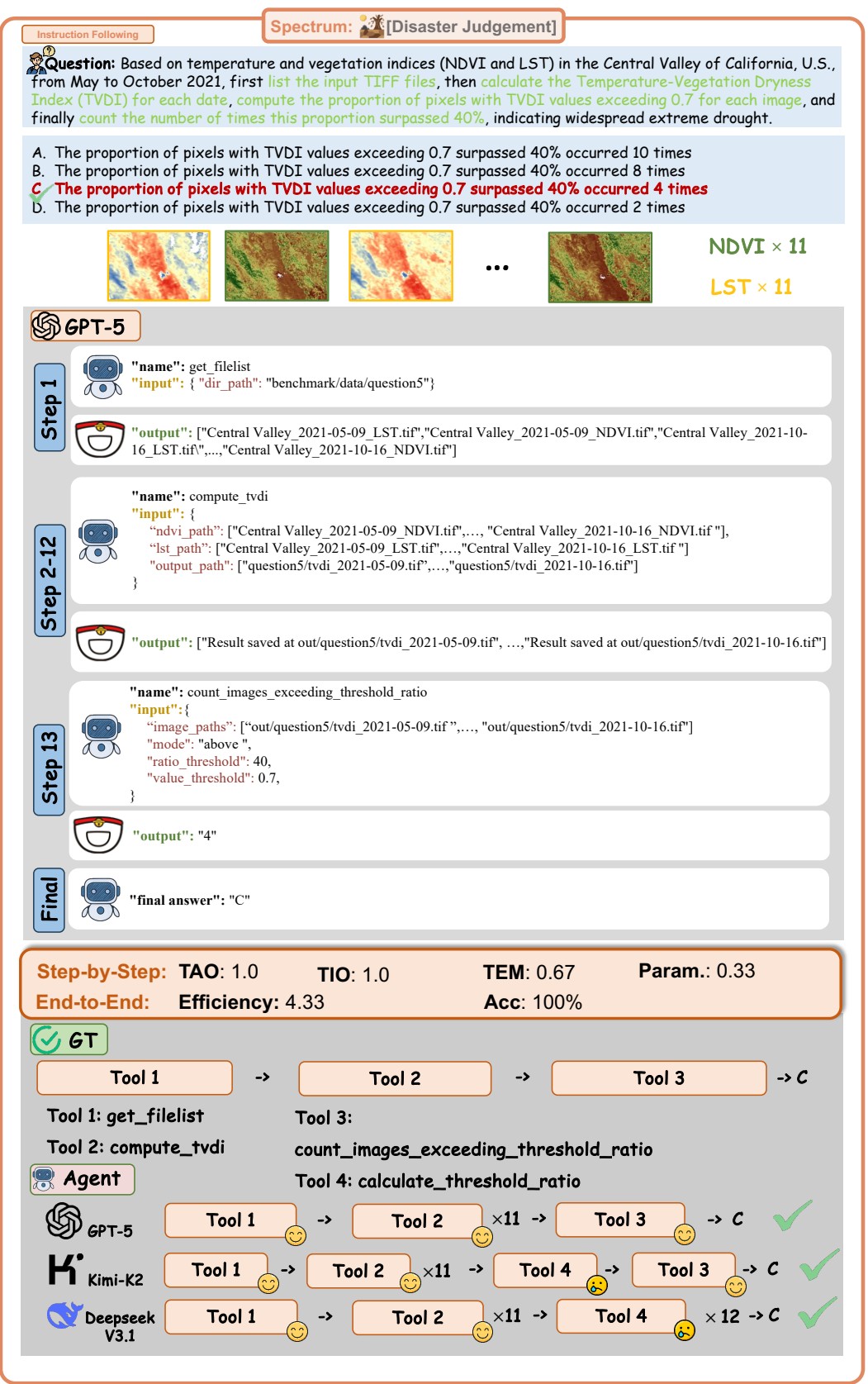

Figure 14: **Example of Disaster Judgement with Spectrum Data under the Instruction-Following Regime.**

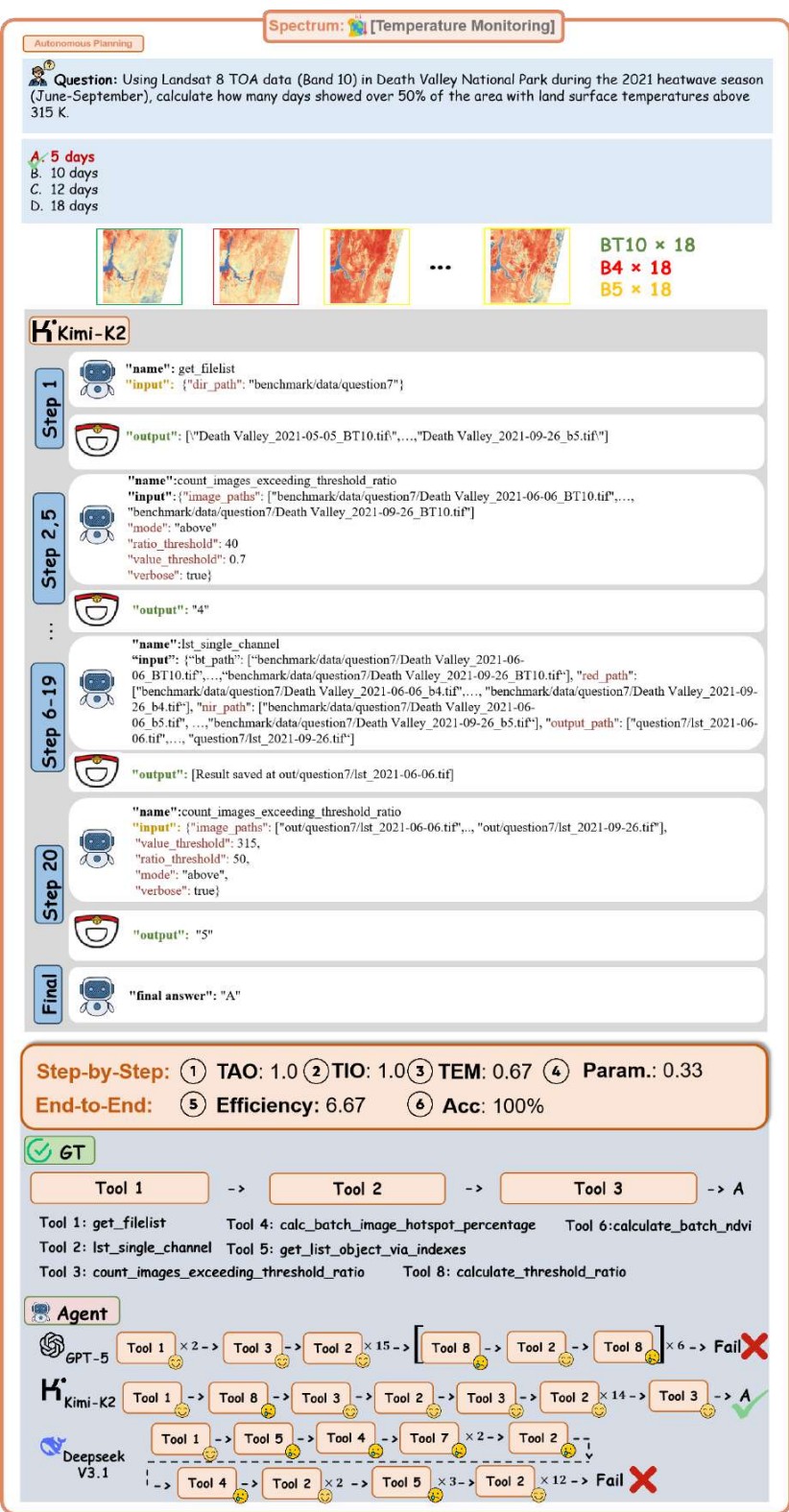

Figure 15: Example of Temperature Monitoring with Spectrum Data under the Auto-Planning Regime.

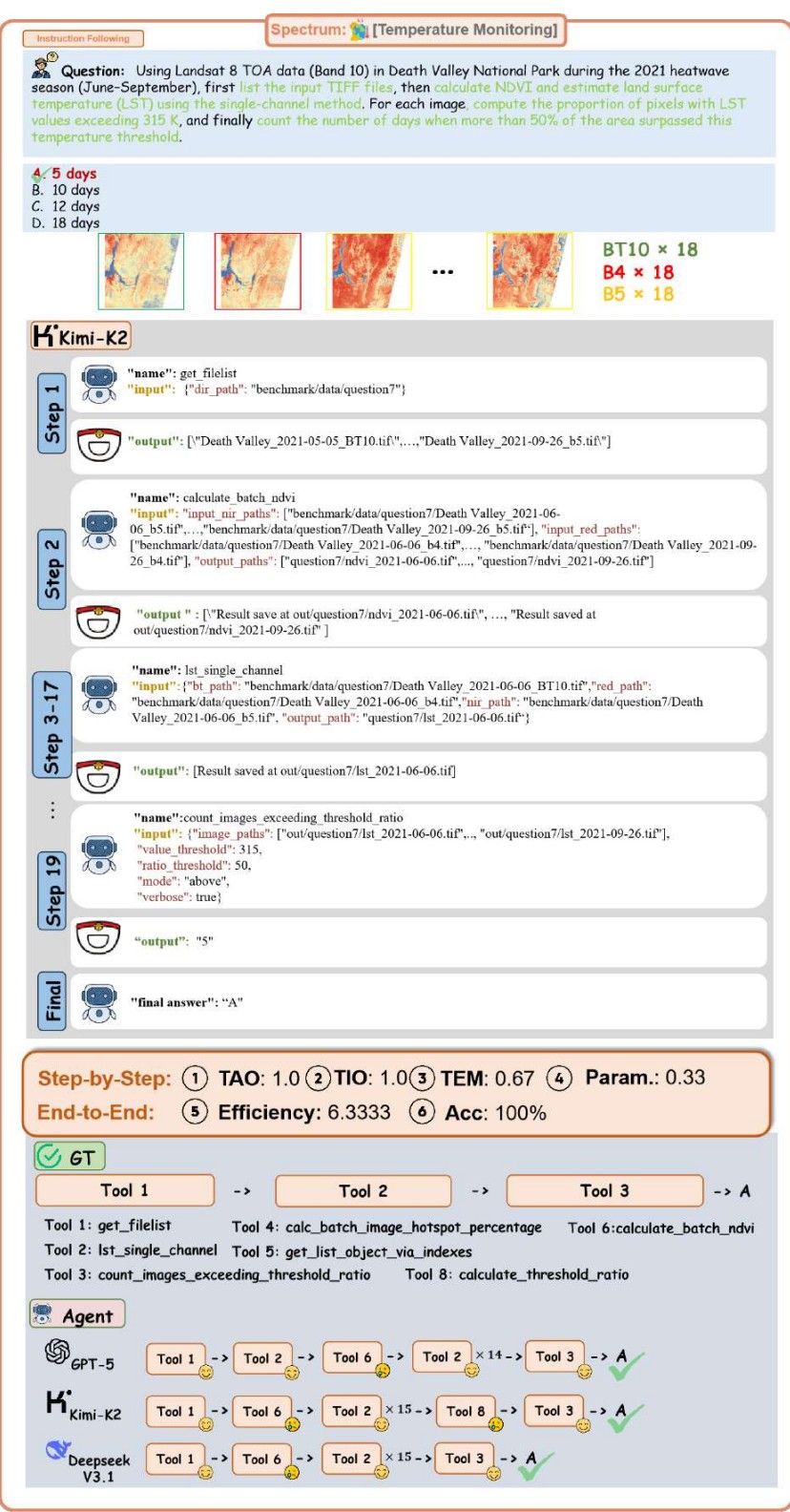

Figure 16: **Example of Temperature Monitoring with Spectrum Data under the Instruction-Following Regime.**

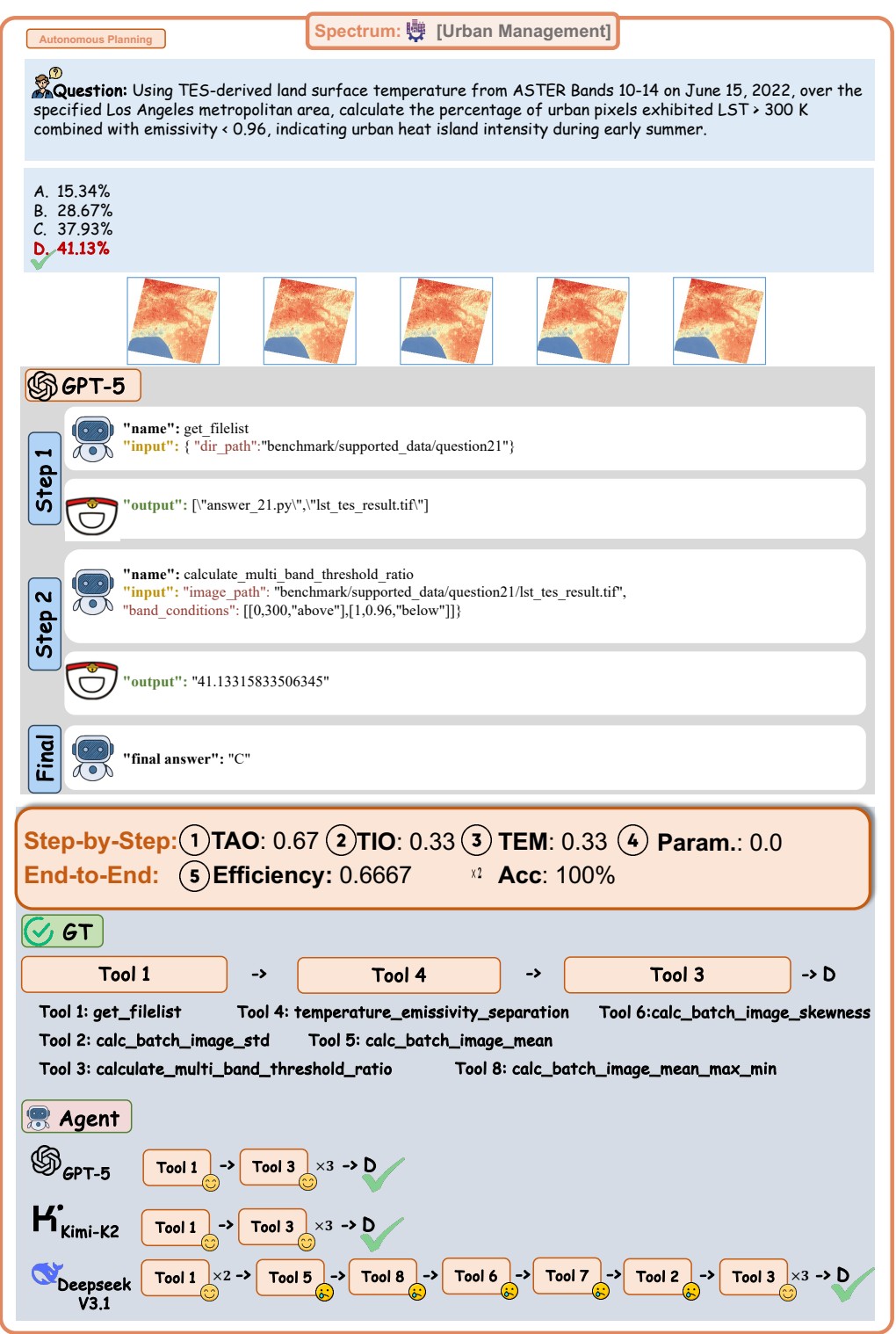

Figure 17: **Example of Urban Management with Spectrum Data under the Auto-Planning Regime.**

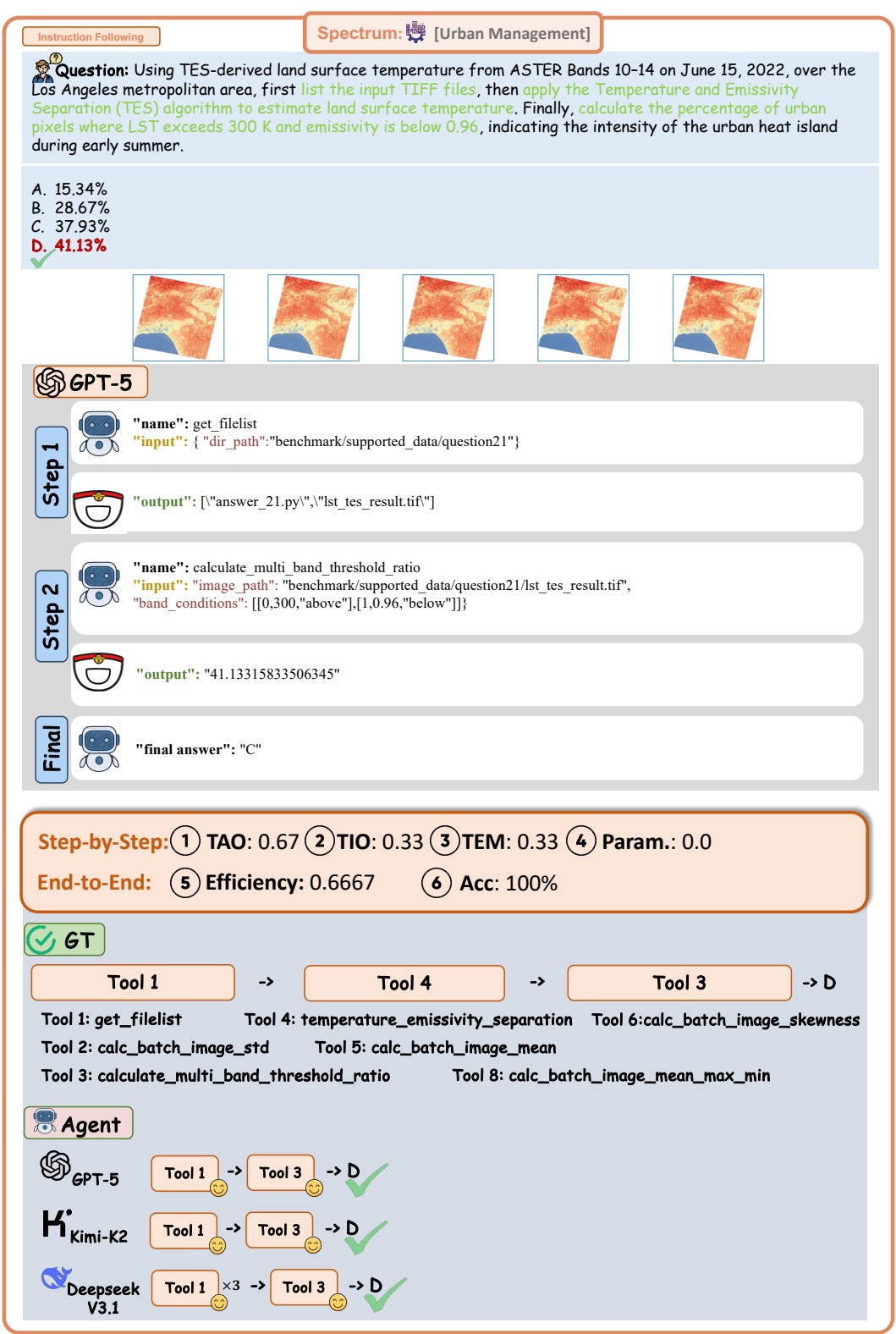

Figure 18: **Example of Urban Management with Spectrum Data under the Instruction-Following Regime.**

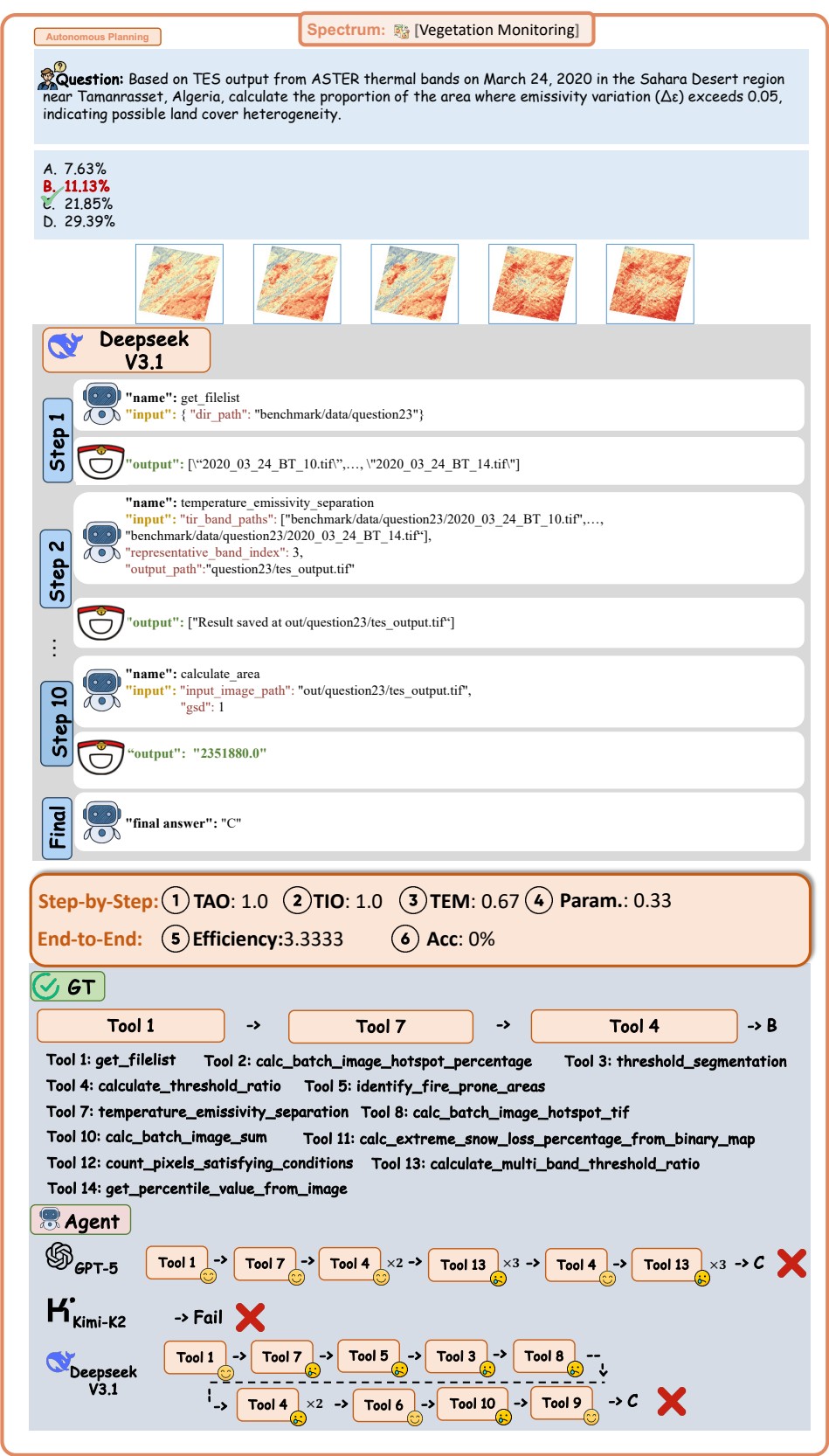

Figure 19: **Example of Vegetation Monitoring with Spectrum Data under the Auto-Planning Regime.**

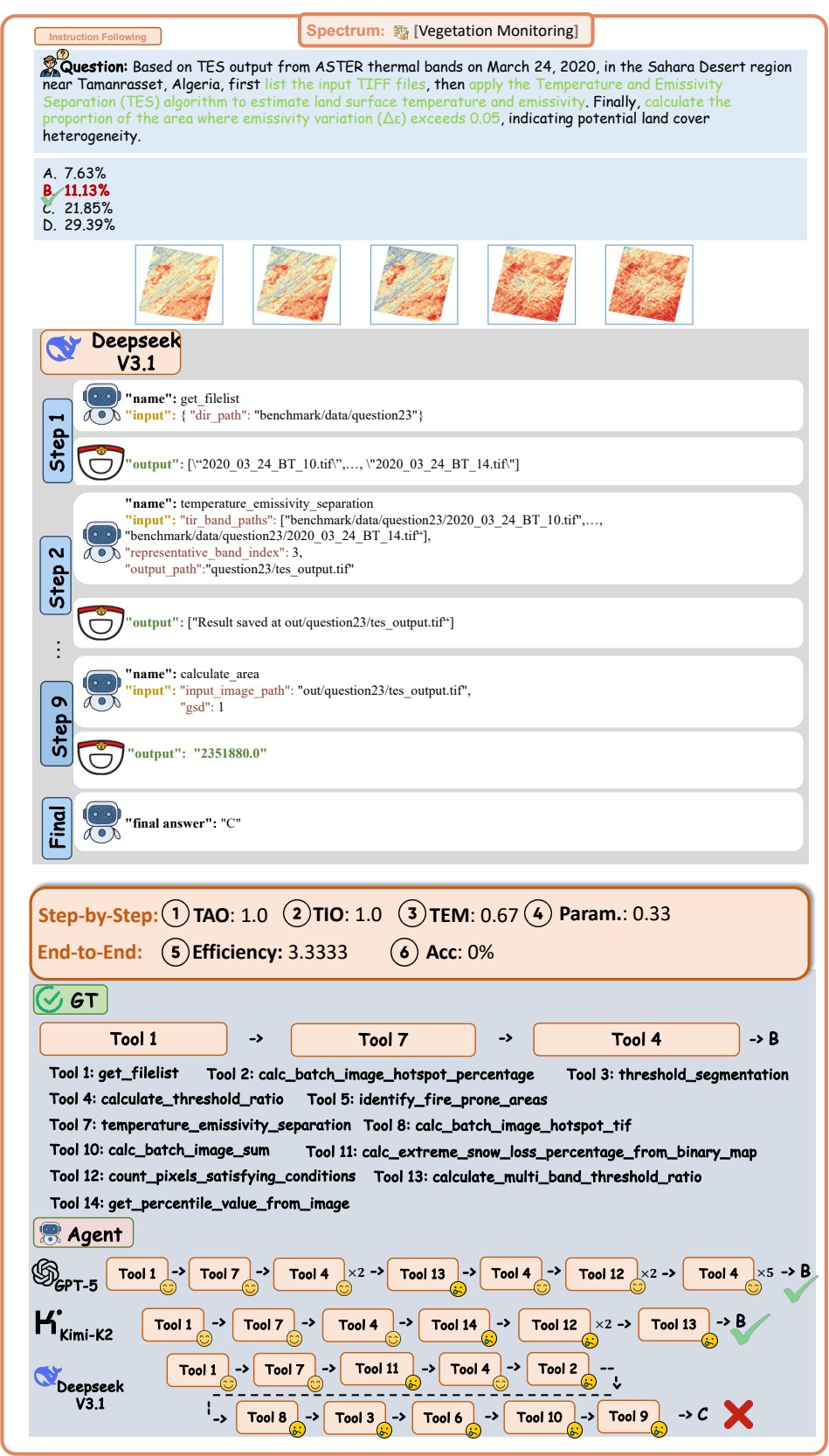

Figure 20: **Example of Vegetation Monitoring with Spectrum Data under the Instruction-Following Regime.**

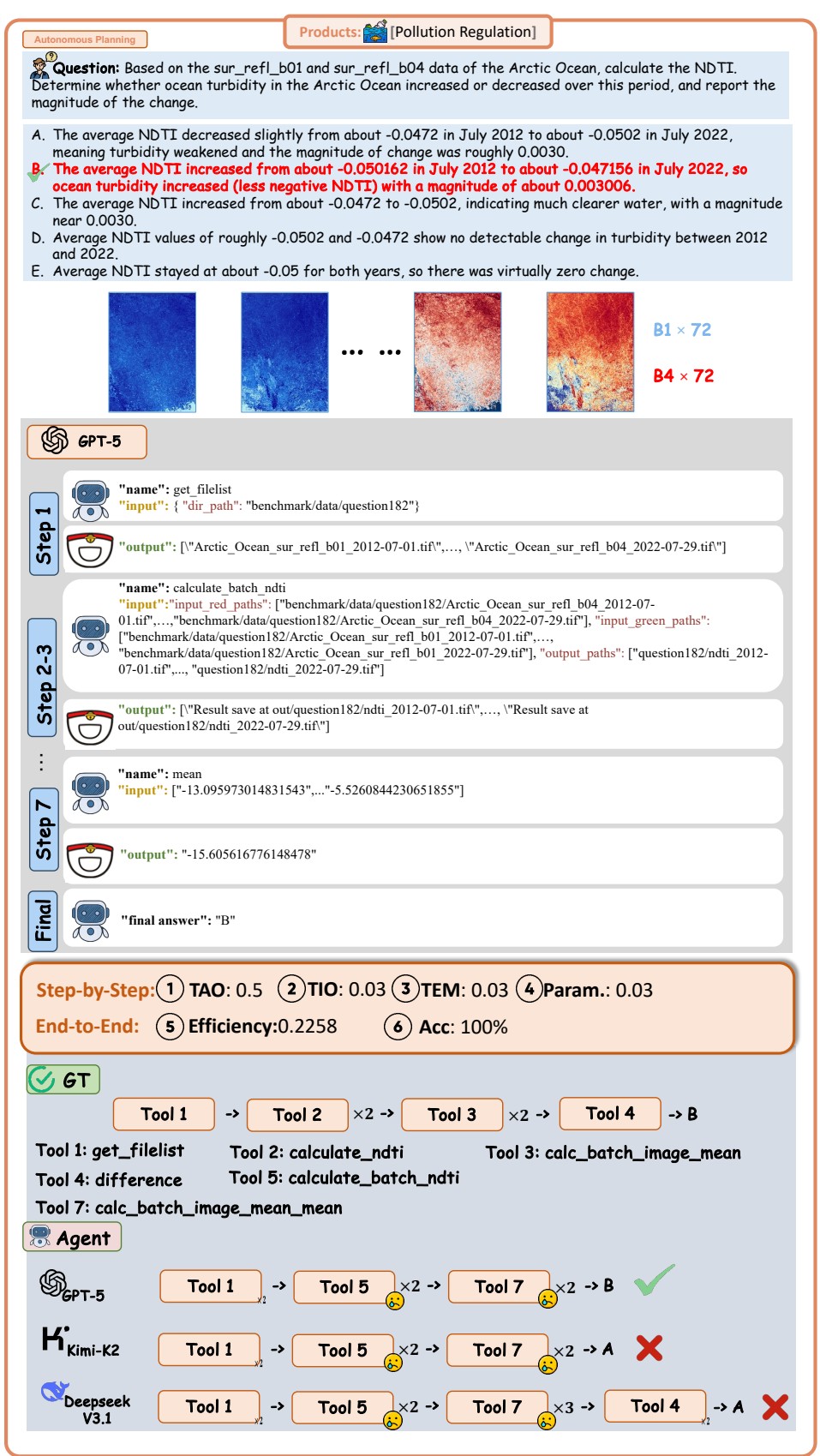

Figure 21: **Example of Pollution Regulation with Products Data under the Auto-Planning Regime.**

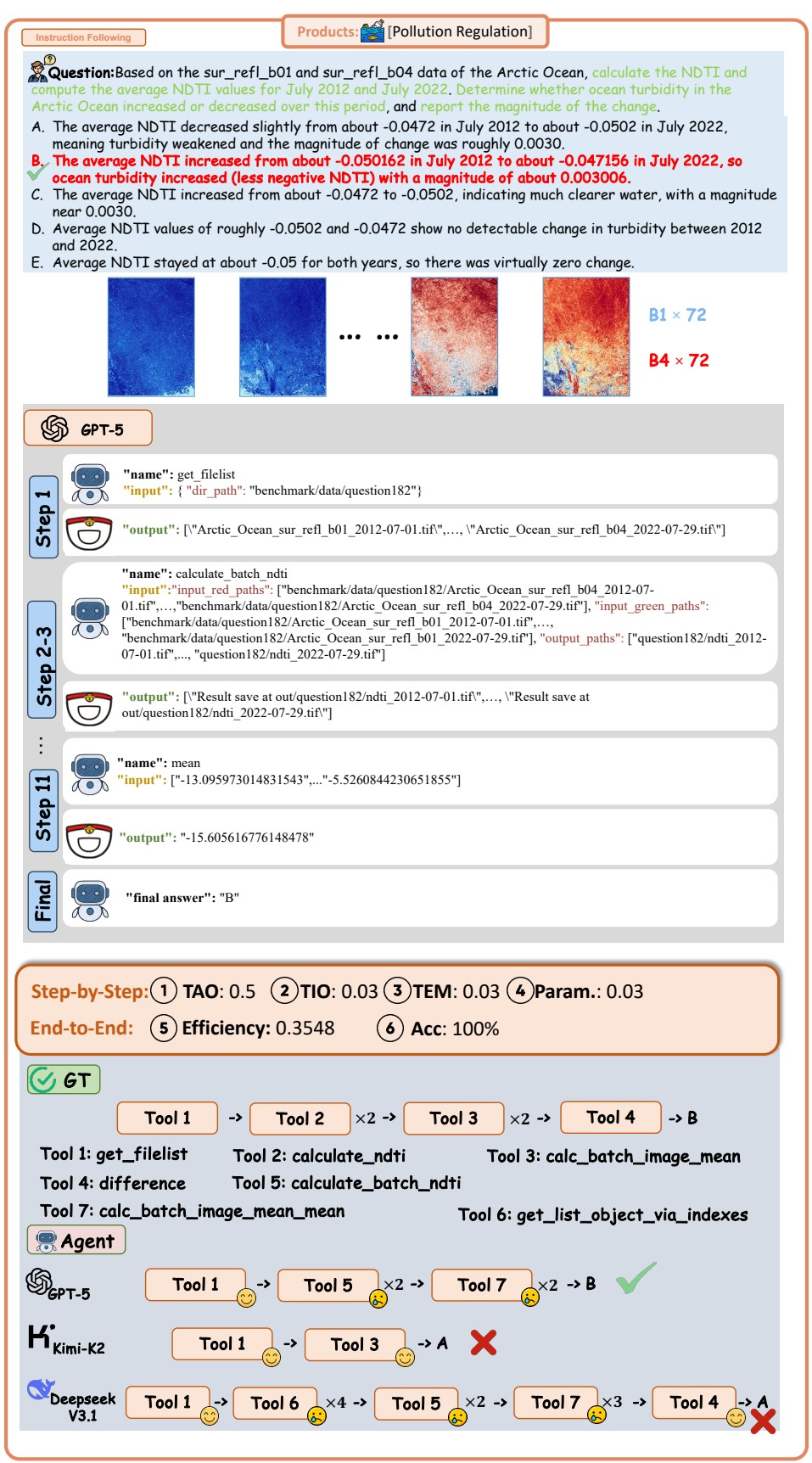

Figure 22: **Example of Pollution Regulation with Products Data under the Instruction-Following Regime.**

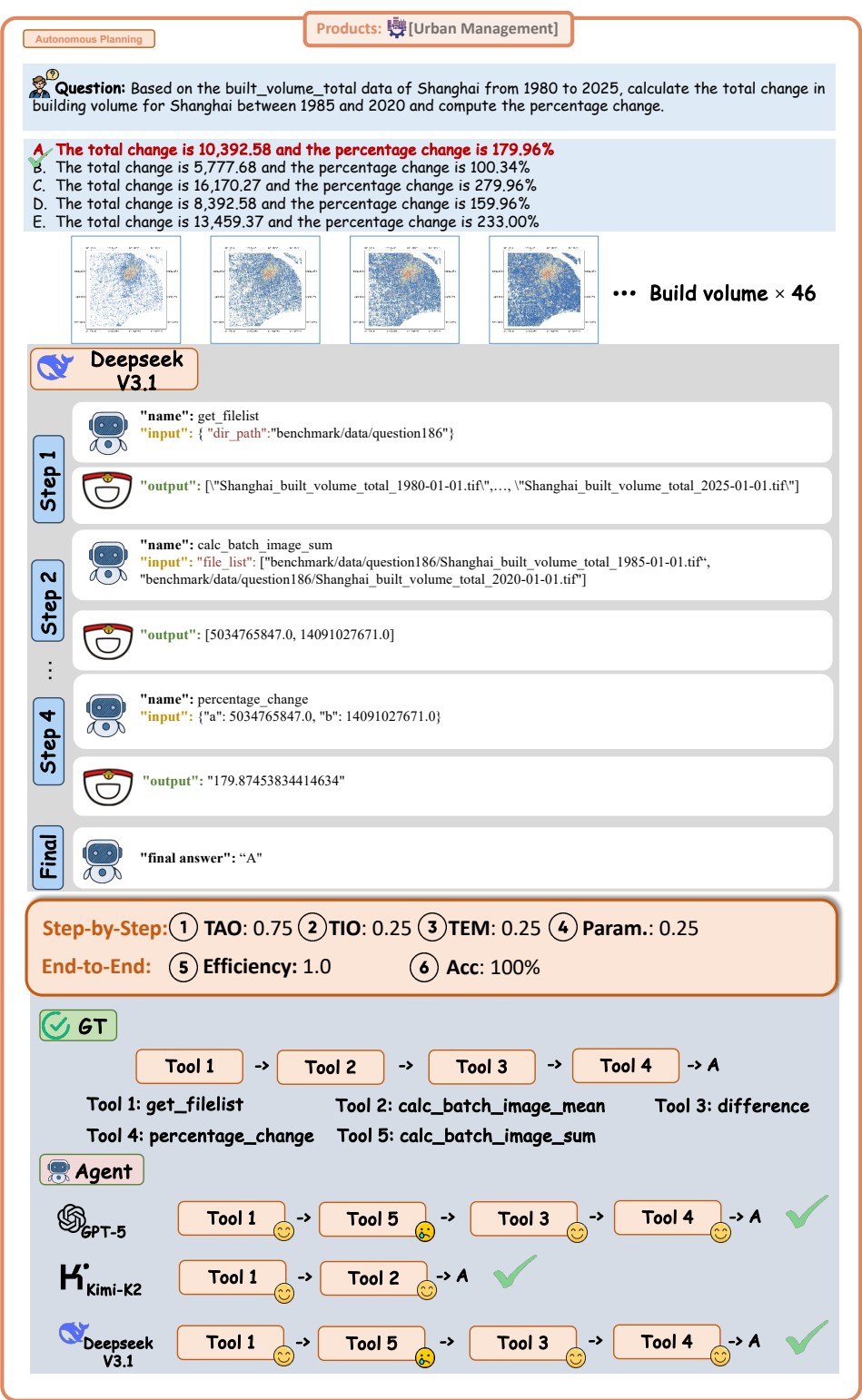

Figure 23: **Example of Urban Management with Products Data under the Auto-Planning Regime.**

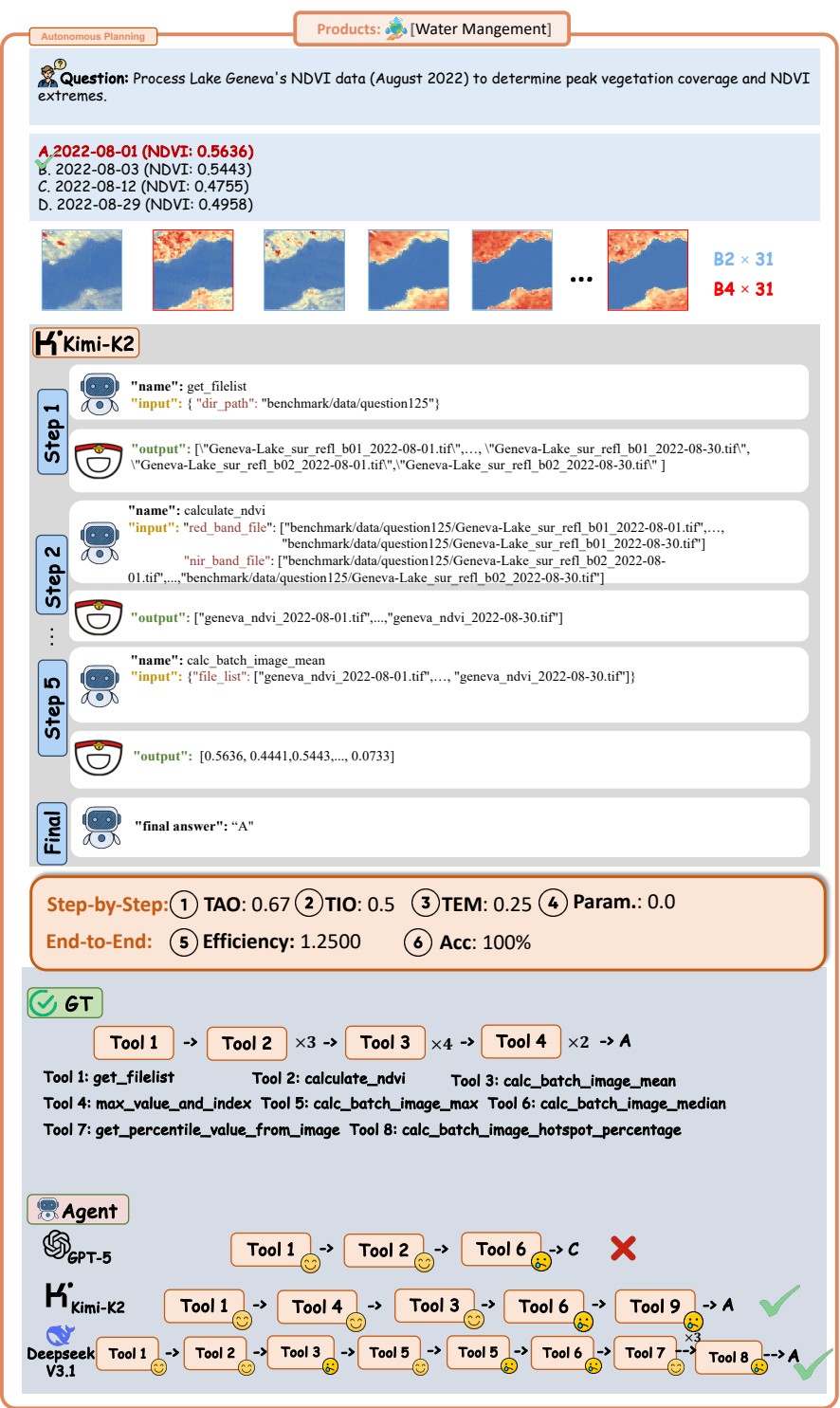

Figure 24: **Example of Water Management with Products Data under the Auto-Planning Regime.**

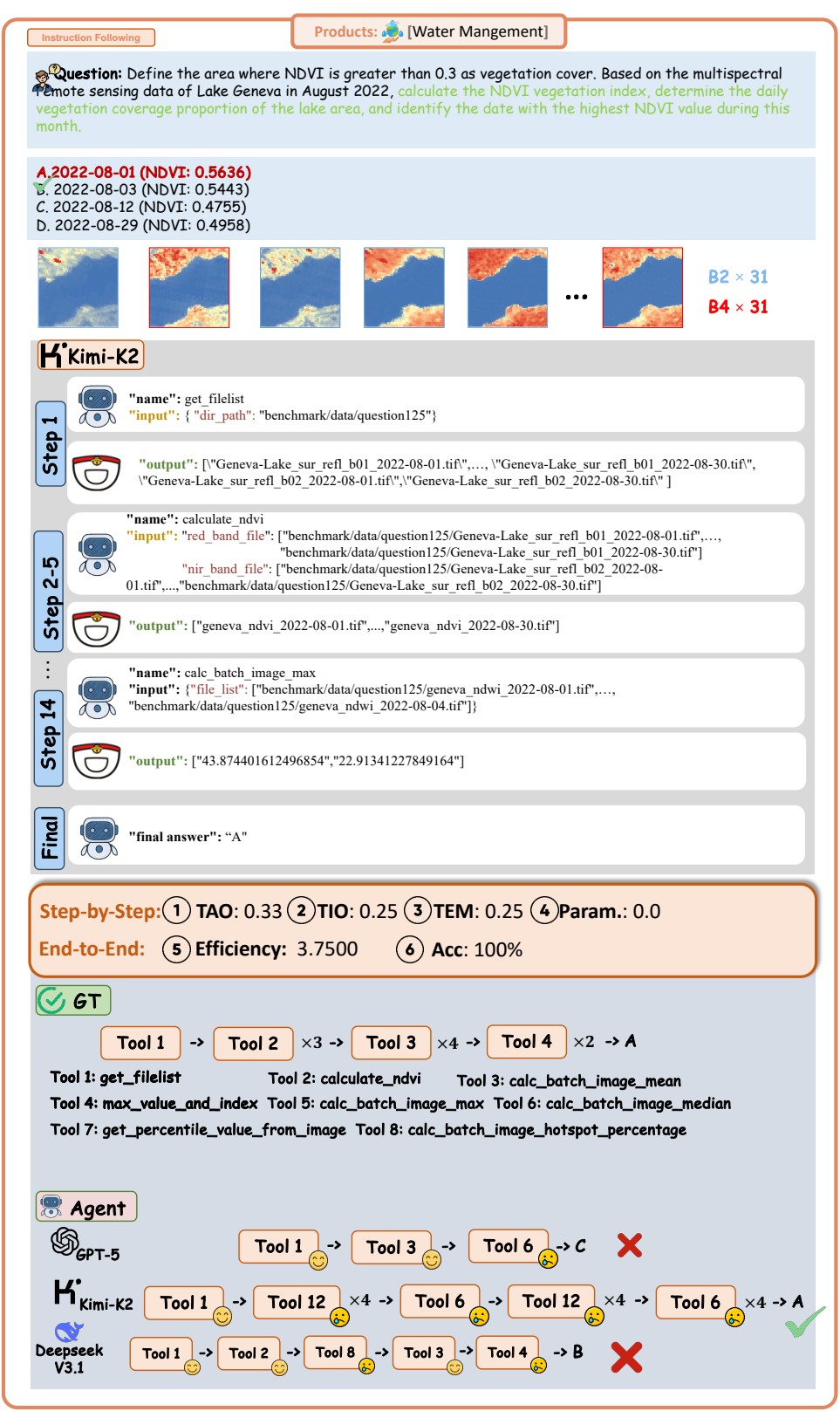

Figure 25: **Example of Water Management with Products Data under the Instruction-Following Regime.**

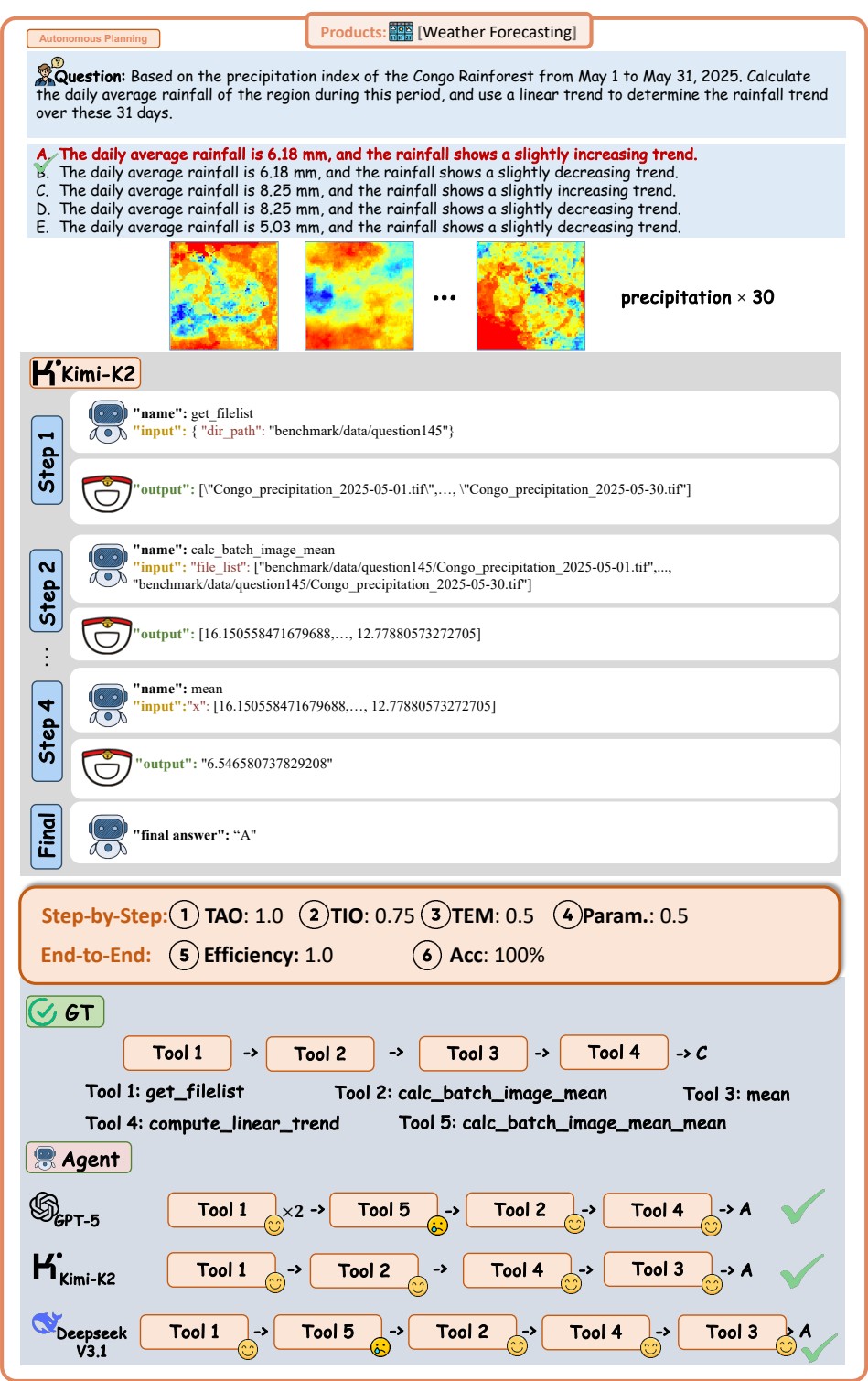

Figure 26: **Example of Weather Management with Products Data under the Auto-Planning Regime.**

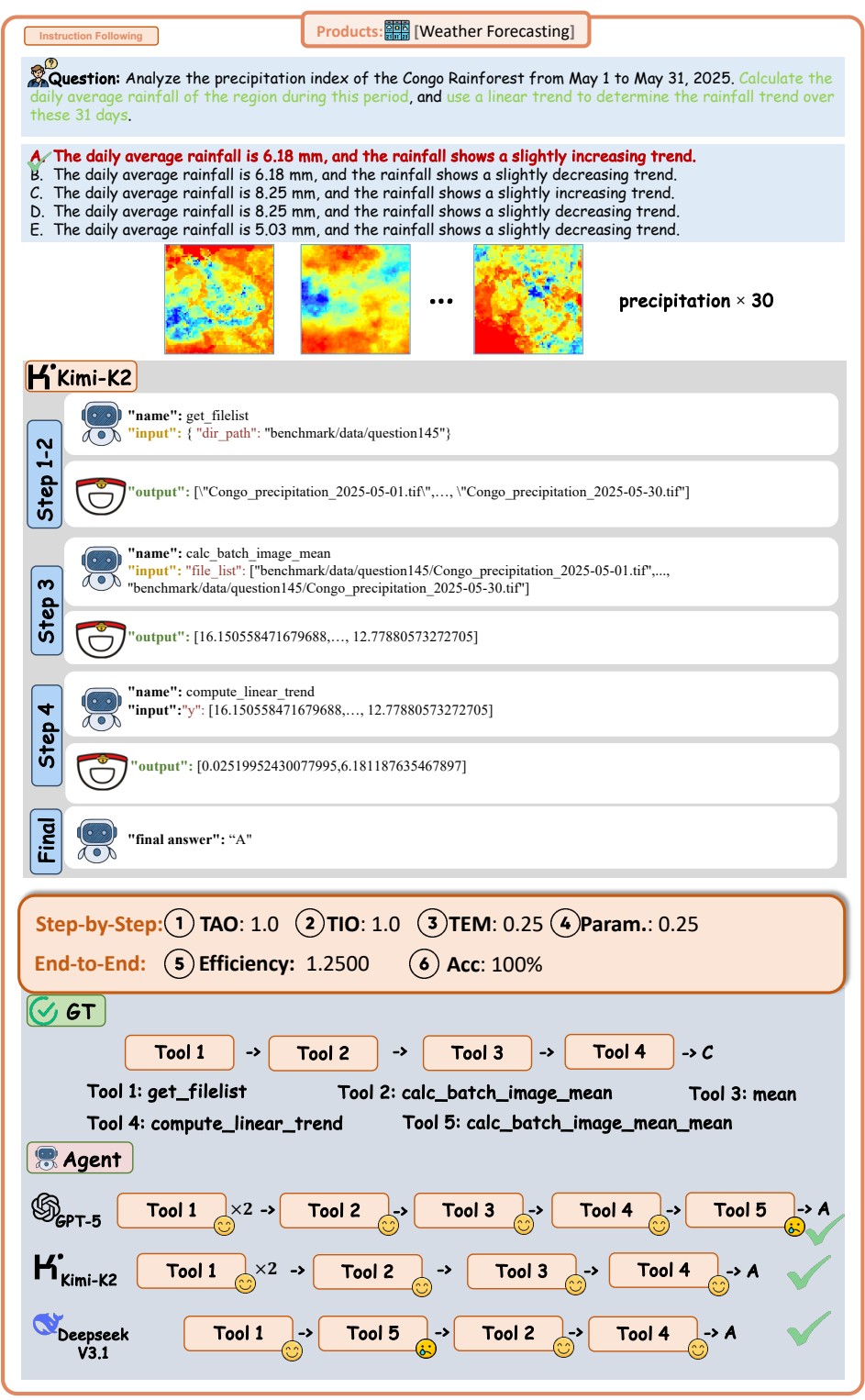

Figure 27: **Example of Weather Management with Products Data under the Instruction-Following Regime.**

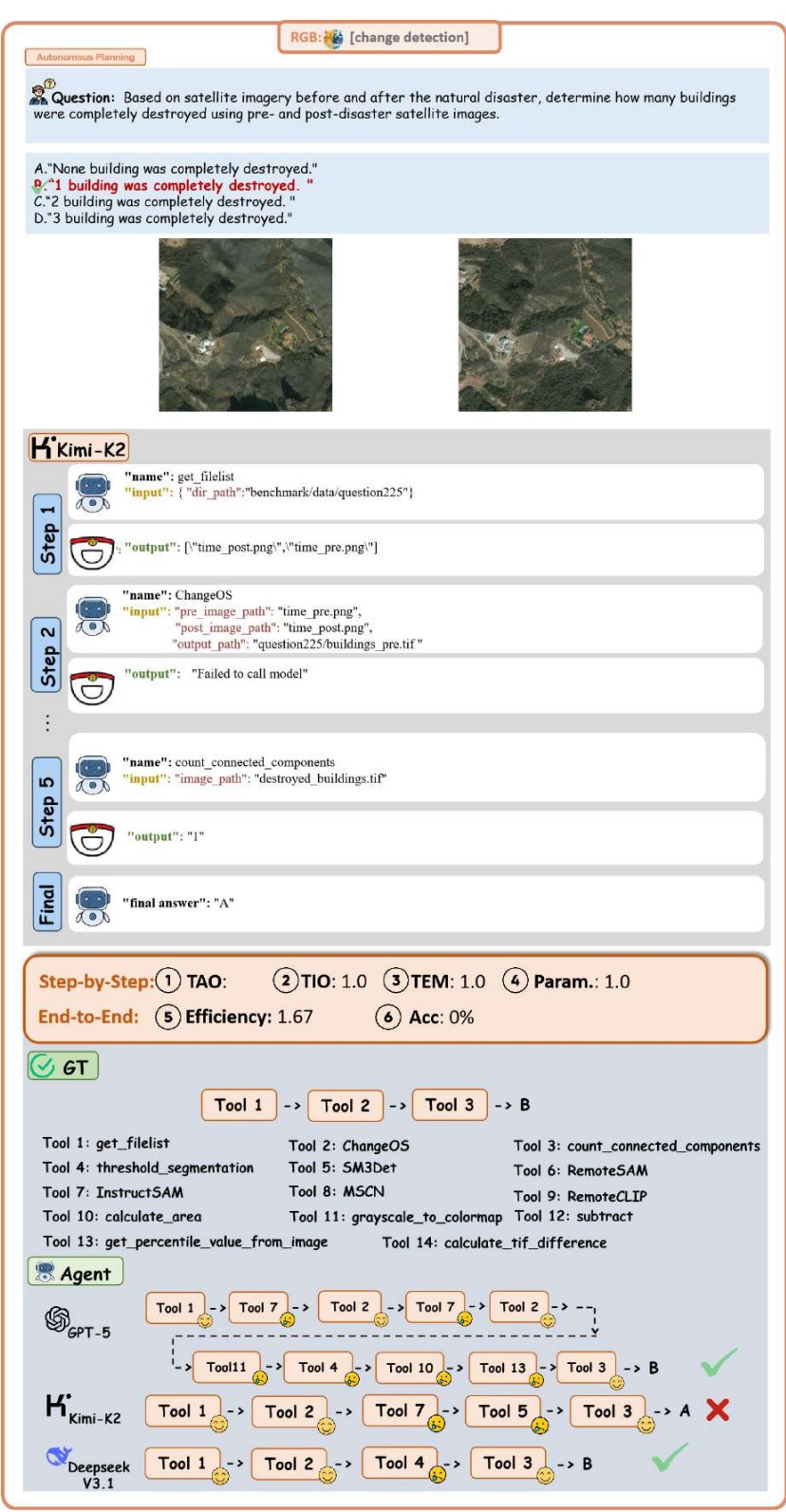

Figure 28: **Example of Change Detection with RGB Data under the Auto-Planning Regime.**

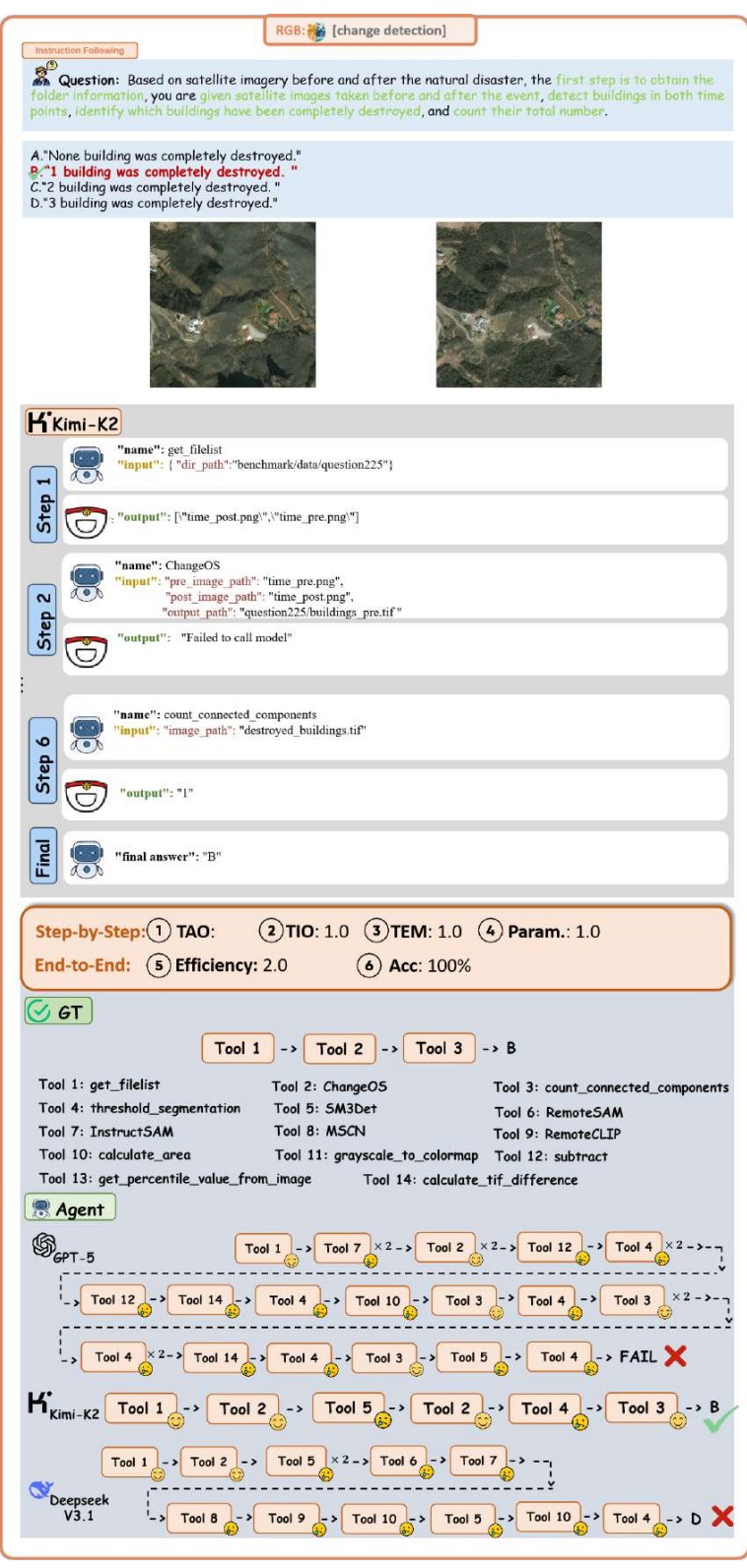

Figure 29: **Example of Change Detection with RGB Data under the Instruction-Following Regime.**

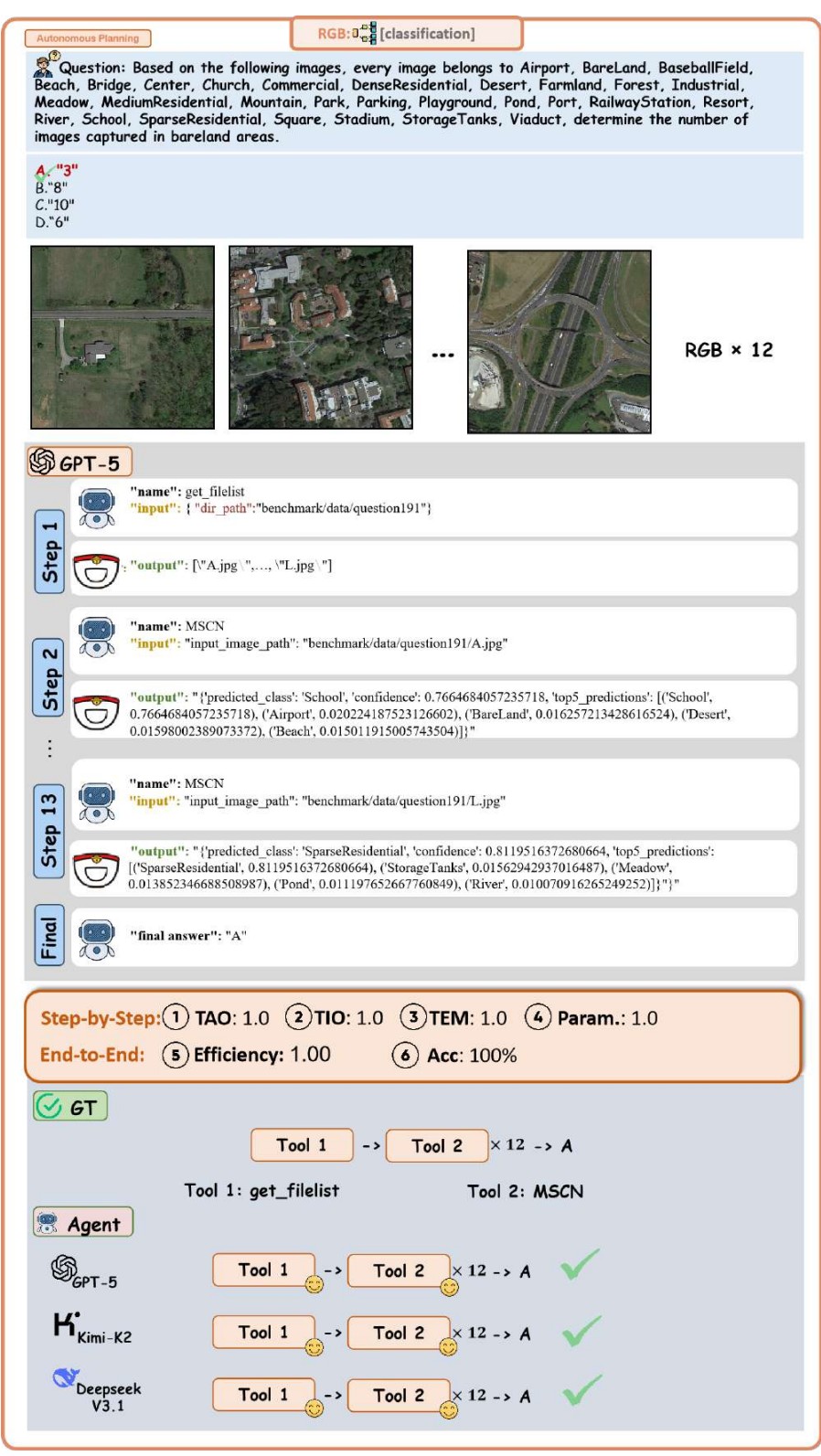

Figure 30: **Example of Classification with RGB Data under the Auto-Planning Regime.**

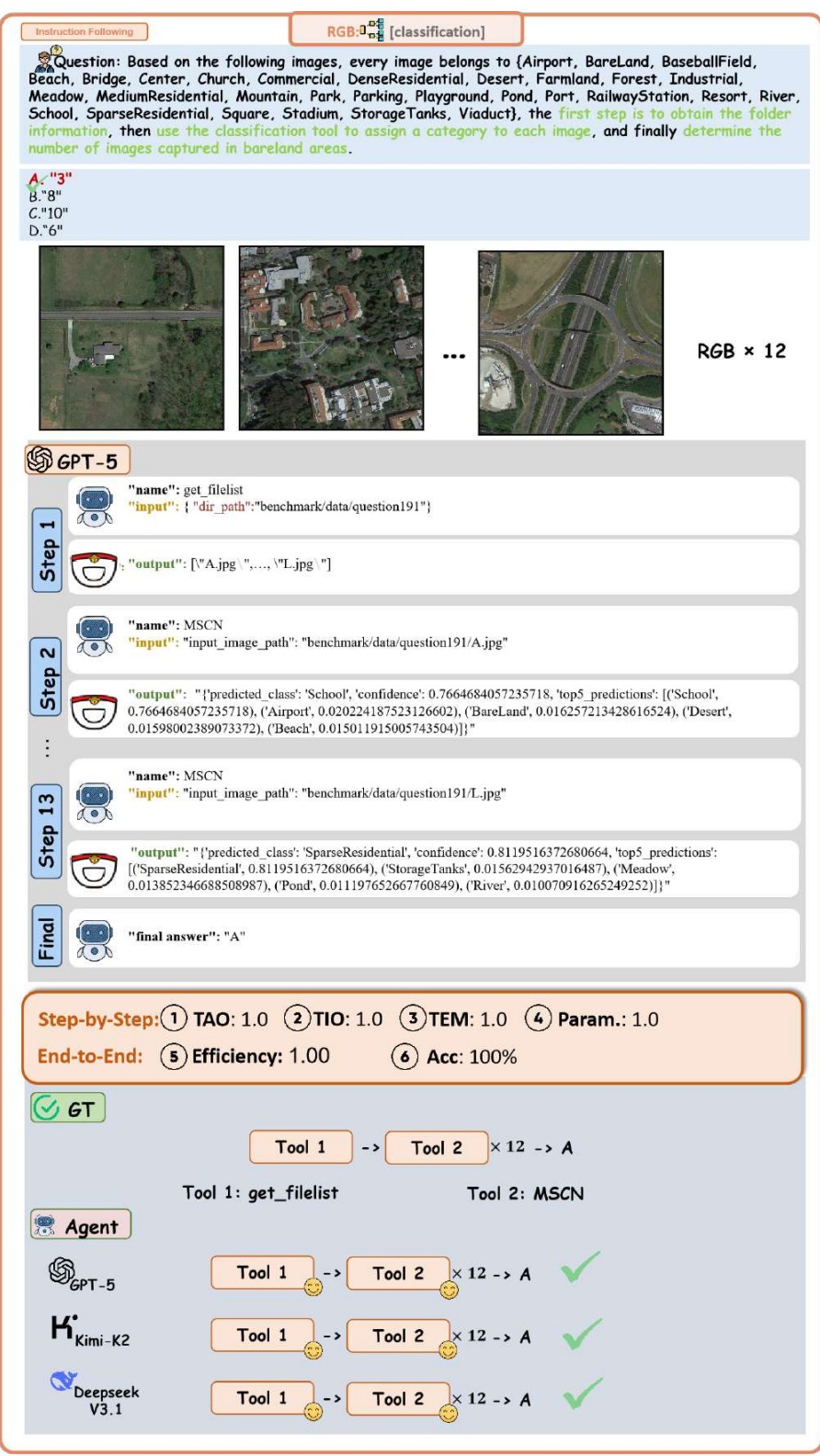

Figure 31: **Example of Classification with RGB Data under the Instruction-Following Regime.**

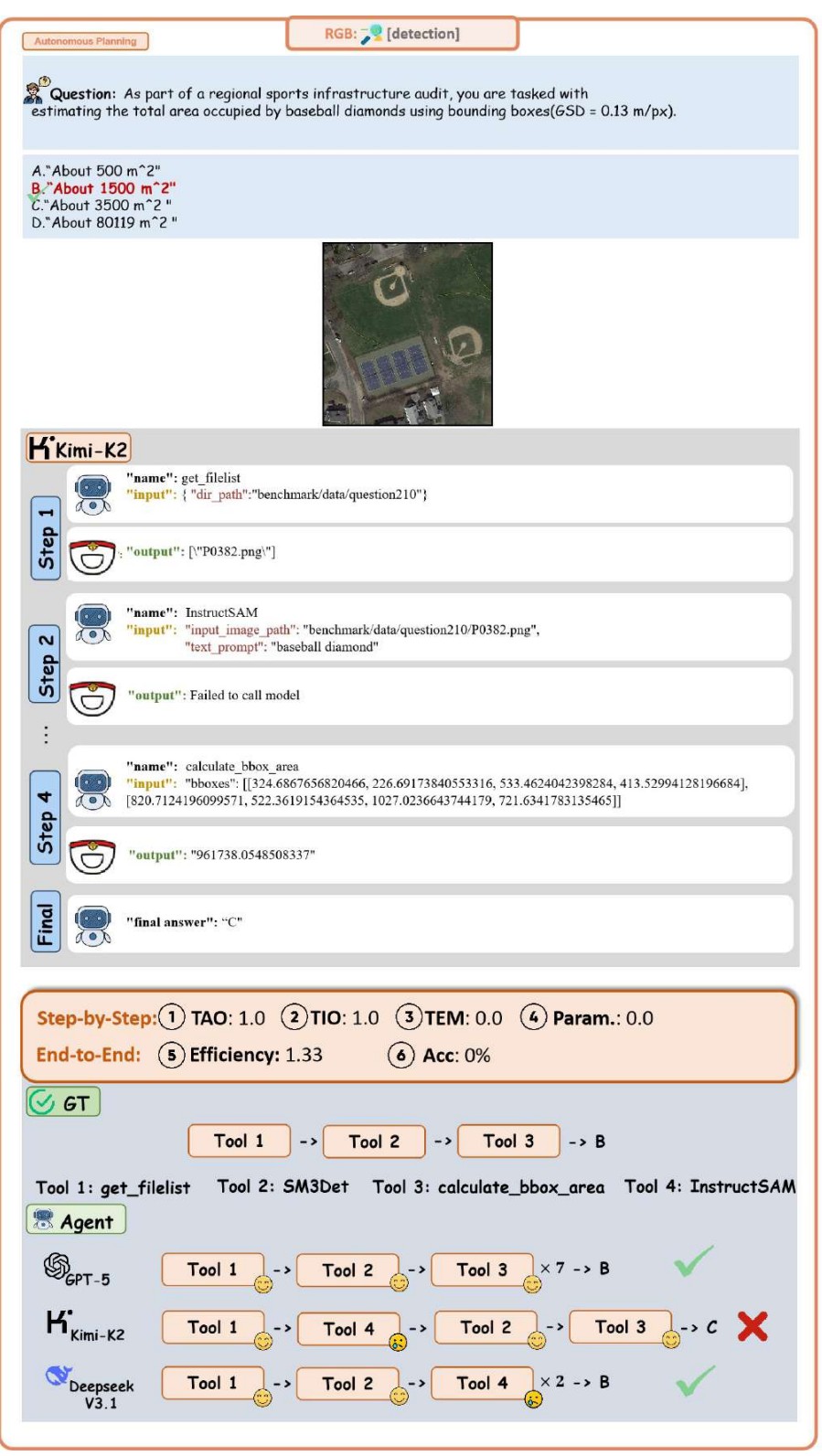

Figure 32: **Example of Detection with RGB Data under the Auto-Planning Regime.**

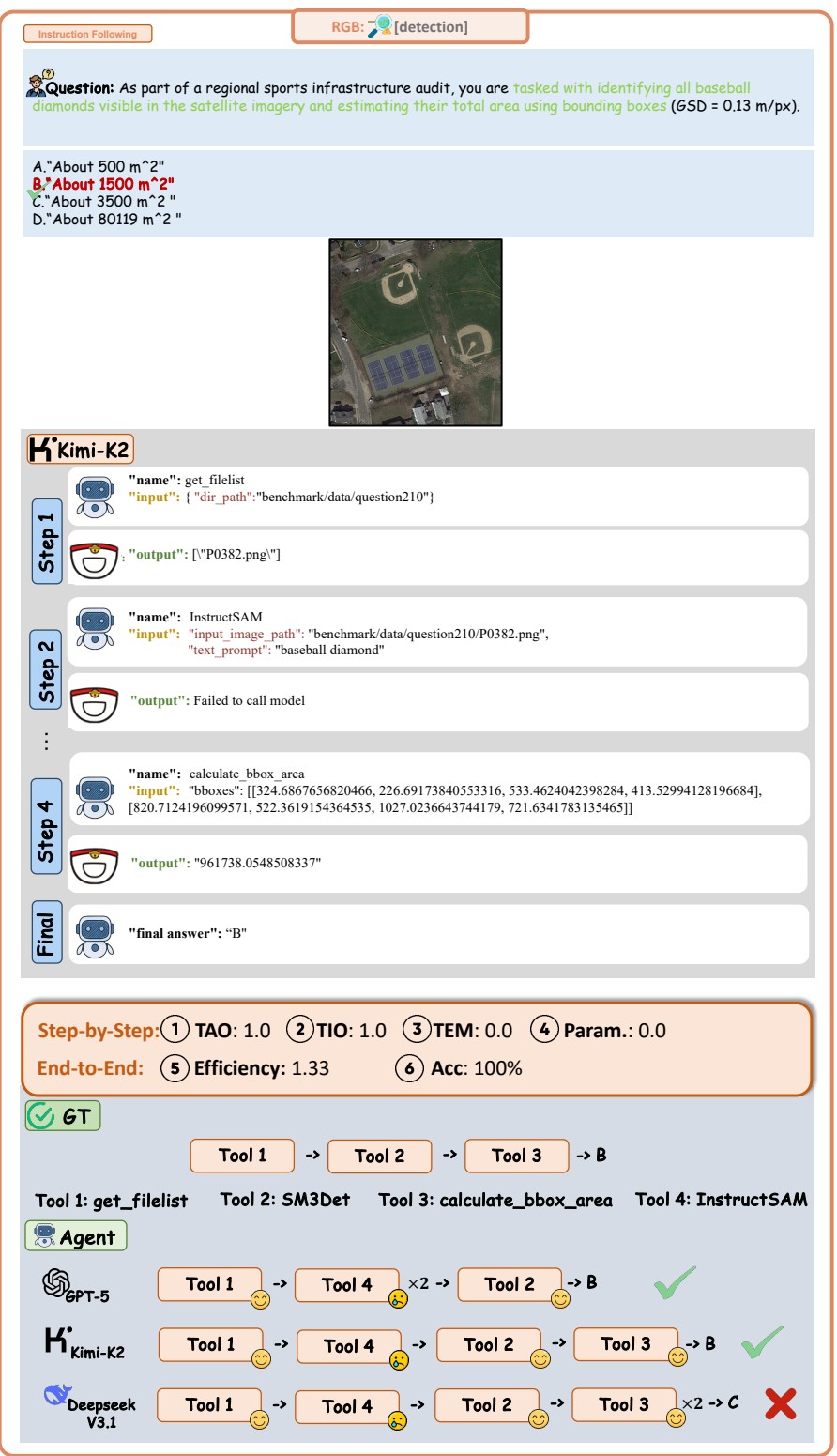

Figure 33: **Example of Detection with RGB Data under the Instruction-Following Regime.**

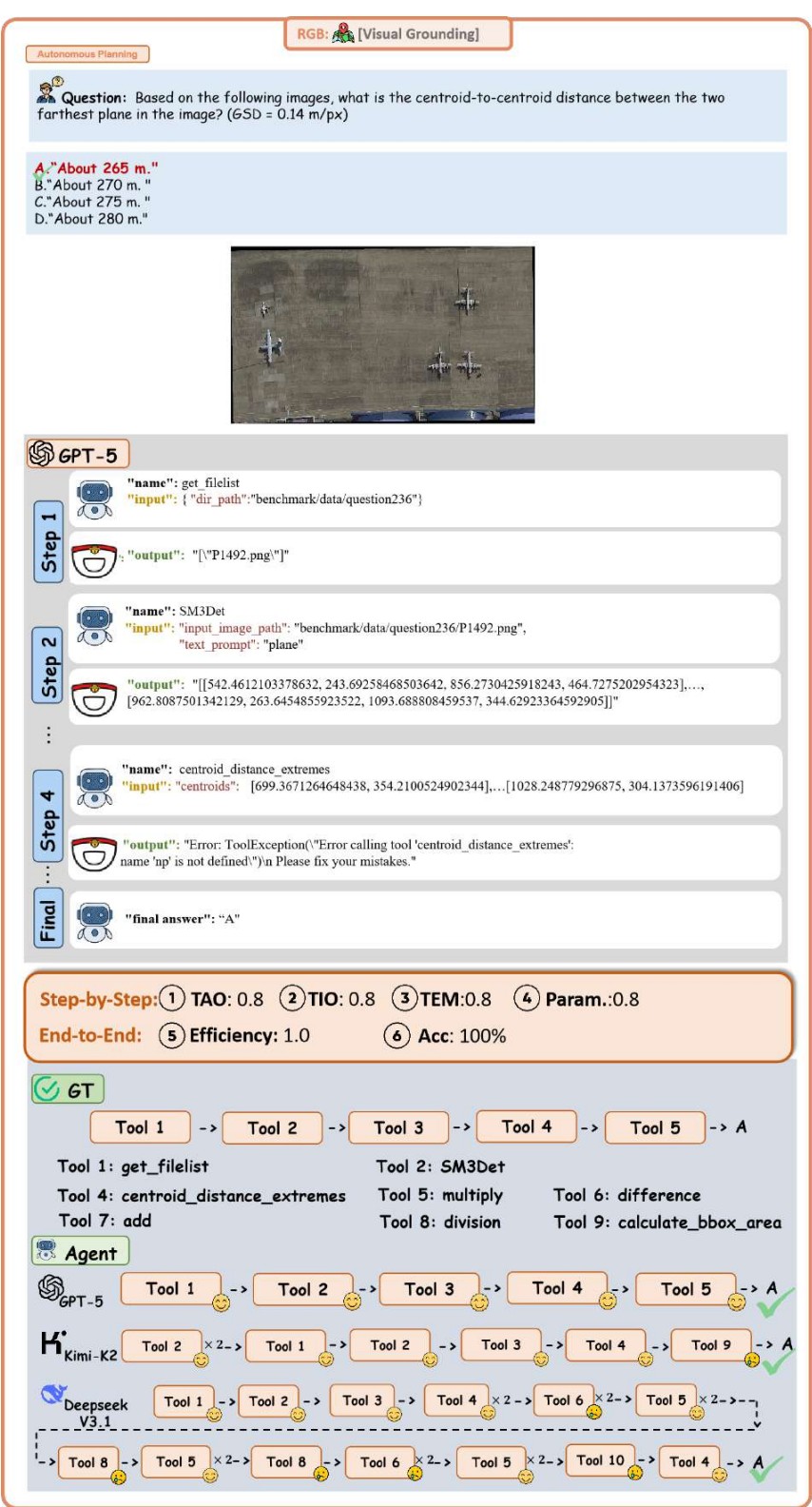

Figure 34: **Example of Visual Grounding with RGB Data under the Auto-Planning Regime.**

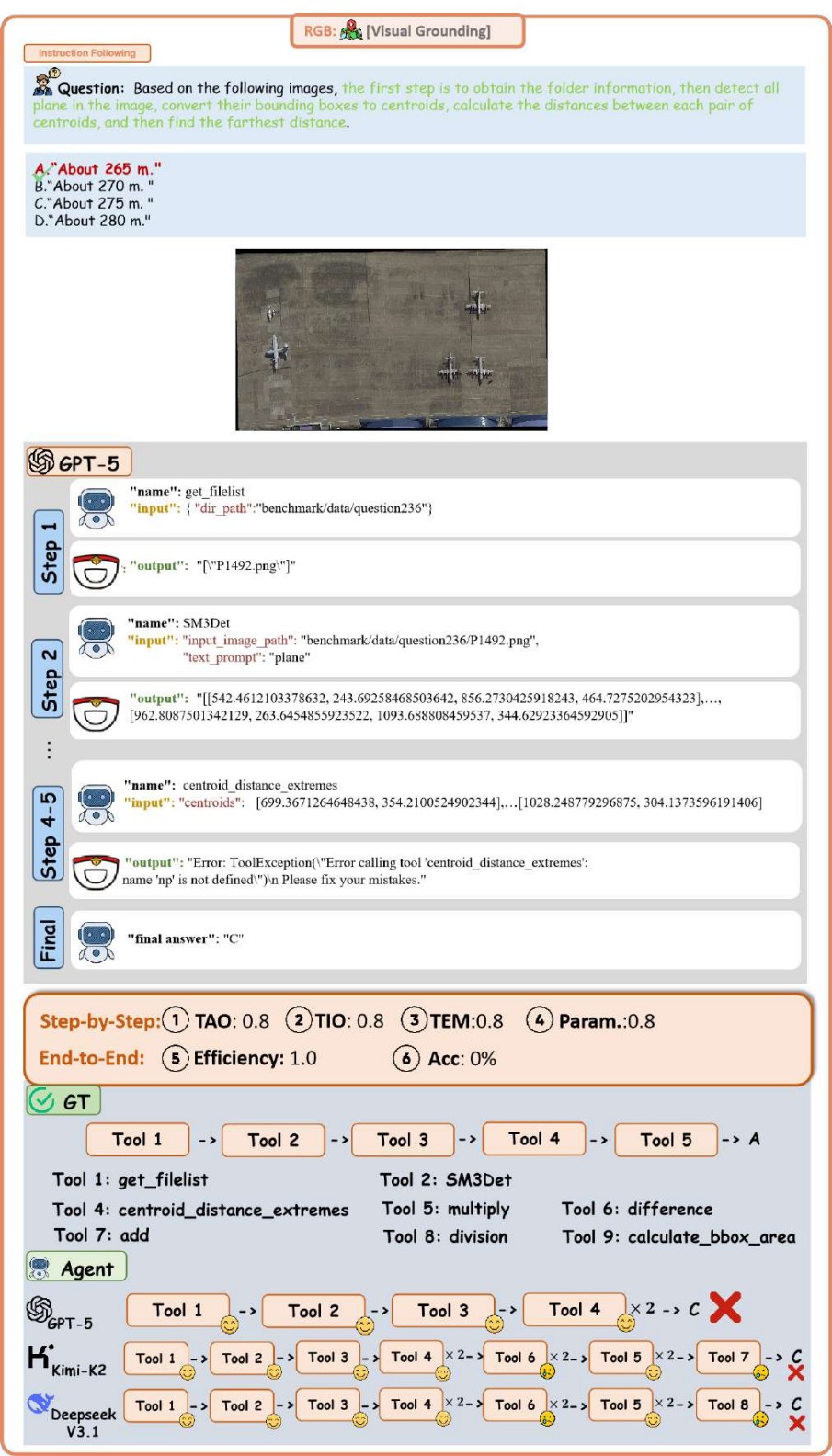

Figure 35: **Example of Visual Grounding with RGB Data under the Instruction-Following Regime.**

# I  CASE STUDY: COMPARE WITH OTHER AGENTS

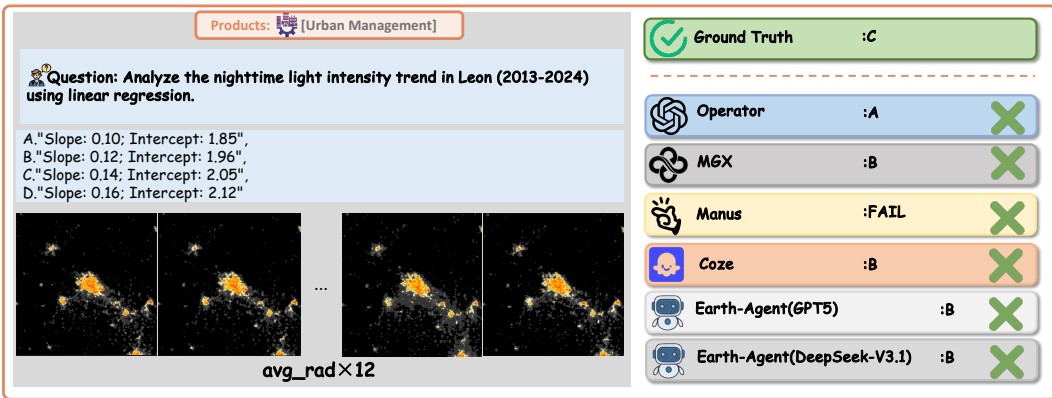

Figure 36: **A Question Case of the Urban Management Task using Products Data with Responses from Different Agent.**

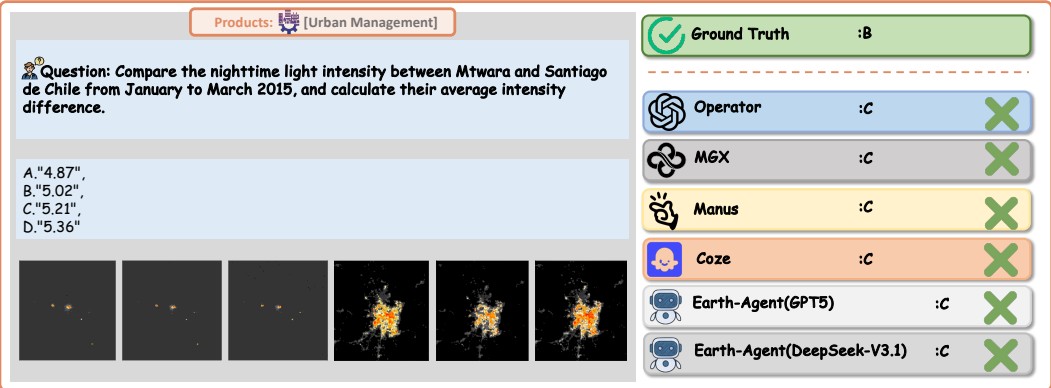

Figure 37: **A Question Case of the Urban Management Task using Products Data with Responses from Different Agent.**

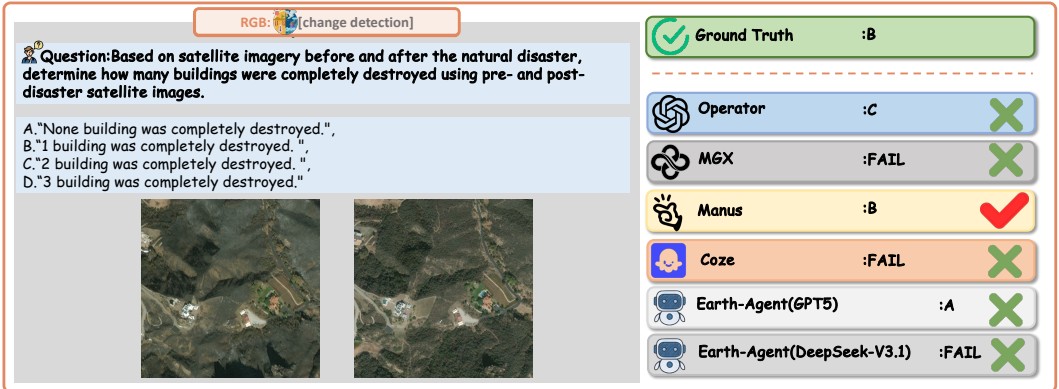

Figure 38: **A Question Case of the Change Detection Task using RGB Data with Responses from Different Agent.**

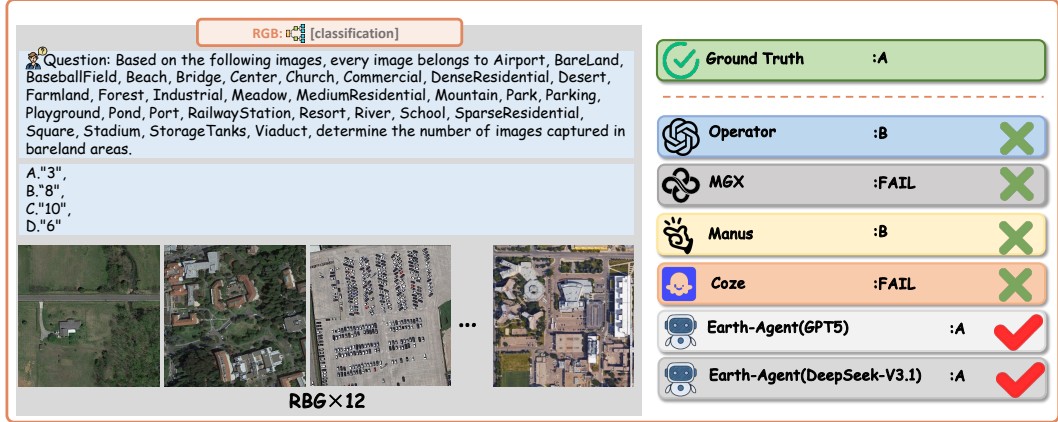

Figure 39: **A Question Case of the Classification Task using RGB Data with Responses from Different Agent.**

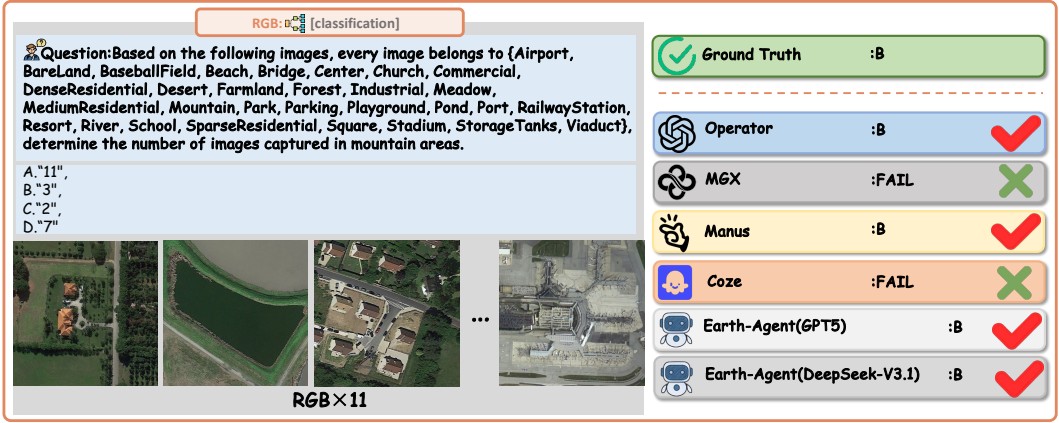

Figure 40: **A Question Case of the Classification Task using RGB Data with Responses from Different Agent.**

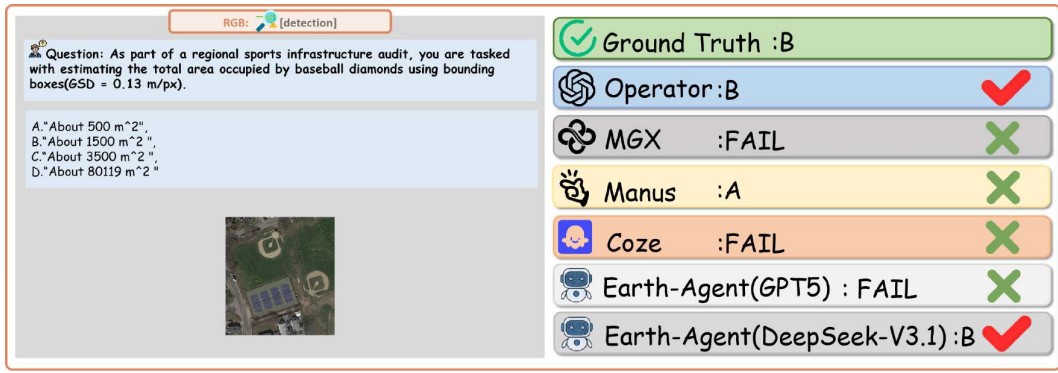

Figure 41: **A Question Case of the Detection Task using RGB Data with Responses from Different Agent.**

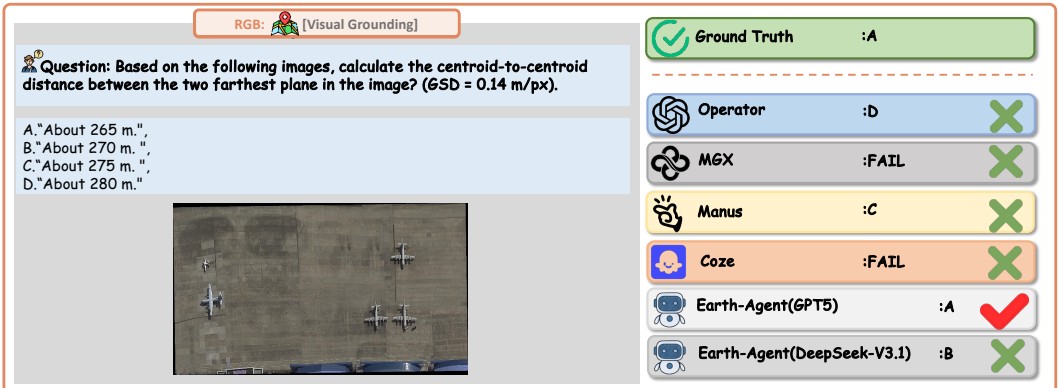

Figure 42: **A Question Case of the Visual Grounding Task using RGB Data with Responses from Different Agent.**

