# OpenReview forum: "Earth-Agent: Unlocking the Full Landscape of Earth Observation with Agents"
_ICLR.cc/2026/Conference — ICLR 2026 Poster_

### Official Review · Reviewer_38ix · 2025-10-28

**Soundness:** 3
**Presentation:** 3
**Contribution:** 3
**Rating:** 6
**Confidence:** 3

**Summary:**

Earth-Agent extends multimodal LLMs for Earth observation (EO) by enabling spectral data understanding, multi-image reasoning, and scientific tool use. It employs a ReAct-style LLM agent that dynamically invokes 104 EO expert tools via a Model-Context Protocol for spatiotemporal analysis and workflow planning. The authors introduce Earth-Bench, a 248-task benchmark (13,729 images) spanning RGB, spectral, and geospatial data, evaluating both final accuracy and reasoning quality. Experiments using backbones like GPT-5, Kimi K2, and DeepSeek-V3.1 show Earth-Agent dramatically outperforms prior MLLMs and generic agents on EO reasoning and traditional benchmarks, establishing a new agentic paradigm for scientific Earth analysis.

**Strengths:**

- Earth-Agent enables multimodal, multi-step EO reasoning by dynamically invoking 100+ expert tools, which is far beyond prior static MLLMs.
- Earth-Bench with 248 tasks and 13k+ images rigorously evaluates both reasoning and results, setting a new standard for EO agents.
- Consistently outperforms general agents and prior MLLMs across modalities and EO benchmarks.
- Extensive comparisons and ablations across LLM backbones validate tool-use effectiveness. Also identify and discuss failure modes like tool/file hallucination, offering clear improvement directions.

**Weaknesses:**

- This zero-shot agent sometimes makes mistakes that a more specialized model might avoid, like tool hallucination and file hallucination. These errors suggest the LLM doesn’t fully understand the tool API or environment state at times. A weakness of the paper is that it does not attempt any intermediate fine-tuning or training to address these issues.
- The approach relies on a single “expert” solution path for each task. It’s unclear how the evaluation handles an agent that finds a correct answer via a different valid approach. The authors do measure partial matches, which helps, but the paper doesn’t fully discuss if an agent could be penalized for creative but correct reasoning that deviates from the expert trajectory. This could be a minor weakness if the evaluation isn’t robust to alternative solutions. Clarification would be helpful to ensure the benchmark is fair to different reasoning styles.

**Questions:**

- The evaluation uses expert-curated solution trajectories for step-by-step scoring. How does Earth-Bench handle cases where the agent’s reasoning differs from the expert’s? For instance, if an agent uses a slightly different sequence of tools or a valid alternative formula to reach the correct answer, do the metrics give it credit? Is the benchmark tolerant to creative solutions? A discussion on how variability in reasoning is treated would be useful for users of Earth-Bench.
- The identified tool and file hallucinations indicate the agent sometimes loses grounding in the environment. What strategies do the authors think could reduce these errors? For example, would fine-tuning the LLM on a corpus of tool-use transcripts or adding a verification step where the agent double-checks the existence of a tool/file before calling it help? During development, did the authors try any prompting techniques to address this?
- The domain of tool-using LLM agents is evolving (e.g., SciToolAgent for scientific workflows, etc.). Could the authors contrast Earth-Agent with such systems? Is the main difference primarily the focus on Earth science modalities, or are there methodological innovations that set it apart?

---

> ### Author Response · Authors · 2025-11-21
> **Response to Weekness 1 & Question 2:**
>
> ### **Strategies to Reduce Agent Detachment from the Environment**
> Thank you for your thoughtful question. The tool and file hallucinations indeed highlight a fundamental challenge in grounding the agent in the interaction with environment. Specifically:
>
> - **File hallucination**: The agent assumes a file exists (e.g., NDVI_average.tif) without verifying the file folder, leading to environment hallucination.
> - **Tool hallucination**:The agent assumes a tool exists (e.g., calculating_snow_coverage) without verifying the toolkit, leading to tool hallucination.
>
> These problems suggest that the agent tends to rely on pretrained knowledge to fill in missing information, instead of actively using tools to verify the current state of the environment. **We think the following training-free strategies can help reduce these errors**:
>
> 1. Add a verification step: As you suggested, introduce a validation step before the model calls any tool or file to ensure their existence, preventing incorrect calls.
>
> 2. Design a tool router that stores and displays the tool and preceding decision steps using a latent variable probabilistic model.
>
> 3. Agent Memory: Earth science analysis involves multi-step reasoning with numerous files, making it challenging for the model to maintain memory through context alone. Introducing an external memory library can help the agent retain state when reasoning, enabling better interpretation of feedback from the environment.
> ___
>
> ### **Our Efforts to Solve This Issue**
> During development, we address tool and file hallucinations using the ReAct Mechanism (Observation → Act → Feedback). This allows the agent to observe feedback and adjust its actions accordingly. Additionally, we provide the agent with 3 attempts to correct tool or parameter errors, enabling it to fix tool and file hallucinations during the reasoning process.
>
> ___
>
> ### **Clarify about the Absence of Fine-Tuning Contribution**
>
> We appreciate your focus on the training methods for agents and the optimization prospect of Earth-Agent. However, we would like to clarify that the absence of intermediate fine-tuning or additional training is due to the motivation of this submission. As stated in line 40 of the paper, ***“Earth-Agent is the first agentic framework that unifies RGB and spectral EO data within an MCP-based tool ecosystem.”*** The core target of this work is to explore an alternative path, distinct from fine-tuning MLLMs, to test whether the Tool-Integrated Reasoning agent model can perform Earth science reasoning well. Thus, introducing "intermediate fine-tuning or training" as part of the method innovation would undermine the clarify of our true research motivation.
>
> As one of the first works to explore earth science using agent, we aim to clarify:
>
> - The feasibility of the agent method for Earth science tasks.
> - The existing challenges (e.g., the hallucination issues you observed).
>
> These findings are precisely the areas we hope to guide subsequent research to improve. We believe that this work serves as a "seed", and future efforts will build upon this precedent, systematically exploring how to construct agents with true domain reasoning capabilities through reinforced feedback and context-adaptive fine-tuning within the Tool-Integrated Reasoning framework we propose.

---

> ### Author Response · Authors · 2025-11-21
> **Response to Weekness 2 & Question 1:**
>
> Thank you for your insightful comment. In the current version of Earth-Bench, we use a rule-based trajectory evaluation mechanism to ensure reproducibility and quantifiability in the early stages of research. Specifically, we adopted the evaluation framework provided by **[langsmith-cookbook](https://github.com/langchain-ai/langsmith-cookbook/blob/main/testing-examples/agent-evals-with-langgraph/langgraph_sql_agent_eval.ipynb)** for agent process trajectories, which offers a detailed comparison of agent actions
> ___
>
> We assess tool usage through four key matching metrics:
> - **Tool-Any-Order**
> - **Tool-In-Order**
> - **Tool-Exact-Match**
> - **Parameters**
>
> In Tool-Exact-Match mode, model trajectories are compared to expert-curated solution trajectories, which serve as the "ground truth" for scoring. As the reviewer points out, the current evaluation mechanism does not account for variations in reasoning between the agent and the expert. In other words, if the agent employs a slightly different sequence of tools or uses a valid alternative formula to reach the correct answer, it will not receive credit unless the sequence and tools match exactly. However, ***Tool-Any-Order and Tool-In-Order offer a degree of tolerance in the evaluation process***. Tool-Any-Order allows the model to introduce irrelevant steps, as long as the steps included match the ground truth (GT) steps, it can still earn a score. Tool-In-Order allows the model to introduce irrelevant steps, as long as the steps are executed in the same order as the GT sequence, it can still earn a score. **You can refer to Figure 3 Dual-level Evaluation on page 5 for a detailed explanation.**
>
> In addition, we have indeed considered the issue about valid alternative formula you propose. For example, in question 144:
>
> ```shell
> Question: Based on the rainfall data of Cairns and Cooktown from January 31 to February 4, 2025, calculate the average unit area rainfall of Cairns and Cooktown during this period, and give the difference between the two.
>
> A. 22.03 mm
> B. 28.14 mm
> C. 32.03 mm
> D. 41.99 mm
> ```
>
> | Step | Expert Annotation                  | GPT Response                         |
> |------|-------------------------------------|--------------------------------------|
> | step1| get_filelist                        | get_filelist                         |
> | step2| calc_batch_image_mean               | calculate_tif_average                |
> | step3| calc_batch_image_mean               | calculate_tif_average                |
> | step4| mean                                | calc_batch_image_mean                |
> | step5| mean                                | calc_batch_image_mean                |
> | step6| difference                          | difference                           |
> | **Answer** | 32.02553631->'C'                       | 32.02553->'C'               |
>
> Although the reasoning process is inconsistent, GPT arrived at the correct answer. Initially, we considered assigning scores based on the results. However, we read a paper, ***LLM Reasoners***[1], which claims that LLMs might generate **false positive reasoning chains**. We are concerned about this phenomenon, especially since the benchmark we set is for multiple-choice questions. To establish a strict and unified baseline in the early version of the benchmark, we adhere to expert-curated trajectories as the "optimal path." This approach ensures comparability and stability across models. We acknowledge that this may limit flexibility in reasoning, but it provides a clear and measurable evaluation standard, which is crucial during the foundational phase of the benchmark.
>
> *References:*
>
> [1]Hao, S., Gu, Y., Luo, H., Liu, T., Shao, X., Wang, X., ... & Hu, Z. (2024). Llm reasoners: New evaluation, library, and analysis of step-by-step reasoning with large language models. arXiv preprint arXiv:2404.05221.

---

> ### Author Response · Authors · 2025-11-24
> **Response to Question 3:**
>
> SciToolAgent and Earth-Agent are two unprecedented systems from Agent4Sci, each focused on different domains. We will compare them specifically in terms of **Dataset** and **Method**.
>
> ---
>
> ## **Dataset Comparison**
>
> | Feature      | SciToolEval | Earth-Bench |
> |--------------|-------------|-------------|
> | Domain       | Physics, Chemistry, Materials Science | Earth Science |
> | Data Modality   | Pure Text | Spectrum, Products, RGB multimodal data |
> | File Operations | ×           | √           |
> | Complexity   | ≈ 2.05 steps | ≈ 5 steps   |
> | Task Types    | Molecular calculations | Spatiotemporal analysis |
>
> SciToolEval focuses on physics, chemistry, and materials science with pure text-based tasks, while Earth-Bench targets Earth Science, incorporating multimodal data formats like Spectrum, Products, and RGB. Earth-Bench requires dynamic tool calls to process geospatial data (e.g., Geotiff files). In SciToolEval, 80% of tasks can be answered in two steps with tool integration, whereas Earth-Bench involves multi-step workflows with multiple tools, resulting in higher task complexity.
>
> ---
>
> ## **Method Comparison**
>
> | Feature      | SciToolAgent | Earth-Agent |
> |--------------|--------------|-------------|
> | Architecture | Planner-Executor-Summarizer | ReAct       |
> | Tool Search  | SciToolKG    | FastMCP |
> | Evaluation Metrics | Exact Match + LLM as a Judge | Four different metrics |
>
> SciToolAgent uses a Planner-Executor-Summarizer architecture, relying on the SciToolKG for tool reasoning and execution. In contrast, Earth-Agent employs the ReAct reasoning architecture, integrating predefined toolkits, which are particularly useful for Earth Science tasks.
>
> SciToolAgent relies on a knowledge graph (SciToolKG) to manage tool relationships and dependencies, but it is static Logic reasoning and requires manual updates in advance, limiting scalability. In contrast, Earth-Agent uses the FastMCP server to automatically register tool functions, making it more flexible and scalable to extend with new tools.
>
> For evaluation metrics, SciToolAgent uses the LLM-as-a-judge when there’s no exact match between the output and ground truth, focusing on aspects like logicality and completeness. Earth-Agent, however, provides a more precise evaluation with four metrics: Tool-Any-Order, Tool-In-Order, Tool-Exact-Match, and Parameters, offering a more granular and analogous assessment of tool usage among different models.
>
> ---
>
> This comparison clearly highlights the differences and innovations between the two systems, particularly in terms of domain focus, data, and tool integration. I hope this helps address your question effectively.

---

### Official Review · Reviewer_ic4k · 2025-10-31

**Soundness:** 2
**Presentation:** 4
**Contribution:** 2
**Rating:** 4
**Confidence:** 3

**Summary:**

This paper proposes Earth-Bench, an expert-certified benchmark accompanied by an agentic framework for evaluation. It includes 248 questions, making it the largest Earth Observation (EO) benchmark to date in the LLM-agent domain. Using their Earth-Agent framework, GPT-5 successfully resolves around 60% of the questions, and the overall ranking aligns well with current expectations of LLM capabilities.

However, the paper lacks clarity regarding the dataset curation process, e.g., whether each question was reviewed by multiple experts. Additionally, several inconsistencies and issues within the benchmark data, as well as insufficient detail about the true number of unique questions (i.e., whether it is genuinely 248 questions or just 124 formulated in two ways), raise concerns about the benchmark’s reliability and robustness.

**Strengths:**

1. Overall Evaluation:
This paper explores a wide range of tools and evaluates how effectively current LLMs, acting as agents, can leverage these tools for question answering. The experiments are comprehensive, and the results are clearly presented and well-analyzed.

2. Contribution of Toolset:
The predefined set of 104 tools covers a broad spectrum of Earth Observation (EO) tasks and represents a valuable contribution toward advancing tool-augmented agents in the EO domain.

**Weaknesses:**

While this paper seems to propose a genetic framework and a comprehensive benchmark, I found several issues that compromise my ratings for this work. Specifically about the **rigor and the reliability of the dataset**. Below are two of the main weaknesses I observed

**1. Unclear data annotation pipeline**

The authors stated, “From these data sources, a team of domain experts ...,” but no further information is provided. For example, what is the background of these domain experts? Is there a cross-reviewing process? What happens if inconsistencies arise? Was an LLM used during the curation process? How did the experts formulate the questions?

Without a clear description of the question validation process, it is difficult to ensure that the benchmark is trustworthy and reliable.

**2. Questions about the benchmark dataset**

I reviewed some of the benchmark data provided, and found several questions.

**- Question Predictability:**
On Page 37, the question involves Landsat 8, which has a 16-day revisit cycle. Thus, between June and September, only about 8 +- 1 observations are available. Thus, even without reading the question carefully, one could guess any option less than 8, which again raises concerns about how informative the task is. (The options provided are (1) 5, (2) 10, (3) 12, and (4) 18 days. )

**- Incorrect Formula Range:**
On Page 43, the Normalized Difference Turbidity Index (NDTI) is reported with values ranging from −666 to 100. However, NDTI is typically computed as (red − green)/(red + green), meaning its range should be [−1, 1]. For example, Option B claims: “The average NDTI increased from −666.47 to −61.17,” which is mathematically implausible and impossible. Feel free to correct me if the authors use different metrics.

**- Methodological Justification:**
On Page 51, the method count_skeleton_contours is used to estimate the number of houses, but no reference or justification is provided. A citation or methodological rationale is necessary to validate this choice.

**- Question Design Issues:**
For example, on Page 53 (Line 2808), only four images are provided, but the answer choices are (A) 3, (B) 8, (C) 10, (D) 16). It seems trivial to select “A” without even understanding the question, which undermines the reliability of the benchmark. A similar issue appears on Page 63 (Line 3356).

**- Self-Conflicting Content:**
The questions on Pages 55 and 56 appear inconsistent. The questions ask about the total area. However, the first question asks about
sports infrastructure including both baseball and tennis courts, while the second focuses only on baseball courts, yet the answers are both 1500m*2, which is confusing to me. The two questions should have different options and the current version makes the intended reasoning unclear.

**- Ambiguity in Measurement Definition:**
On Page 65, the task asks for the distance between the two farthest planes in the image, but it is ambiguous how this distance is defined. By contrast, Page 58 explicitly specifies centroid-based distance, suggesting that the two questions target different notions but are treated as equivalent in the benchmark.

I understand that some technical details may have been omitted for brevity, and I would appreciate clarification from the authors regarding these issues and any potential misunderstandings.

**Questions:**

**Clarification of Anti-Planning and Instruction-Following**

Could the authors clarify whether all 248 questions each have both an auto-planning and an instruction-following version, or whether 124 are auto-planning and the other 124 are instruction-following? If it is the latter, then claiming to have 248 unique questions seems somewhat misleading, as there would actually be only 124 distinct questions expressed in two different formulations.

**Dataset Composition**

How many products or sensors are included in the dataset? So far, I can identify Landsat and MODIS (ASTER), and potentially some high-resolution imagery (Page 52) with unspecified sources. It would be helpful if the authors could describe the data sources, products, and collections in more detail, along with their distribution.

**Figure 7 Inconsistency**

In Figure 7, there appear to be more than 300 error counts, even though the dataset only contains 248 questions. Could the authors clarify how these counts are computed and why they exceed the total number of questions?

**Evaluation Without Tool Use**

How do the authors implement and evaluate model performance without tool use? For RGB-based questions, how do the models perceive or interpret the images if they are not allowed to invoke perception modules?

---

> ### Author Response · Authors · 2025-11-21
> **Response to Weakness 1:**
>
> Thank you for your comment. We appreciate the opportunity to clarify the question validation and data annotation process.
>
> ***We have added detailed explanations in the revised version: Appendix A.1 (page 17) for Dataset Composition and Appendix A.5 (page 17) for Dataset Quality Control.***
> ___
>
> ### **Data Annotation Process**
> Our annotation team consisted of **12 researchers** with expertise in remote sensing science, including **3 core authors** of the paper, who served as senior reviewers. These senior reviewers were responsible for defining the themes and creating template examples for the **Spectrum**, **Products**, and **RGB** categories, guiding the overall direction of the question pool.
>
> The remaining **9 graduate students and senior undergraduates** were tasked with the following:
> 1. **Initial question creation** (approximately 400 questions),
> 2. **Data collection**,
> 3. **Development of Python-based solution scripts**.
>
> Once the questions were created, they underwent thorough review by the three senior reviewers. The review process was based on two primary criteria:
> 1. **Data Integrity**: Ensuring that the raw data involved (e.g., Landsat or MODIS) contained complete band information within the task’s time range and had no anomalies. Any questions linked to data with missing or anomalous values were discarded.
> 2. **Task Difficulty Assessment**: The senior reviewers evaluated the difficulty of each question based on the number and complexity of functions defined in the Python solution. Simpler tasks, such as those involving a single NDVI calculation, were removed to maintain the overall challenge of the benchmark.
>
> ### **Cross-Reviewing Process**
> To ensure the reliability of the questions, we employed a **cross-reviewing process**. All questions were independently written by the research team and then cross-checked by the senior reviewers. If any inconsistencies were found, such as missing data, incorrect formulas, or issues with the question’s difficulty. These questions would be revised or removed. This helped ensure that all questions in the benchmark met the necessary standards for quality and consistency.
>
> ### **Use of Large Language Models (LLM)**
> During the curation process, **no LLMs** were used to generate the questions or solve the problems. The entire question creation and solution validation process was carried out manually by the research team to ensure scientific accuracy and quality.
>
> For **JSON file creation** related to tool calls, we used Cursor's Agent mode to assist the generation of the file structure. This helped avoid omissions or formatting errors. However, Cursor was only used for file generation and not for question formulation or reasoning. Afterward, all generated JSON content was reviewed and verified by the 3 senior reviewers.

---

> ### Author Response · Authors · 2025-11-21
> **Clarification about Benchmark**
>
> We sincerely appreciate your careful reading of our manuscript and the valuable feedback provided. We apologize for any sections that may have been unclear or frustrating. We have carefully examined each question your raised and provide clarifications and corrections one by one as follows:
>
> ___
>
> - **Question Predictability**: This question uses data from Landsat 8, with a total of 15 observations from June to September. You are correct that Landsat 8 has a 16-day revisit cycle. However, **the actual observation frequency is not solely determined by the revisit cycle**. In the case of the **Death Valley region**, the area lies in the overlap between two adjacent Landsat orbits (Path 40 and Path 41). As a result, the observation frequency for this region is an 8-day interval between observations, with Option A, B, and C as possible alternatives. In some questions, **we also provide irrelevant data** in the folder (such as three additional observations from May in this case). You can check the output of `get_filelist` at line 2229. Option D is included as a distractor.
>
> ___
> - **Incorrect Formula Range**: In the question on Page 43, we provided the agent with MODIS Terra Surface Reflectance data obtained from the GEE platform: [MODIS 061 MOD09A1](https://developers.google.com/earth-engine/datasets/catalog/MODIS_061_MOD09A1?hl=zh-cn#bands). This data included some **filling values** or anomalies caused by cloud, shadow, and other factors, such as values of -100, which affected the **NDTI** calculation and led to results that exceeded the theoretical range of [-1, 1]. We directly downloaded the raw data and provided it to the agent for processing, which contributed to the erroneous values. As a result, we have replaced the input data for these problems with versions that have removed the outliers, ensuring that the data used in the calculations is valid and consistent with the expected range. We appreciate your careful review, and we have now corrected this issue to maintain the integrity of the benchmark.
> ___
> - **Methodological Justification**: The method count_skeleton_contours is a custom tool we implemented to count the number of connected components in a binarized image. Because the original name was ambiguous, we have renamed it to count_connected_components.
> ___
> - **Question Design Issues**: In the cases shown on pages 53 and 62, the full questions involved 12 and 16 images respectively. Due to space limitations, however, we displayed only four images from each case in the paper. We have provided the actual number of files for each case to ensure greater transparency. You can check on page 57 and 58.
> ___
> - **Self-Conflicting Content**: Case shown in page 65 mistakenly displayed outdated versions of the question stem, leading to apparent inconsistencies. After verification, the correct question in both cases should ask for the area of baseball diamonds. We have corrected the case figure in the paper PDF.
> ___
> - **Ambiguity in Measurement Definition**: We previously unintentionally conflated the terms “distance to the centroid” and “distance,” which caused the questions to have different semantic meanings under the autonomous planning and instruction-following settings. We have corrected this issue and re-evaluated the agent’s performance using the revised questions.
>
> Thank you for your careful suggestions! We have meticulously revised our manuscript and benchmark, with all modifications highlighted in blue. We remain committed to further enhancing the quality of the questions. Once again, we deeply appreciate your thorough and insightful feedback.

---

> ### Author Response · Authors · 2025-11-21
> **Reponse to Questions:**
>
> ## **Q1: Clarification of Anti-Planning and Instruction-Following**
>
> To clarify, **all 248 questions** (comprising **100 Spectrum**, **88 Products**, and **60 RGB** questions) have both **auto-planning** and **instruction-following** versions. We have not duplicated the questions; instead, each distinct question has been formulated in two different task configurations to assess the model's performance under both query regimes. ***We added a clarification note on line 340 in the revised version.***
>
> ___
>
> ## **Q2: Dataset Composition**
>
> Thank you for this suggestion. The imageries used in our dataset primarily comes from **Landsat** and **MODIS**, with additional high-resolution imagery sourced from open-source datasets. All the products data is obtained from **Google Earth Engine (GEE)**. Below is a more detailed breakdown of the data sources, products, and their distribution:
>
> | **Category** | **Sensor/Data Source** | **Dataset**        | **Samples** |
> |--------------|------------------------|--------------------|-------------|
> | **Spectrum** | Landsat                | Landsat 8 / 9      | 1684        |
> |              | MODIS                  | -                  | 10273       |
> |              | ASTER                  | -                  | 110         |
> | **Product**  | GEE                    | LST                | 183         |
> |              |                        | NDVI               | 369         |
> |              |                        | GPM                | 160         |
> |              |                        | VIIRS              | 164         |
> |              |                        | GHSL               | 68          |
> |              |                        | QA PIXEL           | 212         |
> |              |                        | NDWI               | 69          |
> |              |                        | fire MaxFRP        | 194         |
> | **RGB**      | Public Datasets        | AID                | 169         |
> |              |                        | DIOR-RSVG          | 7           |
> |              |                        | DOTA               | 22          |
> |              |                        | NWPU-VHR-10        | 21          |
> |              |                        | xBD                | 32          |
> | **Total**    | -                      | -                  | 13737       |
>
> ***We have added this information in the revised version: Appendix A.1 (page 17) for Dataset Composition.***
>
> ___
>
> ## **Q3: Figure 7 Inconsistency**
>
> To clarify, the error counts shown in **Figure 7** refer to all the errors made by the agent throughout the entire reasonig process, not just final anwer error per question. Since an LLM may make multiple tool call errors while attempting to solve a single question, we have included all such errors in the statistics. As a result, the total error count exceeds the number of questions (248). We hope this clarifies the discrepancy. Thank you again for your insightful feedback.
>
> ___
>
> ## **Q4: Evaluation Without Tool Use**
>
> In Appendix A.6: Bias Ablation Experiment, our goal was to demonstrate that the performance difference between the ***No Tool*** and ***Tool Use*** settings is entirely due to the interaction with the tools, rather than other factors. Therefore, we removed the model's ability to perceive external tools and relied solely on its pre-trained knowledge to generate responses. For RGB tasks, base64 encoding was not applied especially here. ***However, in the revised version on page 19, Appendix A.6, we have added the results of applying base64 encoding to the RGB tasks. The conclusions remain consistent with the original findings.***

---

> ### Comment · Reviewer_ic4k · 2025-11-24
>
> I appreciate the authors’ detailed rebuttal and the effort involved in curating EO datasets. I decided to maintain my scores.
>
> # Concerns About Dataset Reliability
>
> The issues identified in the appendices raise substantial concerns of mine about the dataset’s overall reliability.
>
> Specifically, after my review and the authors' rebuttal, I think **3 out of the 15 provided examples (20%) contain clear problems, including incorrect value ranges, self-conflicting content, or ambiguous definitions**, of the format provided at the time of submission. (See "Clarification about Benchmark")
> Given that these examples are presumably among the most illustrative and straightforward cases, this error rate suggests that the true inconsistencies rate may be higher.
> This raises concerns: Does accepting this paper provide significant value for EO or tool-augmented LLM benchmarking, or does it introduce more uncertainty to the community than it benefits? The level of error observed here **risks undermining confidence** in both evaluation outcomes and future research built upon this dataset.
>
> One more note: I find the nature of these errors (specifically the incorrect formulas and value ranges) difficult to reconcile with the authors' rebuttal. These are **elementary oversights** that should have been caught during any expert review process. The presence of such basic flaws undermines the claim that '12 researchers with expertise' rigorously verified this dataset."
>
> # Limited Real-World Utility
>
> The authors’ rebuttal regarding the "Incorrect Formula Range" issue raises an additional concern regarding the realism of the dataset. In this case, the original Google Earth Engine raw data uses -100 to fill missing values, which cannot be directly processed with the predefined tools, and thus, the authors further processed the raw data.
>
> While I appreciate the authors' honesty and endeavors, this suggests that: (1) the tool environment is incomplete; (2) the dataset has been tailored to the limitations of the sandbox; and (3) models performing well in this environment may not generalize to real operational EO workflows, thereby limiting the practical value of the benchmark.

---

> > ### Author Response · Authors · 2025-11-26
> > **Reponse to further concerns and suggestions: (1/2)**
> >
> > We sincerely appreciate the time and effort you invested in providing such a detailed review and valuable feedback. After receiving your initial comment, **we conducted a thorough re-examination and update of the dataset, followed by additional testing results these days**. We hope these updates will alleviate your concerns about the dataset's reliability.
> >
> > ___
> >
> > # **Re: Concerns about Dataset Reliability**
> >
> > (1）Some of the concerns like ***Question Predictability*** and ***Question Design*** are due to the presentation of the cases in appendix, rather than flaw with the questions themselves. We clarified these points in the revised version and appreciate your understanding.
> >
> > (2) For the issues related to earth science discipline, we have carefully rechecked each question of the dataset and retested the results. In total, there are 13 questions that may contain subject problems, which accounts for approximately 5% in the total dataset. Specifically：
> >
> > 1. Three questions contained pixels with unremoved invalid values (182,183,184);
> > 2. Four questions involved confusion between distance and centroid distance (233,234,235,236);
> > 3. Six questions had pixels that were not multiplied by the scaling factor (122,124,141,155,156,157).
> >
> > We re-evaluated the performance of different agents on the revised dataset, and below are the updated results:
> >
> > |                    | **Tool-Any-Order** |                 | **Tool-In-Order**  |                 | **Tool-Exact-Match** |                 | **Parameters**     |                 | **Efficiency**     |                 | **Accuracy**       |                 |
> > |:--------------------:|:--------------------:|:-----------------:|:--------------------:|:-----------------:|:----------------------:|:-----------------:|:--------------------:|:-----------------:|:--------------------:|:-----------------:|:--------------------:|:-----------------:|
> > |                    | **AP**             | **IF**          | **AP**             | **IF**          | **AP**               | **IF**          | **AP**             | **IF**          | **AP**             | **IF**          | **AP**             | **IF**          |
> > | **GPT-5**          | 69.11              | 71.16           | 58.25              | 60.68           | 45.57                | 45.79           | 26.10              | 25.80           | 2.4146             | 2.8428          | 65.69              | 63.16           |
> > | **GPT-5(revised)**          | 68.74              | 71.41           | 57.71              | 61.06           | 44.97                | 46.01           | 26.11              | 25.91           | 2.3560             | 2.9093          | 65.99              | 62.35           |
> > | **Gemini-2.5**     | 57.96              | 61.80           | 45.44              | 50.78           | 31.86                | 40.92           | 17.49              | 23.54           | 3.0246             | 2.4709          | 54.66              | 55.06           |
> > | **Gemini-2.5(revised)**     | 58.04              | 61.63           | 45.31              | 50.72           | 31.32                | 41.04           | 17.26              | 23.43           | 2.9958             | 2.4595          | 52.23              | 53.04           |
> > | **GPT-4o**         | 65.73              | 66.88           | 50.70              | 53.20           | 46.17                | 47.47           | 26.71              | 27.77           | 2.1579             | 2.6088          | 45.34              | 44.94           |
> > | **GPT-4o(revised)**         | 65.65              | 67.02           | 50.76              | 53.04           | 46.26                | 47.41           | 26.55              | 27.92           | 2.1211             | 2.6007          | 43.72              | 44.94           |
> > | **Kimik2**         | 71.03              | 78.86           | 57.57              | 68.83           | 42.11                | 51.15           | 25.90              | 30.45           | 1.8542             | 2.1793          | 50.61              | 56.68           |
> > | **Kimik2(revised)**         | 70.88              | 79.15           | 57.68              | 68.93           | 42.07                | 50.95           | 25.89              | 30.53           | 1.8688             | 2.1884          | 49.39             | 56.28           |
> > | **DeepSeek-V3.1**  | 78.40              | 77.98           | 63.25              | 64.33           | 48.89                | 50.01           | 30.76              | 31.34           | 2.6283             | 2.6624          | 50.20              | 52.23           |
> > | **DeepSeek-V3.1(revised)**  | 78.31              | 78.66           | 62.73              | 64.50           | 48.54                | 49.58           | 30.81              | 31.36           | 2.6116             | 2.6303          | 51.42              | 52.23           |

---

> > ### Author Response · Authors · 2025-11-26
> > **Reponse to further concerns and suggestions: (2/2)**
> >
> > The revised testing results show ***minor differences*** compared to the previous ones, and they do not impact our overall analysis. However, as you mentioned, the intention of our work is to inspire more future research on agent paradigm based on Earth-Bench. We do need to take the responsibility to set a precedent with this unprecedented work. We really appreciate your professional identification of these issues, and we hope that these results can help alleviate your concerns to some extent.
> >
> > ___
> >
> > # **Re: Concern about limited real-world utility**
> >
> > Regarding the ***"Incorrect Formula Range"*** issue, raw band data obtained from satellite sensors typically requires latitude-longitude matching, band extraction, and radiometric conversion. The raw data may contain fill or invalid values. In our Earth-Agent tool, **we have indeed considered the handling of common fill values**, such as -9999 and 65535, and relevant tasks process these correctly. However, there are a small number of bands containing other invalid values, such as -100 in this case, which were not excluded initially, leading to erroneous results. Such exceptional cases constitute an extremely low proportion of the tasks. To ensure the rigour of the tasks for futher evaluation, we have implemented manual processing for these circumstances.
> >
> > Thank you for your thoughtful and constructive suggestions. In this work, we focused more on evaluating the potential of Agents in addressing Earth science problems rather than on the full data processing workflow. We have considered common circumstances in the tool designing, but ***different data source may adopt different conventions for invalid values and may require special handling***. This requires supplementing the terms of use for the dataset to Earth-Agent. Your suggestions are very constructive, and they will help Earth-Agent better integrate into real operational EO workflows. Moving forward, we need to add a dedicated preprocessing toolkit module for satellites and sensors to further reduce manual intervention and expand the workflow scope of Earth-Agent, enabling it to reliably provide comprehensive results. We plan to integrate tools for retrieving calibration metadata from online resources or databases in subsequent versions, thereby minimizing the need for human expert intervention and making the system's workflow more comprehensive and automated.
> >
> > ___
> >
> > Overall, we are deeply grateful for your detailed feedback and suggestions on ***Earth-Bench*** and ***Earth-Agent*** from such a professional perspective. After our discussion, **this has become a more rigorous precedent**, which is our most valuable takeaway. We hope that our response can enhance confidence in both the evaluation outcomes and future research built upon our work for you and other researchers. We take ongoing responsibility for the quality of our research, and if you have any further concerns, please feel free to raise them. We assure you that we will provide intime responses. Once again, thank you for your time and comments.

---

> ### Comment · Reviewer_ic4k · 2025-11-27
> **Concerns Regarding Dataset Validity and Quality Assurance**
>
> I am writing to provide specific context for the Area Chairs and other reviewers regarding my concerns about the reliability of this dataset.
>
> **Context: Page43 in the original submission**
>
> **Question**: Based on the sur_refl_b01 and sur_refl_b04 data of the Arctic Ocean, calculate the NDTI. Determine [...]
>
> - **Option A**: The average NDTI increased from about **-666.47** in July 2012 to about **-61.17** in July 2022, indicating a significant decrease in ocean turbidity, with a magnitude change of about **605.30**. (B and C ignored for similar pattern)
>
> - **Option D**: The average NDTI changed from about **-200.00** to about **-100.00**, showing a minor increase in ocean turbidity, with a magnitude change of about **100.00**.
>
> Common knowledge in remote sensing:
> - **Normalized Difference Turbidity Index (NDTI)** formula is (Red - Green) / (Red + Green). Here, "Red" refers to the value at the red end of the spectrum. NDTI is a normalized index bounded strictly between -1 and 1.
> - There can be multiple different prefilled values depending on the sources.
>
> # **Why I am concerned about this dataset**
>
> My concerns consist of two aspects
>
> - **Failure of Annotation**: If human experts were involved as claimed, they failed to catch values that are impossible. This suggests the "expert annotation" was either non-existent or conducted by non-experts.
> - **Failure of Verification**: A robust dataset curation pipeline should at least include basic sanity checks (e.g., range constraints for normalized indices). The presence of these outliers indicates that such checks are missing.
>
> This example undermines the credibility of the dataset's curation process. If "impossible" values pass the quality control checks, it casts doubt on the validity of the "plausible" values as well. Consequently, I cannot confirm the reliability of this benchmark.

---

> > ### Author Response · Authors · 2025-12-03
> > **Reponse to Concerns**
> >
> > ## **Re: Concern about our dataset**
> >
> > We sincerely thank the reviewer for the careful and through assessment of our submission. The issues identified are constructive, and we genuinely appreciate this valuable feedback. The reviewer’s comments helped us uncover several inconsistencies caused by preprocessing oversights, and **we have carefully reflected on and addressed these issues in our revised version**.
> >
> > To be specific, we have taken two concrete steps to remedy these issues:
> >
> > 1. We re-inspected the entire dataset and identified that approximately ***5%*** of the questions were affected. We have **corrected these questions** and **re-evaluated the results concerning all the models**. The updated scores show only minor changes, indicating that the overall conclusions remain reasonable.
> >
> > 2. We **enhanced Earth-Agent with additional preprocessing tools**, and we plan to continue expanding its capabilities to support more complete end-to-end workflows in future work. These improvements directly respond to the reviewer’s suggestions and further strengthen the scientific rigor of our benchmark.
> >
> > ## **Re: Concern about expert annotation**
> >
> > As stated in Section 4.2 of the paper (line 329), all annotations in Earth-Bench were produced entirely by domain experts, ensuring the reliability and correctness of the ground-truth answers. To further enhance transparency, **we have uploaded components of our dataset, including the benchmark, evaluation code, and the model’s evaluation JSON logs, allowing reviewers and future researchers to directly inspect the full dataset**. We believe this openness provides the community with strong confidence in the quality and consistency of the annotations.

---

### Official Review · Reviewer_1ij5 · 2025-11-01

**Soundness:** 4
**Presentation:** 3
**Contribution:** 4
**Rating:** 6
**Confidence:** 5

**Summary:**

This paper has two main accomplishments: First it proposes an agentic framework for earth observation integrating 104 tools and models. Secondly they prepare a benchmark (Earth-Bench) of 248 expert-curated questions with 13,729 image.  They then evaluate the agent on the benchmark, comparing it to openai operator and Manus. They also demonstrate that the model outperforms remote sensing foundation models.

**Strengths:**

The sterngth of this work are the results: both with the agentic system design, the benchmark design, the demonstration of SOTA results across a range of existing remote sensing benchmarks . The agentic system uses a wide set of specialized tools that were reasonably selected and the authors evaluated this system across a range of LLMs. They also constructed a smaller benchmark (earthbench-lite) to enable comparison against generalized agents and show that their agent outperforms consistently

**Weaknesses:**

When reading the paper I would have liked to learn more about the benchmark itself. How it was constructed, what are the types of question in it, and so forth. Some information is in the appendix but I feel more detailed analysis would be helpful.

Also the authors have a long tool list and they also built a benchmark. We need to understand the relationship of the tool list to how the benchmark was constructed: at an extreme limit you could have included tools that were specifically chosen to solve individual sets of questions and thus push benchmark performance way up but in a way that is not extensible to larger sets of questions not included. I'd feel better about this if the authors could give a rationalization for their toolset that is independent of their benchmark.  Some solace is given since they outperform previous benchmarks but as they themselves point out the benchmarks are not really comparable.

One benchmark that might be more useful to compare against is this one: https://openreview.net/pdf?id=oaYShIy3Xe.  I'd be curious how well this system does against that benchmark

**Questions:**

Please respond to the weaknesses I highlighted:
1. Explain more about the benchmark and why you made the choices you did when constructed it
2. Explain the relationship of your toolset to your benchmarks and what you did to make sure you weren't cherry picking tools for benchmark questions
3. Explain how your agent does on the cited benchmark above.

---

> ### Author Response · Authors · 2025-11-21
> **Response to Question 1:**
>
> Thank you for your comment. We are very glad to provide a more detailed explanation of the **Earth-Bench** benchmark construction, the types of questions included, and the underlying design motivations.
>
> ***We have added detailed explanations in the revised version: Appendix A.1 (page 17) for Dataset Composition, Appendix A.3 (page 17) for Question Types and Categories, and Appendix A.5 (page 17) for Dataset Quality Control.***
>
> ---
>
> ### **Benchmark Construction Pipeline**
>
> Our annotation team was composed of **2** computer science experts, **7** remote sensing specialists (including one professor) and **3** Earth science specialists (including one professor).
>
> Three core authors who served as senior reviewers led the pipeline of dataset construction. Each senior reviewer was responsible for guiding the development of the question sets and templates for the **Spectrum**, **Products**, and **RGB** categories. They played a key role in providing strategic direction for the overall question pool.
>
> The remaining **7** team members, consisting of graduate students and senior undergraduates, contributed in the following areas:
>
> 1. Creation of initial questions (approximately 400 questions),
> 2. Raw data collection,
> 3. Development of Python-based solution scripts.
>
> Once the questions were created, they were thoroughly reviewed by the three senior reviewers. The review process focused on two key criteria:
> 1. **Data Integrity**: Ensuring that the raw data involved (e.g., Landsat or MODIS) had complete band information within the task’s time range and did not contain anomalies. Any questions and tif files with missing or anomalous large values were discarded.
> 2. **Task Difficulty Assessment**: Senior reviewers assessed the difficulty of questions based on the number and complexity of the functions defined in the Python solutions. For simpler tasks (e.g., calculating a single NDVI index), these were removed to ensure appropriate challenge across questions.
>
> ---
>
> ### **Question Types and Category**
> As detailed in **Figure 4** on Page 6 of the paper, Earth-Bench questions are categorized into three types based on data sources: **Spectrum**, **Products**, and **RGB**.
>
> - **Spectrum and Products** questions cover **8** application scenarios: Temperature Monitoring, Weather Forecasting, Climate Analysis, Water Management, Pollution Regulation, Vegetation Monitoring, Disaster Judgment, and Urban Management. These align with widely recognized Earth science task classifications from **Google Earth Engine (GEE)** task classification system ([categories](https://developers.google.cn/earth-engine/datasets/categories)).
>
> - **RGB** questions cover **6** traditional visual tasks: Classification, Detection, Grounding, Segmentation, Counting, and Change Detection. These tasks, derived from classic remote sensing datasets, assess models' performance in multimodal perception and visual understanding tasks.
>
> Below is the number and proportion of questions for each category:
>
> | Task                   |   Number |   Percent(%) |
> |:-----------------------|---------:|----------:|
> | Temperature Monitoring |       44 |     17.74 |
> | Weather Forecasting    |       11 |      4.44 |
> | Climate Analysis       |       20 |      8.06 |
> | Water Management       |       22 |      8.87 |
> | Pollution Regulation   |       14 |      5.65 |
> | Vegetation Monitoring  |       28 |     11.29 |
> | Disaster Judgement     |       24 |      9.68 |
> | Urban Management       |       25 |     10.08 |
> | Classification         |       15 |      6.05 |
> | Detection              |       15 |      6.05 |
> | Grounding              |        7 |      2.82 |
> | Segmentation           |        3 |      1.21 |
> | Counting               |        7 |      2.82 |
> | ChangeDetection        |       13 |      5.24 |
>
>
> We hope this more detailed explanation provides the clarity you were looking for regarding the benchmark's construction and details. Thank you for your valuable sugesstion!

---

> ### Author Response · Authors · 2025-11-21
> **Response to Question 2:**
>
> Thank you for your comment. We understand your concern about whether the toolset was specifically tailored for the benchmark questions. Below is an explanation of the relationship between our toolset and the benchmark, as well as the steps we took to ensure that no tools were cherry-picked.
> ___
> ### **Relationship Between Toolset and Benchmark**
> The Earth-Agent evaluation is a **predefined sandbox experiment** to test LLMs' ability to process Earth science tasks using a predefined toolset, not customized for specific benchmark questions. The toolset are developed based on common Earth science requirements (e.g., NDVI, NDWI, and time-series analysis). These tools were designed before the benchmark questions, ensuring that the toolset is independent of specific tasks.
>
> Before constructing the benchmark, the three senior reviewers conducted survey on widely used tools and function modules in Earth science, such as **NDVI**, **NDWI**, and **time-series trend analysis**. Based on this, we developed a standardized toolset consisting of **104** tools.
>
> We followed two core principles to avoid bias:
>
> 1. **Atomicity**: Each tool performs a single function (e.g., calculating NDVI or performing linear trend analysis), rather than multi-step composite tasks. This ensures that the agent needs to call multiple tools to solve a task, testing its planning and reasoning capabilities.
>
> 2. **Generality**: We avoided designing tools specific to a single task. For example, the task "Find the day with the highest NDWI value for the month" (Figure 23, page 48) uses general tools like **calculate_threshold_ratio** and **calc_batch_image_max**, which can be applied across various tasks such as **Weather Forecasting**, **Water Management**, and **Pollution Regulation**, rather than being restricted to NDWI-related tasks.
>
> The benchmark questions were then designed by **7** graduate students and senior undergraduates, ensuring that the toolset served as the foundation for the question design, rather than being customized for specific benchmark tasks. Through this design, we ensure that the toolset is independent and general, eliminating any potential bias where tools are created specifically for the benchmark questions.
>
> ### **Further Validation**
>
> We also reviewed the **top 10 most used tools** in the benchmark, and it can be found that they were not specifically designed for any particular question:
>
> | Top-10 most used tool name  | Number |
> |-------------------------------|-----:|
> | get_filelist                   |  271 |
> | calc_batch_image_mean          |  259 |
> | MSCN                           |  187 |
> | calc_batch_image_hotspot_percentage |  164 |
> | calc_batch_image_mean_mean     |  132 |
> | band_ratio                     |  127 |
> | lst_single_channel             |  114 |
> | ATI                            |  100 |
> | percentage_change              |   90 |
> | lst_multi_channel              |   85 |
>
> On the other hand, there are 22 tools were never used when evaluating,which can further demonstrating that the toolset was pre-constructed based on expert experience, rather than being selectively designed for Earth-Bench.
>
> ```shell
> 1. bbox_expansion, 2. calculate_batch_evi, 3. calculate_batch_frp, 4. calculate_batch_fvc, 5. calculate_batch_ndbi, 6. calculate_batch_wri, 7. ceil_number, 8. celsius_to_kelvin, 9. chang_single_param_inversion, 10. count_images_exceeding_mean_multiplier, 11. count_spikes_from_values, 12. detect_change_points, 13. detect_seasonality_acf, 14. dual_frequency_diff, 15. dual_polarization_differential, 16. dual_polarization_ratio, 17. grayscale_to_colormap, 18. min_value_and_index, 19. multi_freq_bt, 20. radiometric_correction_sr, 21. skewness, 22. stl_decompose
> ```
>
> We hope this explanation clarifies how the toolset was constructed and how we ensured the benchmark's fairness by avoiding cherry picking bias.

---

> ### Author Response · Authors · 2025-11-26
> **Response to Question 3:**
>
> We appreciate the reviewer’s suggestion to include evaluation results on CBGB to validate generalizability. However, this benchmark follows an execution paradigm fundamentally different from that of our framework. CBGB requires the agent to generate executable code that directly interacts with remote cloud-based datasets and services. In contrast, our system is designed around local data processing through tool calling, where the LLM operates a predefined local geospatial tools rather than issuing arbitrary cloud API calls. In addition, many CBGB tasks involve data exceeding the download size limit of Google Earth Engine (50,331,648 bytes), which makes comparison nontrivial.
>
> To address your concern about our framework, we added a lightweight code compiler/executor tool to our agent. This addition does not alter the core design philosophy of our framework. It simply enables the agent to run code when necessary to interface with the cloud-based data used in CBGB. With this capability in place, we conducted the full CBGB evaluation. The following table reports the total number of correctly solved tasks for the base models (CBGB), the error-correction (EC) agents (CBGB), and our Earth-Agent:
>
>
> | Model               | base | ec | **ours**     |
> |---------------------|------|----|----------|
> | gpt-5               | 22   | 36 | 36       |
> | o3-2025-04-16       | 29   | 32 | 30       |
> | gemini-2.5-pro      | 25   | 32 | 32       |
> | claude-3-7-sonnet   | 23   | 31 | 35       |
> | deepseek-reasoner   | 22   | 29 | 29       |
> | o4-mini-2025-04-16  | 22   | 27 | 29       |
> | deepseek-chat       | 20   | 21 | 25       |
> | claude-3-5-haiku    | 15   | 22 | 8        |
> | **total**           | **45**| **45** | **45** |
>
> Based on the table, only Claude-3.5 exhibits low performance, as our Earth-Agent architecture is based on the ***Model Context Protocol (MCP)*** framework, and Claude-3.5 lacks this compatibility, preventing it from accessing the necessary tools to complete these tasks. In contrast, other LLMs were able to correctly utilize the available tools to address the CBGB questions, **with their performance comparable to the error-correction (ec) mode** reported in the paper. We hope this can help strengthen your confidence in the reliability of our architecture.

---

### Official Review · Reviewer_a78N · 2025-11-06

**Soundness:** 3
**Presentation:** 4
**Contribution:** 4
**Rating:** 8
**Confidence:** 4

**Summary:**

The paper introduces Earth-Agent, a novel agent-based framework for Earth Observation (EO) analysis using multimodal large language models (MLLMs). Unlike previous EO models that rely mainly on RGB data and limited reasoning, Earth-Agent integrates RGB and spectral EO data and leverages MCP tools for cross-modal, multi-step, and quantitative spatiotemporal reasoning. It can dynamically employ expert tools for tasks such as geophysical parameter retrieval and spatiotemporal analysis. To evaluate its performance, the authors propose Earth-Bench, a benchmark comprising 248 expert-designed tasks and 13,729 images, offering dual-level evaluation on both reasoning and outcomes. Experiments across various LLM backbones show that Earth-Agent significantly outperforms existing MLLMs and general agent frameworks.

**Strengths:**

- Solving earth observation tasks using an agentic solution is quite novel and timely. Bringing a tool-use angle from agent solutions to improve performance seems very innovative.
- The proposed benchmark can be useful for the research community in terms of evaluating agentic systems.
- Experiments are thorough and comprehensive. Findings are supported by enough experiments.
- Paper is well-written and easy to follow.

**Weaknesses:**

- In any agentic system, a comparison of latency with a single-agent or single-model baseline is expected. Although the performance gains are significant, it is also important to account for the computational cost, including the number of LLM or tool calls and the total end-to-end response time. Such comparisons are important to demonstrate the overall efficiency and effectiveness of the proposed agent framework. Please add a discussion around it in the paper.

- A discussion of performance trends w.r.t. to the number of tools used for a given task is important to understand how the proposed agentic system behaves as task complexity increases and the number of tool calls grows. It would also help evaluate the system’s latency and scalability under such conditions. I would like to see such ablation studies included in the paper to better demonstrate the effectiveness of the proposed method.

**Questions:**

- One aspect that is unclear is how the proposed method handles cases where a tool produces an incorrect output. The framework does not appear to inherently address such failure cases. For example, if an error occurs during the first tool call, it may propagate and amplify throughout the subsequent reasoning chain. Did the authors encounter or observe such cases in their experiments?

- Throughout the paper, the authors repeatedly state that the dataset is expert-curated. However, the paper does not specify who these experts are or provide details about their backgrounds. It would be helpful to clarify whether the curators come from computer science with remote-sensing expertise, or if they are domain specialists primarily from the remote-sensing or Earth observation community.

---

> ### Author Response · Authors · 2025-11-21
> **Response to Weakness 1:**
>
> We sincerely thank the Reviewer for emphasizing the importance of comparing latency with a single-agent or single-model baseline. ***We have added this discussion in the revised version on page 24, in Appendix E: Latency Experiment.***
>
> ---
>
> ## **Re: W1 - Latency Experiment**
> We ultilized Langsmith to record the latency of Earth-Agent. As shown in Table 1, we compared the latency of our Earth-Agent framework (driven by different models like GPT, DeepSeek, and Kimik2) with baseline agents, such as GPT-Agent, MGX, and Manus. Earth-Agent **remains within a reasonable range** when compared to existing general agents. Moreover, the significant performance gains in task completion justify the added computational cost.
>
>
> | Model                         | Spectrum | Products | RGB   | Avg.   | Latency |
> |-------------------------------|:----------:|:----------:|:-------:|:--------:|:---------:|
> | GPT-Agent                      | 45.00    | 31.60    | 45.26 | 40.42  | 300 min |
> | MGX                            | 40.00    | 15.80    | 0.00  | 18.60  | 60 min  |
> | Manus                          | 15.00    | 15.80    | 47.62 | 26.14  | 150 min |
> | Coze                           | 35.00    | 10.50    | 0.00  | 15.30  | 120 min |
> | **Earth-Agent (GPT5)**             | 65.00    | 36.84    | 65.71 | 55.83  | 158 min |
> | **Earth-Agent (DeepSeek-V3.1)**    | 50.00    | 42.11    | 51.43 | 47.84  | 79 min  |
> | **Earth-Agent (Kimik2)**           | 36.84    | 50.00    | 50.00 | 45.95  | 131 min |
>
> ___
>
> In Table 2, we further break down the latency into ***LLM Latency*** and ***Tool Latency***. It is evident that most of the latency is spent on model calls (***LLM Latency***), rather than on external tool calls(***Tool Latency***). This suggests that reducing the frequency of calling model could further improve latency.
>
> | Model                         | Latency   | Tool Call Number | LLM Latency  | Tool Latency |
> |------------------------------|:---------:|:---------------:|:------------:|:------------:|
> | Earth-Agent (GPT)              | 9494.92s  | 669             | 6969.26s     | 2525.66s     |
> | Earth-Agent (DeepSeek-V3.1)         | 4716.99s  | 540             | 3066.76s     | 1650.23s     |
> | Earth-Agent (Kimik2)           | 7903.07s  | 502             | 6306.59s     | 1596.47s     |
>
> ___
>
> In fact, we have considered this issue when designing Earth-Agent architecture. We optimized the design of our tools to minimize unnecessary interactions with the model. For example, by using batch calculating earth indexes like NDVI calculation, we significantly reduce the frequency of model interactions, which in turn lowers the overall latency. Here is an example of our method:
>
> ```python
> def calculate_batch_ndvi(
>     input_nir_paths: list[str],
>     input_red_paths: list[str],
>     output_paths: list[str]
> ) -> list[str]:
>     return [
>         calculate_ndvi(nir_path, red_path, out_path)
>         for nir_path, red_path, out_path in zip(input_nir_paths, input_red_paths, output_paths)
>     ]# The agent only needs to call LLM once using batch method

---

> ### Author Response · Authors · 2025-11-21
> **Response to Weakness 2: （1/2）**
>
> ***We have added this discussion in the revised version on page 26, in Appendix F: Scalability Discussion***. Feel free to check it!
> ___
>
> ## **Re: W2 - Performance Trends w.r.t. to the Number of Tools**
>
> We appreciate the reviewer's comment regarding the scalability of Earth-Agent. This analysis is important for evaluating both the latency and scalability of the system under such conditions.
>
> To address this, we have extracted the ***tool number*** and ***accuracy results*** for three state-of-the-art models: GPT5, DeepSeek-V3.1, and Kimik2. Below are the results, with the **high performance range** highlighted.
>
> | ***Tool Number*** | **GPT5**               |                 | **DeepSeek-V3.1**          |                 | **Kimik2**            |                 |
> |------------|-----------------------|-----------------|-----------------------|-----------------|-----------------------|-----------------|
> |            | Total questions        | Accuracy (%)| Total questions         | Accuracy (%)| Total questions         | Accuracy (%)|
> | 0          | 25                    | 8.00            | -                    | -              | 25                    | 4.00            |
> | 1          | 3                     | **66.67**           | 2                     | 0.00            | 1                     | 0.00            |
> | 2          | 8                     | **75.00**           | -                    | -              | 5                     | 40.00           |
> | 3          | 42                    | **80.95**           | 23                    | **73.91**           | 17                    | **76.47**           |
> | 4          | 29                    | **89.66**           | 23                    | **86.96**           | 21                    | **66.67**           |
> | 5          | 30                    | **60.00**           | 19                    | **68.42**           | 21                    | **71.43**           |
> | 6          | 7                     | **85.71**           | 15                    | **60.00**           | 27                    | **70.37**           |
> | 7          | 10                    | **90.00**           | 15                    | 40.00           | 16                    | **56.25**           |
> | 8          | 10                    | **80.00**           | 8                     | **87.50**           | 20                    | **65.00**           |
> | 9          | 4                     | **75.00**           | 10                    | **60.00**           | 4                     | **25.00**           |
> | 10         | 10                    | **70.00**           | 11                    | **54.55**           | 5                     | **40.00**           |
> | 11         | 4                     | **50.00**           | 8                     | **75.00**           | 11                    | **54.55**           |
> | 12         | 3                     | 33.33           | 8                     | **75.00**           | 9                     | **66.67**           |
> | 13         | 6                     | **100.00**          | 17                    | **70.59**           | 20                    | **80.00**           |
> | 14         | 4                     | **75.00**           | 4                     | **50.00**           | 3                     | 33.33           |
> | 15         | 2                     | 0.00            | 5                     | **100.00**          | 4                     | 50.00           |
> | 16         | 2                     | 100.00          | 3                     | 66.67           | 1                     | 0.00            |
> | 17         | 4                     | 50.00           | 4                     | 25.00           | 3                     | 66.67           |
> | 18         | 4                     | 50.00           | 4                     | 25.00           | 2                     | 50.00           |
> | ...        | ...                   | ...             | ...                   | ...             | ...                   | ...             |
> | 159        | 1                     | 100.00          | -                    | -              | -                    | -              |
>
> We observed that for Earth-Agent driven by GPT5, high accuracy is mainly concentrated in the **range of 1 to 14** ***tool number***; for Earth-Agent driven by DeepSeek-V3.1, high accuracy is mainly concentrated in the **range of 3 to 15** ***tool number***; and for Earth-Agent driven by Kimik2, high accuracy is mainly concentrated in the **range of 3 to 13** ***tool number***. These **high performance ranges** are within the expected scope. Using too few tools (with the extreme case being zero tool) indicates that the agent is confused by the task, often resulting in errors at the beginning. On the other hand, low performance when using too many tools suggests that the current capabilities of the base LLMs are insufficient to handle long chains of reasoning.

---

> ### Author Response · Authors · 2025-11-21
> **Response to Weakness 2: （2/2）**
>
> Specifically, we have plotted a bubble chart in appendix F.2 to visually illustrate the relationship between ***model performance*** and the ***number of unique tools***. Feel free to check it!
> ___
> ## **Re: W2 - Performance Trends w.r.t. to the Number of Tools**
> Based on your insightful suggestion, we also explored the relationship between the ***unique number of tools*** used and ***performance trends***. The results are as follows, with the **high performance range** highlighted.
>
> | ***Unique Tool Number*** | **GPT5**               |                 | **DeepSeek-V3.1**          |                 | **Kimik2**            |                 |
> |-------------------|-----------------------|----------------------|-----------------------|--------------------|-----------------------|--------------------|
> |                   | Total questions         | Accuracy (%)| Total questions         | Accuracy (%)| Total questions        | Accuracy (%)|
> | 0                 | 25                    | 8.00            | -                     | -               | 25                    | 4.00            |
> | 1                 | 3                     | **66.67**           | 2                     | 0.00            | 2                     | 50.00           |
> | 2                 | 29                    | **79.31**           | 24                    | **66.67**           | 29                    | **65.52**           |
> | 3                 | 90                    | **71.11**           | 57                    | **63.16**           | 47                    | **63.83**           |
> | 4                 | 56                    | **80.36**           | 39                    | **64.10**           | 53                    | **58.49**           |
> | 5                 | 20                    | 45.00           | 39                    | **56.41**           | 36                    | **58.33**           |
> | 6                 | 13                    | **92.31**           | 22                    | 31.82           | 23                    | 39.13           |
> | 7                 | 3                     | 33.33           | 14                    | 35.71           | 13                    | **53.85**           |
> | 8                 | 2                     | **50.00**           | 16                    | 43.75           | 8                     | 50.00           |
> | 9                 | 2                     | **100.00**          | 13                    | 23.08           | 5                     | 40.00           |
> | 10                | 2                     | 50.00           | 7                     | 0.00            | 4                     | 0.00            |
> | 11                | 2                     | 0.00            | 3                     | 0.00            | 1                     | 0.00            |
> | 12                | -                     | -               | 4                     | 75.00           | 1                     | 0.00            |
> | 13                | -                     | -               | 2                     | 0.00            | -                     | -               |
> | 14                | -                     | -               | 1                     | 0.00            | -                     | -               |
> | 15                | -                     | -               | 1                     | 0.00            | -                     | -               |
> | 16                | -                     | -               | 2                     | 0.00            | -                     | -               |
> | 18                | -                     | -               | 1                     | 0.00            | -                     | -               |
>
>
> We observed that in the Earth-Agent system, when driven by GPT5, high accuracy is mainly concentrated in the **range of 2 to 5** ***unique tool number***. For DeepSeek, high accuracy is mainly concentrated in the **range of 2 to 6** ***unique tool number***. Similarly, for Kimik2, high accuracy is mainly concentrated in the **range of 1 to 6** ***unique tool number***. The agent performs poorly when the number of tool calls is excessive. This aligns with our expectations regarding the current capabilities of LLMs.
>
> We believe that this analysis provides valuable insights into the system's scalability as task complexity increases. It also offers valuable inspiration for future agent training: (1) focusing on the agent's startup phase and (2) building long-chain reasoning dataset to strengthen the agent's tool calling capabilities.

---

> ### Author Response · Authors · 2025-11-21
> **Response to Questions:**
>
> ## **Re: Q1 - Handing Intermediate Errors & Cascading Errors**
>
> ### **Handing Intermediate Errors**
>
> We completely resonate with this point you propose. In response to the concern regarding handling tool output errors, we would like to clarify the following two points:
>
> 1. **ReAct Mechanism (Observation → Act → Feedback)**: Our framework does not have a dedicated module for handling tool output errors. However, we enhance Earth-Agent's robustness by using the ReAct mechanism, which allows the agent to adjust its actions based on previous tool outputs. This helps the agent identify and correct errors during the reasoning steps, minimizing the propagation of mistakes.
>
> 2. **Prompt Engineering**: In our experimental setup (line 332 in our paper), we compare **Auto-Planning (AP)** and **Instruction-Following (IF)** strategies. The IF strategy allows users to embed problem-solving steps into the question prompt, guiding the agent, while the AP strategy requires the agent to autonomously plan and execute the steps. We compared the differences between these two strategies to assess their impact on the agent's performance and reasoning process.
> ___
>
> ### **Cascading Errors in Agent Reasoning**
>
> We observed the phenomenon you mentioned: **Cascading Errors**. Specifically, during the comparison of **Auto-Planning (AP)** and **Instruction-Following (IF)**, we found that while IF improves tool usage accuracy, it may **introduce irrelevant steps during reasoning** (line 423 in our paper), which leads to unnecessary intermediate errors. As a result, the overall performance does not always improve like tool calling. This implies that enhancing the agent's ability to handle errors cannot rely solely on embeding step instructions into prompts simplily.
>
> Recently, we have also been exploring methods to improve the robustness of agent systems, such as ***AEGIS***[1] and ***Factored Agents***[2]. Our desire to address this issue aligns with yours. However, for this paper, our core motivation is to *set a strong precedent for the Agent4Earth field*. Therefore, we focused on evaluating the feasibility and potential of agents in Earth Science tasks, which we believe is critical for laying the foundation of future work in this domain.
>
> ## **Re: Q2 - Background of Our Experts**
>
> Thanks for your suggestion. ***We added this information in the revised version at line 329. And we also added one subsection on page 26, in Appendix A.5 to clarify this question.***
> ___
>
> Our annotation team was composed of **2** computer science experts, **7** remote sensing specialists (including one professor) and **3** Earth science specialists (including one professor).
>
> Three core authors who served as senior reviewers led the pipeline of dataset construction. Each senior reviewer was responsible for guiding the development of the question sets and templates for the **Spectrum**, **Products**, and **RGB** categories. They played a key role in providing strategic direction for the overall question pool.
>
> The remaining 7 team members, consisting of graduate students and senior undergraduates, contributed in the following areas:
>
> 1. Creation of initial questions (approximately 400 questions),
> 2. Raw data collection,
> 3. Development of Python-based solution scripts.
>
> This collaborative structure ensured that the dataset was curated by a diverse team with expertise from the fullandscape of Earth observation fields, enabling a well-rounded and comprehensive dataset for evaluation.
>
> *References:*
>
> [1] Kong, F., Zhang, R., Yin, H., Zhang, G., Zhang, X., Chen, Z., ... & Feng, X. (2025). Aegis: Automated Error Generation and Attribution for Multi-Agent Systems. arXiv preprint arXiv:2509.14295.
>
> [2] Roth, N., Hidey, C., Spangher, L., Arnold, W. F., Ye, C., Masiewicki, N., ... & Ie, E. (2025). Factored Agents: Decoupling In-Context Learning and Memorization for Robust Tool Use. arXiv preprint arXiv:2503.22931.

---

> ### Comment · Reviewer_a78N · 2025-11-27
>
> The reviewer thanks the authors for their rebuttal. The authors have answered my qyestions well. I will keep my positive score.

---

> > ### Author Response · Authors · 2025-11-27
> > **Official Comment by Authors**
> >
> > Dear Reviewer a78N,
> >
> > We sincerely appreciate your support of our work. Your valuable feedback has greatly contributed to improving its quality and helping it become a more complete precedent in this field!
> >
> > Best regards,
> >
> > The Authors

---

### Public Comment · ~Charlie_Brown1 · 2025-11-26
**Critique on "Reasoning" Capabilities and Task Trivialization**

The paper asserts that Earth-Agent performs "Autonomous Planning," yet the benchmarks provided (e.g., Figure 11) suggest otherwise. The user queries are overly prescriptive, detailing the exact data sources (Landsat 8), bands (Thermal Band 10, NDVI), and algorithms (single-channel method) required. Consequently, the agent functions more as a code translator than a scientific planner.

Real-world EO challenges involve ambiguity that this dataset seems to strip away. A scientist must decide whether LST is an appropriate proxy for temperature in a specific urban context, or select between competing satellite products (e.g., derived LST vs. standard products like MOD11). By encapsulating complex workflows into high-level tools like lst_single_channel and providing step-by-step instructions in the prompt, the benchmark essentially tests the model's ability to retrieve tools from a list, rather than its capacity for Earth Science reasoning. I am concerned that this setup does not generalize to scenarios where the execution plan is not provided by the user.

Question:
Could the authors clarify how this constitutes "planning"? It appears the model is simply executing a sequence of function calls extracted from the prompt (get_filelist, lst_single_channel, etc.).

---

> ### Author Response · Authors · 2025-12-03
> **Response to Charlie Brown**
>
> Thank you, Charlie Brown. We greatly appreciate your timely attention to our work. We believe there may be several misunderstandings regarding the definitions, and we are eager to clarify them.
>
> **[Clarification: 1]**
>
> `Autonomous Planning Does Not Equate to Task Ambiguity`
>
>
> The concept of autonomous planning lies in the ability to determine a plan, combinations of actions that satisfies a goal in a given domain, which forms the core of an agent's planning ability. However, this does not imply that tasks should be ambiguous. To ensure the uniqueness of the Ground Truth, ***it is necessary to provide a detailed context for the tasks in the agent's domain***. This is a standard formula, as seen in respected datasets such as GAIA[1], where normalized background descriptions are included to ensure the uniqueness of the goal:
>
> ```shell
> Question: If this whole pint is made up of ice cream, how many percent aboveor below the US federal standards for butterfat content is it when using thestandards as reported by Wikipedia in 2020? Answer as + or - a number roundedto one decimal place.
> Ground truth: +4.6
> ```
>
> In our dataset, to ensure that **the Ground Truth is definable and unique**, we provide specific satellite products and Earth science indices. For example, the case you referenced on page 38, Figure 11, requires specifying satellite data and index. ***LST***, ***the single-channel method***, ***summer and autumn*** time period, and ***mean difference*** calculation are all clearly specified, ensuring a unique and calculable Ground Truth. As shown in the following:
>
> ```shell
> Using Landsat 8 thermal and reflectance data in 2022 over New York City, estimate LST based on NDVI and thermal band 10 using the single-channel method, then calculate the average LST for summer andautumn, and determine the mean difference to assess seasonal temperature variation between these two periods.
> A. 8.65 K
> B. 10.89 K
> C. 12.42 K
> D. 14.75 K
> ```
>
> In addition, there are many questions intentionally contain relatively limited background information, because these questions do not specify a target index to be computed. For example:
>
> ```shell
> Question 1：
> Using Landsat 8 TOA data (Band 10) in Death Valley National Park during the 2021 heatwave season (June-September), calculate how many days showed over 50% of the area with land surface temperatures above 315 K .
>
> Question 2:
> Based on brightness temperature and reflectance data from Landsat 8 Band 10, Band 4, and Band 5 on August 11, 2021, in the Okavango Delta ,  calculate the proportion of the area had surface temperature below 300 K.
> ```
>
> All components of our dataset, including the **benchmark**, **evaluation code**, and the **model’s evaluation JSON logs** have been fully open-sourced. You can inspect the detailed reasoning traces of the model for further verification.
>
>
> **[Response to Question]**
>
> As stated in line 337 of the paper, Auto-Planning corresponds to the step-implicit setting  and evaluates the agent’s ability to autonomously plan its solution trajectory. While the input may specify certain data sources or methods to ensure the uniqueness of the answer, the agent needs to determine the sequence of actions (tools & models) necessary to achieve the final answer. This represents planning ablity, where the agent decides how to approach the problem, rather than simply executing predefined steps. Regarding the need for ***selecting between competitive satellite products***, the agent system would require additional background knowledge to address such demands. We acknowledge this and plan to incorporate such considerations into future versions of the dataset and model, ensuring a more comprehensive solution for real-world Earth Observation challenges.
>
> *References:*
>
> [1] Mialon, G., Fourrier, C., Wolf, T., LeCun, Y., & Scialom, T. (2023, October). Gaia: a benchmark for general ai assistants. In The Twelfth International Conference on Learning Representations.

---

### Author Response · Authors · 2025-12-03
**Global Response: Summary of the Rebuttal**

We sincerely thank all the reviewers for their constructive suggestions and thoughtful engagement throughout the discussion period. Their valuable feedback and proposed experiments have greatly contributed to clarifying our contributions. Below, we provide a concise summary of the key points of consensus, significant improvements made, and supporting evidence added during the rebuttal process.
___

## **Commonly Recognized Strengths**

Across reviews, the following strengths were commonly acknowledged:
- ***S1 —*** **Introducing the agent paradigm to address Earth observation tasks is novel, timely, and well-motivated**. (`a78N`, `1ij5`)
- ***S2 —*** **The Earth-Bench benchmark is holistic and practical for subsequent research.** (`a78N`, `38ix`)
- ***S3 —*** **The experiments are comprehensive and clear.** (`a78N`, `1ij5`, `ic4k`, `38ix`)
- ***S4 —*** **The provision of tools covering a broad range of EO tasks represents a valuable contribution to the development of agents in the EO field.** (`ic4k`, `1ij5`)
- ***S5 —*** **The paper is clearly written and well-structured**. (`a78N`, `1ij5`, `ic4k`, `38ix`)
___
## **What Concerns Are Resolved**
| Number | Question/Concern | Raised by | Our Actions |
|--------|-----------------|---------------|-------------|
| 1 | Need to add a detailed description of the composition of Earth-Bench, the curation process, and the background of the experts | `a78N`, `1ij5`, `ic4k` | (1) Add the Earth-Bench data source statistics in **Appendix A.1** and the task type distribution in **Appendix A.3**. (2) Clarify the background of the experts involved. (3) Provide a detailed description of the process in which the experts participated in **Appendix A.5**. |
| 2 | Concerns about the robustness of the Zero-Shot Agent system | `a78N`, `3Bix` | Explain the phenomenon of **cascading errors in agent reasoning** and clarify how we use the **ReAct mechanism** to mitigate losing grounding from the environment during reasoning. |
| 3 | Need to add discussion on the computational efficiency of the Agent framework | `a78N` | We have added a comparison of the latency between Earth-Agent and the general Agent framework in **Appendix B: Latency Experiment**. We also **provide detailed examples of methods used by Earth-Agent to reduce LLM latency**. |
| 4 | Discuss the scalability of the Agent system's performance with respect to the tools. | `a78N` | In **Appendix F: Scalability Discussion**, we have added the scalability trends for ***Accuracy*** with respect to ***Tool Number*** and ***Unique Tool Number***. We provided two key insights: **(1) ensuring the agent startup**, and **(2) constructing a long-chain reasoning dataset**. |
| 5 | Tools and datasets may be customized and dependent | `1ij5` | (1) Clarify that **the construction of the toolset occurred before the dataset**, and provide 22 tools that have never been used in the evaluation to demonstrate that the toolset was not selectively designed for Earth-Bench. (2) Validate Earth-Agent's generalization performance on the **CBGB dataset provided by the reviewer**, showing that it still performs excellently. |
| 6 | Ambiguity and errors in the benchmark dataset cases | `ic4k` | (1) Explain the cases where misunderstandings occurred in the demonstration. (2) Re-tested all 248 tasks in the Earth-Bench dataset and provided the updated results |
| 7 | How does the evaluation scheme for Earth-Agent account for different action sequences and alternative action sequences? | `3Bix` | (1) Clarify that our agent's reasoning process is evaluated across three dimensions: ***Tool-Any-Order***, ***Tool-In-Order***, and ***Tool-Exact-Match***, which **ensures tolerance** for different action sequence paths. (2) Explain that we avoid rewarding agents for alternative reasoning chains based on final results, as this would take into account **false positive reasoning chains**. |
| 8 | Comparison between Earth-Agent and SciToolAgent | `3Bix` | (1) **Comparison of Benchmarks**: Earth-Bench involves multi-step reasoning and multiple tools, with a higher task complexity. (2) **Comparison of Method**: Earth-Agent employs FastMCP & ReAct, enabling dynamic logical reasoning and greater scalability. |
___
## **Summary**
Earth-Agent pioneered the introduction of the Agent paradigm within Earth Science field, unlocking the full landscape of Earth domains: Spectrum, Products, and RGB imagery through comprehensive and extensive experiments.During our rebuttal, we supplemented the precise evidence requested by reviewers, including latency experiments for the Agent system, expert background validation, generalisation verification, and clarified ambiguities within the paper. After the rebuttal, it has become a more complete precedent in this field!

We appreciate your pivotal role in this final assessment and thank you for considering the substantial evidence provided in our rebuttal.

Best regards,

The Authors

---

### Meta-Review · Program_Chairs · 2026-01-09

**Summary:**

This paper presents a detailed VQA benchmark task for evaluating the performance of agents in Earth observation tasks, when the agent has access to existing specialized tools for Earth observation. The reviewers are generally positive, but one raised concerns about whether several significant errors in the framing of the benchmark (as illustrated in the examples given) could point to deeper problems in how the benchmark was designed and whether the experts consulted were sufficiently careful and knowledgeable about the relevant problem space. This should be addressed in the camera-ready.

**Reviewer Concerns:**

Reviewer a78N (score: 8)
- Writes that they will keep their score.

Reviewer 1ij5 (score: 6)
- Wants to learn more about how the benchmark was designed, including how the tools were chosen. Fully addressed - the authors give an in-depth response.

Reviewer 38ix (score: 6):
- Some agents hallucinate tools or data. Partly addressed - the authors discuss this, but agree it's an issue.
- Are the metrics penalizing agents that get the right answer for a good logical reason that is not the *expected* logical pathway? Partly addressed - the authors discuss why they picked their metric, but say that fundamentally yes this could happen.

Reviewer ic4k (score: 4):
This is the primary objection to accepting the paper, and they have iterated extensively with the authors and give a very compelling argument for not changing their score despite the authors' responses.
Their main concern, and the one which persists through the author-reviewer discussion, is that they identify several major errors in the dataset, even in the example problems listed by the authors. These errors should arguably have been obvious to the supposed experts creating and proofreading the dataset, and also point to issues in how the original data pipelines were created by the authors since data filtering wasn't correctly performed. While the authors check for additional errors of the form observed and it doesn't change the overall performance much (a) they *do* find more errors (5% of the dataset, which is significant!), (b) it suggests that the whole process could have been flawed from the start, including in ways that aren't as clear.

**Reviewer Scores:**

The critical reviewer is ic4k, who has explicitly said that they will not change their score.

Reviewer 38ix (while overall positive about the paper) also raises good concerns that are not mostly, in my view, not addressed by the authors.

---

### Decision · Program_Chairs · 2026-01-26

Accept (Poster)